# Dysregulation of PRMT5 in chronic lymphocytic leukemia promotes progression with high risk of Richter's transformation

Zachary A. Hing[1,13], Janek S. Walker [1,13], Ethan C. Whipp [1,13], Lindsey Brinton[1], Matthew Cannon[1], Pu Zhang[1], Steven Sher [1], Casey B. Cempre[1], Fiona Brown[1], Porsha L. Smith[1], Claudio Agostinelli[2], Stefano A. Pileri[3,4], Jordan N. Skinner[1], Katie Williams[1], Hannah Phillips[1], Jami Shaffer[1], Larry P. Beaver[1], Alexander Pan[1], Kyle Shin[1], Charles T. Gregory [1], Gulcin H. Ozer[5], Selen A. Yilmaz[5], Bonnie K. Harrington[1,6,7], Amy M. Lehman[8], Lianbo Yu[8], Vincenzo Coppola [9], Pearlly Yan [1], Peggy Scherle[10], Min Wang[10], Philip Pitis[10], Chaoyi Xu[10], Kris Vaddi[10], Selina Chen-Kiang[11], Jennifer Woyach [1], James S. Blachly [1], Lapo Alinari[1], Yiping Yang[1], John C. Byrd[1,12], Robert A. Baiocchi[1], Bradley W. Blaser [1,13] & Rosa Lapalombella [1,13] ✉

Richter's Transformation (RT) is a poorly understood and fatal progression of chronic lymphocytic leukemia (CLL) manifesting histologically as diffuse large B-cell lymphoma. Protein arginine methyltransferase 5 (PRMT5) is implicated in lymphomagenesis, but its role in CLL or RT progression is unknown. We demonstrate herein that tumors uniformly overexpress PRMT5 in patients with progression to RT. Furthermore, mice with B-specific overexpression of hPRMT5 develop a B-lymphoid expansion with increased risk of death, and Eμ-PRMT5/TCL1 double transgenic mice develop a highly aggressive disease with transformation that histologically resembles RT; where large-scale transcriptional profiling identifies oncogenic pathways mediating PRMT5-driven disease progression. Lastly, we report the development of a SAM-competitive PRMT5 inhibitor, PRT382, with exclusive selectivity and optimal in vitro and in vivo activity compared to available PRMT5 inhibitors. Taken together, the discovery that PRMT5 drives oncogenic pathways promoting RT provides a compelling rationale for clinical investigation of PRMT5 inhibitors such as PRT382 in aggressive CLL/RT cases.

Definitive molecular mechanisms driving the establishment and progression of chronic lymphocytic leukemia (CLL) have yet to be fully understood. Contributing to this lack of understanding, select patients experience a more aggressive disease course in the absence of known high-risk genetic alterations, suggesting the existence of some undiscovered factors influencing CLL pathogenesis and outcome.

Richter's transformation (RT) is the morphologic evolution from CLL/small lymphocytic lymphoma to a high-grade diffuse large B-cell (DLBCL) or immunoblastic lymphoma, occurring in up to 10% of CLL patients. RT is the most common disease progression in CLL patients within the first 18 months of receiving targeted therapies (i.e., ibrutinib, venetoclax), with overall incidence expected to rise as more patients receive these treatments in the front line and relapse/refractory setting[1,2]. Unlike de novo DLBCL, generally chemotherapy-responsive and potentially curable, RT is chemotherapy-resistant and associated with a dreadful prognosis[3]. While initially described in 1928,

the molecular, genetic, and epigenetic events driving RT remain poorly characterized. At transformation, between 25 and 60% of RT tumors display inactivation of TP53 and ~50% acquire epigenetic changes affecting c-MYC expression, suggesting dysregulation of these pathways contribute to the pathogenesis of RT[4–7]. Complex karyotype, *CDKN2A/B* gene locus deletion, and aberrant NOTCH1 activation are also associated with RT[8–11]. While these discoveries contributed to the understanding of RT biology, comprehensive studies underlying CLL-to-RT progression are needed to identify patients at risk of transformation and to discover targeted therapeutic approaches.

Arginine methylation regulates critical cellular processes, including proliferation and differentiation[3,12–18]. The major type II protein arginine methyltransferase 5 (PRMT5) mediates symmetric dimethylation by catalyzing the transfer of a methyl group from S-adenosylmethionine (SAM) to arginine residues on histones (i.e., H3 and H4) and non-histone proteins [e.g., retinoblastoma (Rb), p53, and Sm proteins of the spliceosome] thereby influencing gene transcription, splicing[19], and protein activity. PRMT5 is essential for the eIF4E-mediated 5′-cap-dependent translation of c-MYC and CYCLIN D1[20] and is required for the establishment and maintenance of c-Myc and NOTCH1-driven lymphomas[21,22]. Moreover, co-deletion of *MTAP* and *CDKN2A* as a result of del(9)(p21.3), seen in up to 30% of RT tumors[23,24], has been shown to confer vulnerability to PRMT5-targeted therapies, suggesting dependence on PRMT5 for the proliferation of these tumors[25–27]. While literature strongly supports a requisite role for PRMT5 in the context of large cell lymphomas, emerging evidence now suggests aberrant PRMT5 expression additionally disrupts critical regulatory mechanisms and contributes to tumor progression in CLL[28]. Accordingly, we hypothesized mechanisms stemming from dysregulated PRMT5 activity to play a major role in facilitating CLL-to-RT evolution.

We herein demonstrate that RT tumors overexpress PRMT5, and PRMT5 upregulation may be observed in some cases up to one year prior to RT diagnosis. We then developed and characterized a murine model of B-cell-specific expression of human PRMT5 (Eμ-PRMT5), producing an overt B-lymphoid expansion eventually developing a CLL-like phenotype. When crossed with the Eμ-TCL1 model, double transgenic mice uniformly developed a highly aggressive disease with select focal lymphoid tumors transforming to histologically resemble RT, altogether proposing a direct cause-effect relationship for PRMT5 in the pathogenesis of RT and identifying an attractive therapeutic target for RT and CLL at risk of RT. Lastly, we report the development of a small molecule inhibitor (PRT382) with enhanced PRMT5 selectivity, improved tolerability, and optimal in vitro and in vivo antitumor activity compared to currently available PRMT5 inhibitors.

## Results

### PRMT5 is overexpressed in CLL patients progressing with RT

Serial peripheral blood samples (PBMCs) from CLL patients and lymph node (LN) biopsy with PBMCs was collected for those with progression to RT. In a CLL patient with three previous lines of therapy, we obtained PBMC and LN samples at RT diagnosis in addition to CLL-phase PBMCs 2 months prior to RT and conducted single-cell RNA-sequencing (scRNA-seq; Fig. 1a). By evaluating VDJ regions across all three samples we observed >99% of all B cells (CD79A+) sequenced contained identical VDJ sequences, confirming the RT-phase tumor in this patient derived as a clonal progression from the existing CLL (Supplemental Fig. 1A). Combining samples for joint clustering of transcriptionally-related cell populations by uniform manifold approximation and projection (UMAP) criteria, we observed a divergence in the transcriptional landscape and an increase in population density in B cells occupying both the PBMC and LN compartments at the time of RT diagnosis relative to previously acquired CLL-phase PBMCs (Fig. 1a). Markers to define immune cell type in CLL and RT samples are described in Supplemental Fig. 1B. Divergent B-cell populations were notably observed with elevated expression of

*PRMT5*, an epigenetic modifier with an established role in B-cell lymphomagenesis.

Upon this observation, we next evaluated the PRMT5 protein expression in CLL cells from the blood of patients that eventually did (CLL pre-RT) or did not (CLL) transform to RT within one year from the time the sample was collected. PRMT5 expression was variable in CLL patients who did not undergo transformation, ranging from not expressed to weakly expressed, whereas PRMT5 was consistently overexpressed in pre-RT patients (Fig. 1b). We then performed tissue microarray studies with lymph node biopsies from 70 CLL cases—classified by proliferation center (PC)-rich or typical-PC distribution[29,30]—and 15 RT cases. PC-rich CLL cases were defined as those with confluent PCs whereas typical CLL cases showed scattered, small, ill-defined PCs in a monotonous background of small, relatively round lymphocytes (small PC). The small lymphocytes within typical CLL cases stained negative for PRMT5 expression, however, small PC areas showed weak PRMT5 expression (Fig. 1c—pink stain). PC-rich CLL cases contained weak to moderate positivity for PRMT5 in prolymphocytes (PL) and paraimmunoblasts (PI). CLL LN tumors were intermittently populated with small CD3+ T cells with distinct cytologic morphology from small-to-medium-sized tumor blasts with PRMT5 positivity (Supplementary Fig. 1C). In stark contrast from CLL tumors, all 15 RT cases showed strong positivity for PRMT5 (Fig. 1c). Further evident at ×400 magnification, the strongly PRMT5+ medium-sized tumor blasts in RT LN tumors demonstrate cytology explicitly divergent from CLL LN cases with limited penetrance of small lymphoid cells, indicating these tumors were largely comprised of PRMT5+ large lymphocytes and not small lymphocytes from the tumor microenvironment (Supplementary Fig. 1D).

To further elucidate a role for PRMT5 in RT, we evaluated scRNA-seq in LN biopsies from two patients collected at the time of RT diagnosis (Fig. 1d and Supplemental Fig. 1E). Notably, "RT Pt 2" possessed a chromosomal rearrangement with the *MYC* locus as identified by FISH while "RT Pt 1" did not present any identifiable rearrangement. Transcriptional heterogeneity was observed in B cells between samples; noting that *PRMT5* was not abundantly expressed in all B cells, but was instead restricted to distinct subpopulations also displaying strong *MYC* and *MKI67* expression (Supplemental Fig. 1F), indicative of a highly proliferative population within the overall tumor burden. Clustering of cells with strong *PRMT5* and *MYC* expression was observed in both patients regardless of identifiable MYC rearrangement.

Understanding the limitations of evaluating this scRNA-seq dataset from tumors of just two RT patients, we compared the expression signature observed here with those that have been previously defined in the literature. We obtained CLL and RT RNA-seq gene expression data available from a recent study by Nadeu and colleagues[31] and plotted the aggregate gene expression profile defined therein against CLL PBMCs, RT PBMCs, and RT LN tumor scRNA-seq expression profiles from the two patients described in our study (Fig. 1e and Supplemental Fig. 1E). Using this approach, we observed that the CLL-specific gene signature predominantly mapped to B-cell populations enriched in CLL PBMCs and subsets of similar RT PBMC & LN cell populations described herein. More importantly, the published RT-specific gene signature exclusively mapped to B-cell populations in RT LN tumors in our study, supporting the RT-specific B-cell gene signature observed in our study as in concordance with previously defined RT gene expression profiling.

To provide a more robust description of the cell populations comprising the LN tumors, we implemented Leiden-based clustering to label cell populations with similar transcriptional activity. We identified 18 distinct clusters within B, T, and monocytic cells comprising the RT LN (Fig. 1f). Ten distinct clusters were identified in the B-cell compartment with varying abundance, however, clusters 1 and 9 were predominantly absent in RT LN tissue while clusters 3 and cluster 11 were exclusively observed in the RT-phase and were enriched in the

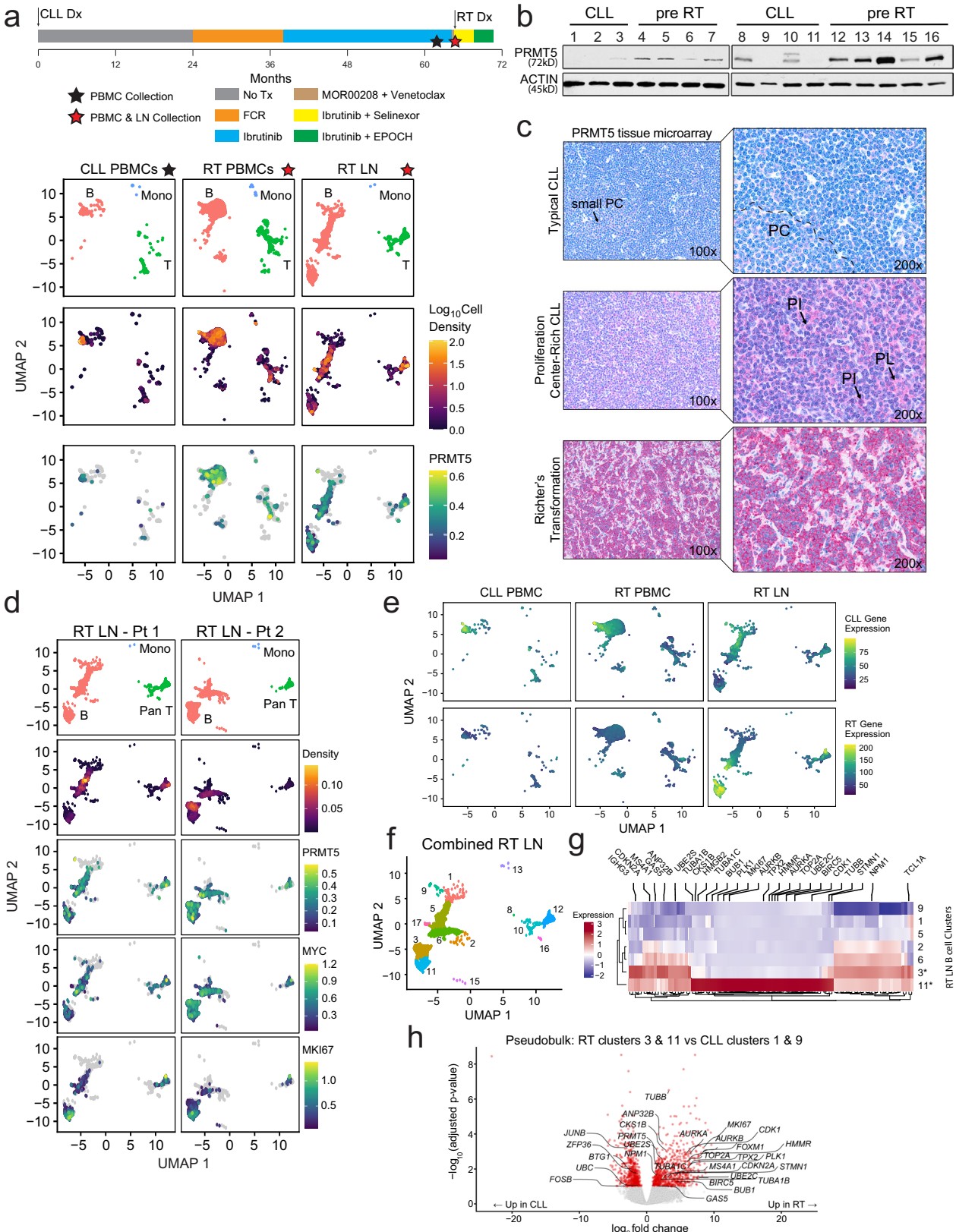

LN tissue (Supplemental Fig. 1G). Evaluating gene expression enriched in RT LN clusters 3 and 11 amongst the most abundant clusters in the LN B-cell compartment, we observed enrichment of genes including *BIRC5* (*survivin*), *CDK1*, and *TCL1A*, among others, likely contributing to CLL-to-RT progression by stimulating proliferation and anti-apoptotic signaling with enhanced activity along the BCR-signaling axis (Fig. 1g).

In concordance with *MKI67* expression indicating highly proliferative cells, we also observed enrichment for genetic markers of proliferating cells, including *TOP2A* and *HMGB2*, ubiquitin-conjugating enzyme genes *UBE2C/S*, and tubulin-encoding genes *TUBA1B/C*.

The shift in UMAP coordinates between RT clusters 3 and 11 and CLL clusters 1 and 9 suggested distinct global transcriptional profiles.

**Fig. 1 | PRMT5 is overexpressed in RT and in CLL undergoing RT. a** Patient 1 (Pt 1) timeline. ScRNA-seq UMAP plots stratified by disease/tissue visualizing cell type clusters, local $\log_{10}$(cell density), and PRMT5 expression. Cell density color scale is shown as the Log10-transformed cell count within hexagonal bins. PRMT5 expression is shown as the Log10-transformed expression where expression is size factor normalized UMI counts. **b** PRMT5 expression in CLL B cells via western blot. Patient samples were collected during the CLL-phase ($n = 16$) and retrospectively identified as those eventually progressing (pre-RT) or not (CLL) to RT within one year from the time of collection. **c** Representative PRMT5 staining by tissue microarray in lymph node biopsies from 70 CLL cases (typical distribution of proliferation centers or proliferation center-rich) and 15 RT cases. PL prolymphocytes, PI paraimmunoblasts. **d** ScRNA-seq of two RT patient tumors collected at the time of RT diagnosis. UMAP plots stratified by RT patient LN biopsy sample visualizing cell type clusters; cell density; and expression of PRMT5, MYC, and MKI67. Cell types were assigned by cell type markers, Supplemental Fig. 1B. Cell density is calculated as the 2d kernel density estimate mapped to color scale. Gene expression is shown as the Log10-transformed expression where expression is size factor normalized UMI counts. **e** Aggregate gene expression of CLL and RT-specific genes identified from literature (Nadeu et al. 2022, Supplemental Table 11b)[31] in longitudinal samples (Pt 1). Expression is shown as the Log10-transformed size factor normalized UMI counts scaled for each gene module. **f** RT Pt 1 and Pt 2 nodal cells with Leiden clusters distinguished by color and number. **g** Heatmap highlighting the top 50 enriched genes in RT-specific nodal B-cell clusters 3 and 11, from Pt 1 and Pt 2 shown in (**d, f**). Gene expression across 100 genes is shown as the average Log10-transformed expression in nodal B-cell clusters. Genes associated with leukemia and lymphoma are highlighted. **h** Pseudobulk differential gene expression between RT LN (3 & 11) and CLL (1 & 9) enriched B-cell clusters. Highlighted genes have adjusted *P* value <0.1 and fold change >1.5. Source data are provided as a Source Data file.

To evaluate these transcriptional alterations, we performed pseudo-bulk analysis to further identify differentially regulated genes between RT and CLL tumorigenic B cells (Fig. 1h). Grouping RT clusters 3 & 11, we observed upregulation of *PRMT5* and pro-proliferative genes, including *FOXM1*, while CLL clusters 1 & 9 upregulated AP-1 complex members (*JUNB, FOSB*). Gene set enrichment analysis revealed that RT cells were in a proliferative state with enhanced *MYC* and *E2F* activity while the CLL cells maintained upregulated interferon-gamma response signaling genes, likely in support of their growth and survival[32] (Supplemental Fig. 1H).

**Eμ-PRMT5 mice develop a clonal B-cell-lymphoproliferative disorder**
Ablation of PRMT5 prevents lymphomagenesis in the Eμ-Myc mouse model by spurring aberrant pre-mRNA splicing events[21]. However, in vivo effects of B-cell-specific PRMT5 overexpression are unknown. Thus, we generated a transgenic mouse expressing human PRMT5 in immature and mature B cells (Eμ-PRMT5) under the control of a $V_H$ promoter-Ig$_H$-Eμ enhancer (Fig. 2a). Three transgenic founders (namely M806, M807, and L25) were generated and bred to establish separate mouse lines, and line L25 was selected for full characterization. The median survival time for L25 Eμ-PRMT5 mouse progeny ($n = 67$) was significantly shorter compared to wild-type (WT) littermates (Fig. 2b); with 40/75 (53%) mice meeting early removal criteria (ERC) due to signs of clinical deterioration (Fig. 2c). Starting at 5 months of age, peripheral blood from a cohort of mice ($n = 26$) was collected monthly and monitored for disease development by flow cytometry. Abnormal expansion over time of hematopoietic cell populations was commonly observed in Eμ-PRMT5 mice, including a Cd5+Cd19+B220dim B-cell population with similarity to the CLL-like disease in Eμ-TCL1 mice[33] (Fig. 2d, e and Supplemental Fig. 2A, B). This CLL-like expansion is a rare occurrence in most WT mouse strains, and was not observed in the WT mice in our study[34]. Cd23, Cd11b, and Cd49b expression was consistently low in these circulating CLL-like cells, supporting the interpretation of lymphoid lineage.

Upon euthanization of symptomatic mice, nodal and extranodal tissues were collected and examined. The histopathologic analysis demonstrated evidence of hematopoietic neoplasia in 4/11 Eμ-PRMT5 mice (Fig. 2f). Overall, inter-specimen variability in morphology of the neoplastic cells was observed, ranging from anaplastic large lymphocytes to small lymphocytes commonly invading the spleen, lymph nodes, lung, liver and kidney, although not all tissues were affected in every mouse. Mice numbers L31, L35, and L36 contained anaplastic large lymphocytes (L, appear blue-purple) and bland histiocyte infiltrates (H, light pink cytoplasm). Mouse L32 showed sheets of small lymphocytes (S) in the lymph nodes and spleen, consistent with a small lymphocytic lymphoma/chronic lymphocytic leukemia. Immunohistochemistry in Eμ-PRMT5 mice further demonstrated these tumors were largely composed of a background of large B220+ B cells with scattered Cd3+ T cells and F4/80+ macrophages (Fig. 2g). Overall, this mixture of cell types is consistent with the murine counterpart of the human disease T-cell and Histiocyte-rich large B-cell lymphoma.

**Eμ-PRMT5 tumors contain a CLL-like gene expression signature**
We next performed RNA-sequencing to evaluate the transcriptome and BCR gene usage[35] in Cd5+Cd19+ cells (Supplemental Fig. 2B) isolated from the spleen of Eμ-PRMT5, Eμ-TCL1 and WT mice. Predominant usage of distinct kappa and heavy chain genes in each Eμ-PRMT5 and Eμ-TCL1 mouse, but not WT mice, was observed (Fig. 2h and Supplemental Fig. 2C), indicating an expansion of cells with minimal clonal diversity. In concordance with the observed frequency of CLL-like development in Eμ-PRMT5 mice, aberrant *PRMT5* expression as the sole transgenic event appeared as a less-robust murine leukemogenic driver as a greater degree of clonal diversity was observed in Eμ-PRMT5 mice when directly compared to the B-cell compartment of Eμ-TCL1 mice. A low mutational burden in the *IgHV* gene sequence (<2%) was observed in the dominant clones across all samples, consistent with classification as *unmutated IgHV* (Supplemental Fig. 2D). Gene expression analysis showed minimal differentially expressed genes (DEGs; $n = 288$ in Eμ-PRMT5, $n = 222$ in Eμ-TCL1) between Eμ-PRMT5 and Eμ-TCL1 splenic B cells, suggesting a similar transcriptome profile within CLL-like cells in these two models (Fig. 2i). Among these few DEGs we observed upregulation of *Arg2, Cxcl16, Elane, Il1r2,* and *Zap70* in the Eμ-PRMT5 tumors, and enrichment of *Plk2* and *Fos/Jun* family members in Eμ-TCL1 tumors. Transcriptional silencing of *PLK2* is a frequent event in B-cell malignancies[36], and associated with increased cell proliferation and decreased apoptosis in B-cell lymphoma.

To expand our study of DEGs in murine CLL-like cells and their surrounding tumor microenvironment, unselected live cells from spleens of Eμ-PRMT5 and Eμ-TCL1 mice at ERC (Supplemental Fig. 2B) were processed for 3' scRNA-seq. ScRNA expression profiles from both models were combined and visualized by UMAP criteria. Analysis of 30,171 Eμ-PRMT5 and 25,886 Eμ-TCL1 spleen cells revealed graph-based clustering of diverse cell populations with distinct transcriptional activity, including B- and T-lymphocytes, monocytes and neutrophils with varying population density (Fig. 3a and Supplemental Fig. 2E). Cd19+/Cd5+ "CLL-like" cell populations were observed in both Eμ-PRMT5 and Eμ-TCL1 mice but did not comprise the entire B-cell compartment, yet clustered in regions with the highest cell density within each model (Fig. 3a, b). Diffuse distribution of *IgM* and *IgD* expressing *Pax5*+ B lymphocytes was observed between both models, whereas *IgE*+ *Pax5*+ expression was only observed in a small subset of Eμ-PRMT5 B lymphocytes (Fig. 3c). B cells in both models presented with *Cd43* (*Spn*) and *Cd69* positivity, indicating these as activated B1a lineage lymphocytes and supporting description as a CLL-like phenotype. *Cd93* expression was exclusively observed in a subset of Eμ-TCL1 B cells. By K-means clustering, $n = 10$ distinct cell clusters were

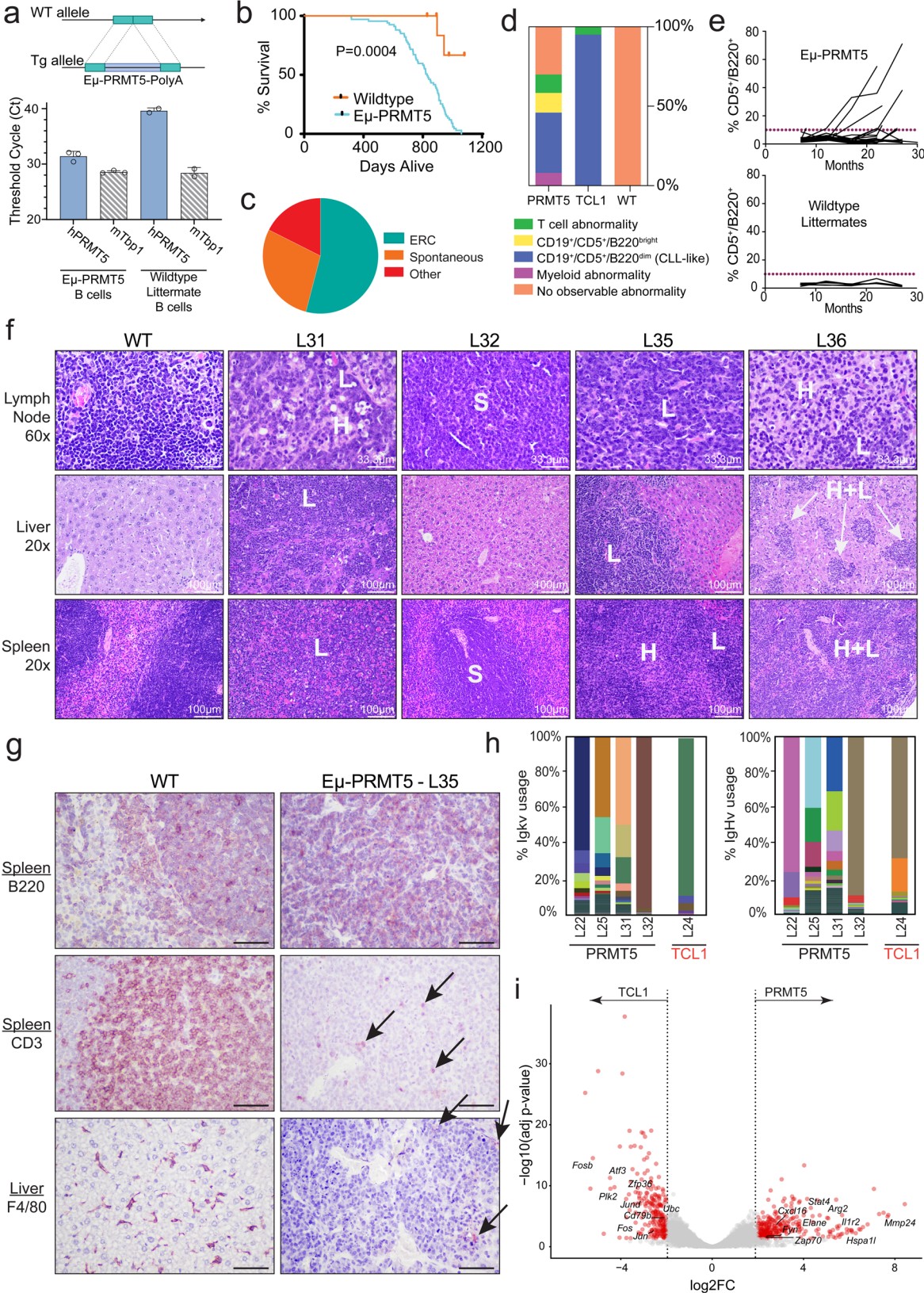

observed in varying ratios within Eµ-PRMT5 and Eµ-TCL1 spleen populations (Fig. 3d). Focusing on B-cell populations, clusters 3.4 and 3.6 appeared enriched in cells from the spleen of Eµ-PRMT5 mice, whereas clusters 3.2 and 3.3 appeared enriched in cells from the spleen of Eµ-TCL1 mice. Eµ-PRMT5-enriched clusters 3.4 and 3.6 noticeably presented elevated expression of *Cd69* and *Myc* (Supplemental

Fig. 2F), and further, *Mki67* expression was observed to be restricted to cluster 3.6, presenting a population of *IgM+, Pax5+, Ebf1-*, and *Myc+* highly proliferative transformed B lymphocytes most evident in Eµ-PRMT5 mice.

Exploring genes most enriched in clusters 3.4 and/or 3.6 in addition to *Myc* and *Cd69*, we also observed upregulation of *Atf4, Ccna2,*

**Fig. 2 | Overexpression of PRMT5 induces lymphoid proliferation in vivo.**
**a** Schematic representation of the construct used to generate the Eμ-PRMT5 transgenic mouse. RT-qPCR for *hPRMT5* in B cells of Eμ-PRMT5 mice and WT littermates. mTbp1 shown as control. Circles indicate individual mice ($n = 3$, Eμ-PRMT5 mice; $n = 2$, WT mice). Bars indicate mean ± SD. **b** Eμ-PRMT5 mice ($n = 67$) and WT littermates ($n = 10$) survival. Eμ-PRMT5 mice median survival = 829 days. Kaplan−Meier plot statistics: Mantel−Cox test [$P = 0.0004$, hazard ratio = 0.12 95% CI (0.069−0.23)]. **c** Cause of death (COD) observations for Eμ-PRMT5 transgenic mice ($n = 75$). Categorical COD was recorded as either spontaneous death or meeting predefined euthanasia criteria. Unresolved cause of death cases were grouped as "other". **d** A cohort of Eμ-PRMT5 ($n = 26$), Eμ-TCL1 ($n = 35$), and WT littermate mice ($n = 9$) were assessed monthly for spontaneous disease expansion by flow cytometry of peripheral blood. Expansive populations of T cells (Cd45+/ Cd5+), B cells (Cd19+/Cd5+/B220$^{bright}$), CLL-like cells (Cd19+/Cd5+/B220$^{dim}$), or myeloid cells (SSC$^{high}$ & Cd19−/Cd5−) were observed in varying ratios within groups.

**e** Rate of CLL-like cell accumulation in Eμ-PRMT5 and WT littermate mice (from **d**). The dotted line indicates 10% Cd5+ /B220+ cells in the blood observed via flow cytometry. **f** Histopathology analysis via hematoxylin and eosin (H&E) staining of Eμ-PRMT5 ($n = 4$) mice at ERC and age-matched WT littermate ($n = 3$). L anaplastic large lymphocytes, H bland histiocytes, S sheets of small lymphocytes. **g** Representative immunohistochemistry of tumors from Eμ-PRMT5 ($n = 5$) mice. Arrows indicate scattered CD3+ and F4/80+ cells. This mixture of cell types is consistent with the diagnosis of histiocyte-associated lymphoma, which is the murine counterpart of the human disease T-cell and Histiocyte-rich B-cell lymphoma. B220 and CD3 shown in spleen, F4/80 shown in liver. Images shown at ×40 magnification. Scale bar is 50 μm. **h** B-cell receptor variable gene usage in splenic CLL-like cells from Eμ-PRMT5 ($n = 4$) and Eμ-TCL1 ($n = 1$) mice evaluated via RNA-seq. Colors indicate BCR gene usage. **i** Bulk RNA-seq in splenic CLL-like cells from Eμ-PRMT5 (from **h**) and Eμ-TCL1 mice ($n = 3$). Red highlighted genes have adjusted *P* value <0.05 and fold change >4. Source data are provided as a Source Data file.

*Ccr7*, *Junb*, *Mki67*, *Nr4a1*, and *Top2a*, among other genes commonly identified In proliferating tumors (Fig. 3e). Deregulated expression of AP-1 transcription factors (*Junb*) is implicated in the pathogenesis of several large cell lymphomas[37], suggesting a role for hyperactive MAPK activity in CLL progression. Clusters 3.4 and 3.6 also featured upregulation of genes implicated in large cell lymphomas, and notably overlapping with several genes enriched in RT tumors compared to matched CLL precursors, including *Birc5*, *Cdk1*, *Mki67*, and *Npm1* (Supplemental Fig. 2F). Overall, enrichment of genes in clusters 3.4 & 3.6 contributed to the activation of pathways including B-cell development and signaling, innate & adaptive immune signaling, DNA damage response, glycolysis, and cell cycle control (Supplemental Fig. 2G).

## PRMT5 accelerates CLL progression and promotes transformation to aggressive lymphoma in vivo

To provide a more robust exploration of the contribution of PRMT5 in CLL disease progression, L25 Eμ-PRMT5 mice were crossed to Eμ-TCL1 mice to generate double transgenic animals (Eμ-PRMT5/TCL1). All Eμ-PRMT5/TCL1 mice between ages 6–15 months became visibly ill and developed a neoplasia with a similar CLL-like immunophenotype to Eμ-TCL1 mice; although Eμ-PRMT5/TCL1 mice frequently presented with large cells circulating in the blood (FSC$^{high}$), a rare event in Eμ-TCL1 mice (Supplemental Fig. 3A, B). While the large cells in Eμ-PRMT5/TCL1 mice maintained the Cd19+ /Cd5" "CLL-like" phenotype, these cells were predominantly B220$^{bright}$, a feature consistent with murine models of large cell lymphoma[35], in contrast with B220$^{dim}$ expression observed on the "CLL-like" small cells in both Eμ-PRMT5 and Eμ-TCL1 mice. The leukemic population appeared to completely replace other Cd45+ populations, such as Cd5+ T cells and Cd19-/Cd5- myeloid cells, seen in WT and Eμ-TCL1 mice (Fig. 4a). Time to disease onset and survival were markedly reduced in Eμ-PRMT5/TCL1 compared to Eμ-TCL1 mice (Fig. 4b). Reduction in survival was also observed in Eμ-PRMT5/TCL1 crosses derived from the two additional PRMT5 founder lines (Supplemental Fig. 3C).

To interrogate the timeline of disease progression between double and single transgenic models, we conducted interim analysis of 3 and 6-month-old Eμ-PRMT5/TCL1 and Eμ-TCL1 mice. By flow cytometry analysis of the blood and spleen at 3 months of age, both Eμ-PRMT5/ TCL1 and Eμ-TCL1 mice were observed with Cd11b-/Cd3-/Cd19+ B cells comprising the majority of these compartments (Supplemental Fig. 3D). Varying Cd3+ T cell and Cd11b+ myeloid populations were found in both models, with limited evidence of Cd19+/Cd5+ "CLL-like" disease emerging at this age. By 6 months of age, prominent expansion of "CLL-like" cells could be identified in both blood and spleen of Eμ-PRMT5/TCL1 mice but not in Eμ-TCL1 mice. The "CLL-like" expansion appeared to completely dominate the blood and spleen compartment, outcompeting both T cell and myeloid populations. Similar T cell and myeloid populations were observed in Eμ-TCL1 mice at 6 months compared to 3-month Eμ-PRMT5/TCL1 and Eμ-TCL1 mice.

Similarly, by histopathology analysis, at 3 months both Eμ-PRMT5/ TCL1 and Eμ-TCL1 mice showed evidence of emerging germinal center hyperplasia of the spleen (Fig. 4c, dashed circles; and Supplemental Fig. 3E). By 6 months of age, effacement of normal architecture including loss of germinal centers and indistinct splenic white pulp with rare foci of neoplastic lymphocytes encroaching upon splenic white pulp was exclusively present in the spleens of Eμ-PRMT5/TCL1 mice (arrows), whereas Eμ-TCL1 mice revealed persistent hyperplasia in these regions (Supplemental Fig. 3F). This finding was confirmed via Ki67 staining, which highlights the proliferative germinal centers (Fig. 4c). These data suggested leukemic cells penetrating the spleen of the Eμ-PRMT5/TCL1 animals produced a more aggressive phenotype, replacing normal B and T lymphocytes with rapidly progressing leukemic cells.

In addition, lymph nodes from Eμ-PRMT5/TCL1 mice were more severely affected, demonstrating hyperplasia and increased B-cell proliferation in lymphatic germinal centers at 3 months of age which progressed to effacement of lymphoid architecture by 6 months, as evident by scant regions of cortex with few germinal centers (Fig. 4c, dashed circles). Hyperplastic involvement of lymph node germinal center tissue was infrequently observed in Eμ-TCL1 mice at 3 months and regularly observed in analyzed mice at 6 months of age.

With aging, key clinical characteristics of Eμ-PRMT5/TCL1 mice at ERC included pronounced cervical lymphadenopathy (Fig. 4d) and palpable splenomegaly. Mandibular, mesenteric, and thoracic lymph nodes were markedly enlarged in Eu-PRMT5/TCL1 mice, a hallmark of human RT. By contrast, even mildly enlarged mandibular or other peripheral, thoracic and abdominal lymph nodes were rarely observed in Eμ-TCL1 mice. In general, histopathology analysis of tissues from Eμ-TCL1 and Eμ-PRMT5/TCL1 mice reaching endpoint criteria demonstrated a similar histologic phenotype, observing intermediate-sized lymphocytes with central nuclei containing stippled chromatin and scant basophilic cytoplasm infiltrated the spleen, liver, and lung (Fig. 4e). Notably, a partial or complete transformation to a diffuse large cell phenotype in the lymph node was observed in 2/5 analyzed Eμ-PRMT5/TCL1 mice, in contrast to the densely packed lymphocytes observed in the lymph nodes of Eμ-TCL1 mice. In involved lymph nodes and salivary glands, three-dimensional structures were completely destroyed and not histologically recognizable.

## Early transcriptional changes in Eμ-PRMT5/TCL1 B cells

To interrogate the transcriptional changes contributing to a more aggressive disease phenotype in the Eμ-PRMT5/TCL1 model, we performed RNA-seq in B cells isolated from spleens of Eμ-PRMT5/TCL1 and Eμ-TCL1 mice at 3 months and 6 months of age – prior to accumulation of a lethal circulating disease (Fig. 5a, b). Consistent with the histologic evaluation, we identified 2412 DEGs between 3-month and 6-month Eμ-PRMT5/TCL1 samples, whereas no DEGs were identified between 3- and 6-month Eμ-TCL1 samples. No DEGs were found between 3-month

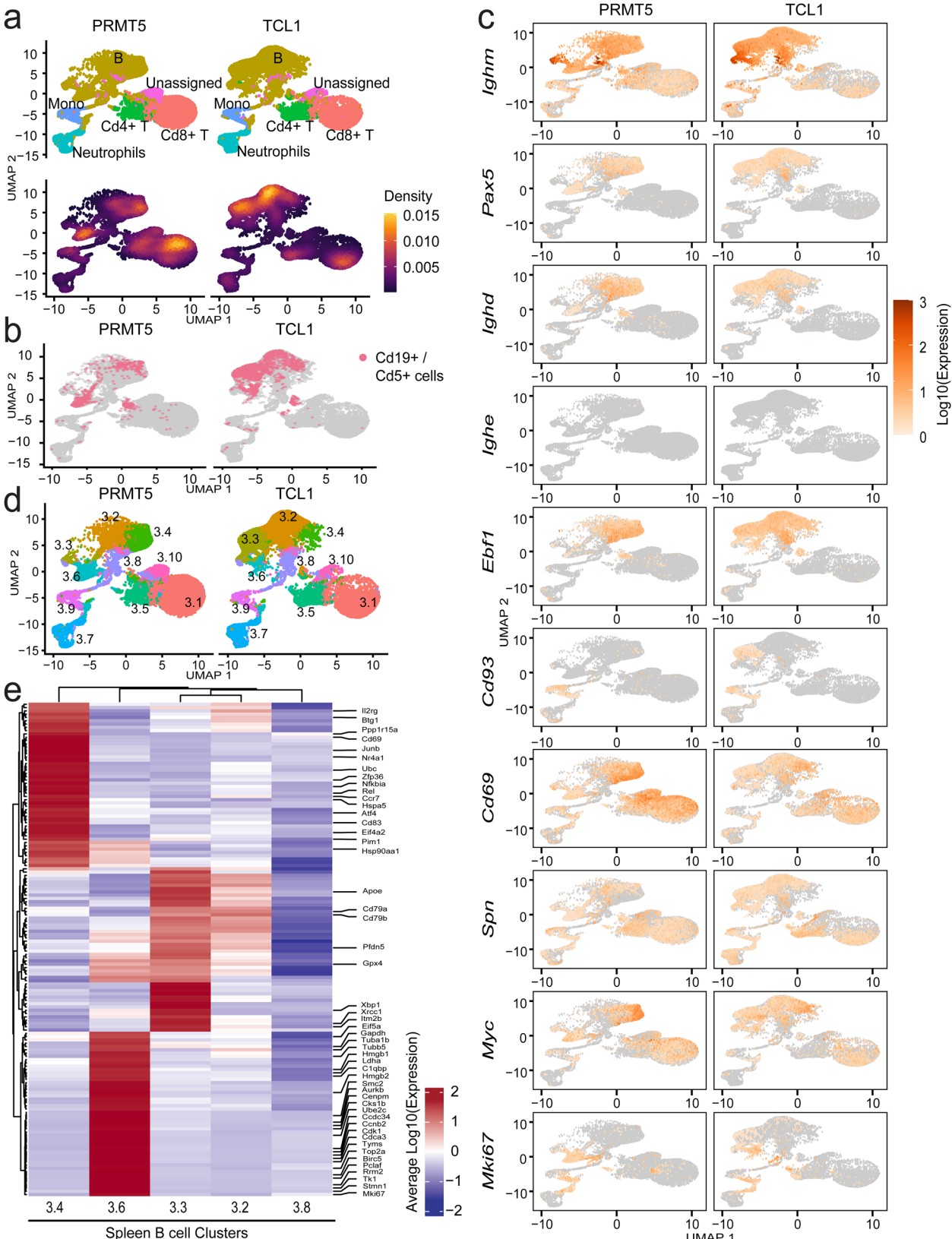

Eμ-TCL1 and Eμ-PRMT5/TCL1 samples. In contrast, a total of 2167 genes (807 upregulated and 1360 downregulated) were dysregulated in Eμ-PRMT5/TCL1 samples compared to all other groups analyzed (Eμ-TCL1 at 3 and 6 months, Eμ-PRMT5/TCL1 at 3 months). Among these, we identified *Myc, Il-10, Bcl11b, Runx3, Sox4*, and *ltk*. All DEGs observed in Eμ-PRMT5/TCL1 at 6 months compared to all other groups were

annotated with gene set enrichment analysis (GSEA), identifying convergence on pathways highlighted by MYC and E2F targets, cell cycle checkpoints, and cell cycle control (Supplemental Fig. 3G).

A concordance between ATAC-seq and RNA-seq data, areas of open chromatin were significantly associated with increased gene expression (Fig. 5c). Consistent with the literature on the role of

**Fig. 3 | PRMT5 promotes a distinct gene expression signature in murine CLL-like cells resembling human RT. a** ScRNA-seq analysis of spleen cells in Eμ-PRMT5 ($n = 5$) and Eμ-TCL1 ($n = 4$) mice, visualized via UMAP and clustered according to K-means ($n = 10$). B cell (Cd19+, Ms4a1+, Cd79a+), T cell (Cd3+, Cd4+, Cd8+), Mono: monocyte (Cd14+, Itgam), and neutrophil (Cd177, Itgam) clusters were assigned as indicated, Supplemental Fig. 2E. Cell density is calculated as the 2d kernel density estimate mapped to color scale. **b** "CLL-like" cells with co-expression of Cd19 and Cd5 (UMI counts >0) identified in the spleen of both Eμ-PRMT5 and Eμ-TCL1 (mice from **a**), localized to specific cell clusters. **c** Relative gene expression of B-cell maturation and leukemogenic markers in the spleen cells of Eμ-PRMT5 and Eμ-TCL1 mice (from **a**). Gene expression is shown as the Log10-transformed expression where gene expression is size factor normalized UMI counts. **d** K-means clusters stratified by genotype. Clusters are distinguished by color and number. **e** Heatmap highlighting the top 50 cluster-specific genes among B-cell clusters enriched in Eμ-PRMT5 (3.4, 3.6) and Eμ-TCL1 (3.4) mice (from **a**), showing expression relative to additional B-cell clusters 3.2 and 3.8. Gene expression across 150 genes is shown as the average Log10-transformed expression in splenic B-cell clusters. Genes associated with leukemia and lymphoma are highlighted. Source data are provided as a Source Data file.

PRMT5 in splicing regulation[19], rMATS analysis revealed an increase in the number of aberrant splicing events enriched in cassette exons with higher rates of exon inclusion (Fig. 5d) in Eμ-PRMT5/TCL1 cells at 3 and 6 months compared to Eμ-TCL1.

### PRMT5 activates oncogenic signaling pathways in the spleen and lymph nodes of Eμ-PRMT5/TCL1 mice

We conducted bulk RNA-seq in leukemic B cells from the spleens of diseased Eμ-PRMT5/TCL1 and Eμ-TCL1 mice at ERC having a CLL-like phenotype (Supplemental Fig. 4A). BCR gene rearrangement analysis revealed minimal diversification in heavy and kappa chain gene usage in mice from both groups, confirming a clonal expansion in both Eμ-PRMT5/TCL1 and Eμ-TCL1 models (Supplemental Fig. 4B, C). A low mutational burden in the *IgHV* region was observed across all groups, consistent with *unmutated IgHV* (Supplemental Fig. 4D).

Splenocytes from Eμ-PRMT5/TCL1 and Eμ-TCL1 mice at ERC were also analyzed by 3' scRNA-seq. Gene expression profiles of 16,004 Eμ-PRMT5/TCL1 cells and 25,886 Eμ-TCL1 cells revealed graph-based clustering of B-, T-lymphocytes, monocytes, and neutrophil cell populations with distinct transcriptional activity (Fig. 5e and Supplemental Fig. 4E), with Cd19+ /Cd5+ "CLL-like" cells demonstrating definite clustering patterns in regions with the highest cell density within each model (Supplemental Fig. 4F). In general, B- and T cells from Eμ-PRMT5/TCL1 and Eμ-TCL1 spleens shared similar gene expression patterns, however, B cells from Eμ-PRMT5/TCL1 mice displayed an enrichment for oncogenic and immune-modulatory gene transcripts such as *Myc*, *Mki67*, *Egr1*, *Cxcr5*, *Ccr7*, *Il-10*, *Ctla4*, and *Pd-L1 (Cd274)* (Fig. 5g, h). As observed in B cells from Eμ-PRMT5 animals, *Cd93* expression was absent in the spleen of Eμ-PRMT5/TCL1 animals, suggesting a Eμ-TCL1-specific role for this gene (Supplemental Fig. 4F). Six transcriptionally distinct clusters within the B-cell populations were identified; of these, cluster 5.6 was nearly exclusively present in Eμ-PRMT5/TCL1 mice while all the other clusters were represented in varying ratios in both models. Pathway analysis of the top enriched genes in cluster 5.6 revealed activation of EIF2, HIF1α, mTOR and Rho family GTPase signaling pathways in addition to B-cell development and actin cytoskeleton signaling pathways (Supplemental Fig. 4G).

Alternatively, clusters 5.1, 5.3, and 5.8 were present in similar ratios and population density in both Eμ-PRMT5/TCL1 and Eμ-TCL1 spleen tissue, displaying a gene signature distinct from that in cluster 5.6 which was only observed in Eμ-PRMT5/TCL1 mice. Clusters 5.1, 5.3, and 5.8 were observed with an enrichment in genes including *S100a6/10/11* encoding calcium-binding components.

The transcriptional profile in tumor cells derived from lymph nodes of mice at ERC was also assessed via 3' scRNA-seq. Gene expression analysis of 20,362 live cells from Eμ-PRMT5/TCL1 and 12,898 live cells from Eμ-TCL1 sub-mandibular lymph nodes again revealed graph-based clustering of B-, T-lymphocytes, monocytes, and neutrophil cell populations with distinct transcriptional activity (Supplemental Fig. 5A, B). Similar to that observed in spleen cells from both Eμ-PRMT5/TCL1 and Eμ-TCL1 models, Cd19+ /Cd5+ "CLL-like" cells were identified clustering in regions with high population density within the B-cell compartment in each model (Supplemental Fig. 5C). B-cell clusters from Eμ-PRMT5/TCL1 lymph nodes displayed

enrichment of oncogenic genes, including *Ly6a* (*Sca-1*)−a marker for Myc-driven cancer stem cells in human and mouse lymphoma models[38], *Myc*, and *Il-10* as well as increased expression of markers indicating activated cell populations, including *Cd274* (*Pd-L1*), *Mki67*, and *Ccr7* (Supplemental Fig. 5D).

Four transcriptionally distinct clusters amongst B cells of the lymph node compartment were identified (clusters LN.2, LN.4, LN.5, and LN.7). Of these, cluster LN.2 was predominantly present in Eμ-PRMT5/TCL1 mice while cluster LN.4 was almost exclusively present in Eμ-TCL1 mice, although these two clusters have very similar gene expression profiles (Supplemental Fig. 5E). Cluster LN.5 was equally represented in both models, and cluster LN.7, which contains over-expressed genes such as *Mki67*, *Myc*, *Birc5*, *Dek*, *Top2A*, and *Ccr7*, was more abundantly represented in the Eμ-PRMT5/TCL1 animals.

In relation to other lymph node B-cell clusters, cluster LN.7 was found to feature reduced abundance in genes including *S100a6/10/11*, also found to be reduced in B-cell cluster 5.6 (Fig. 5g) in the spleen of Eμ-PRMT5/TCL1 mice (Supplemental Fig. 5E). Alternatively, cluster LN.7 featured reduction of genes including *Aldh2*, *Cd37*, *Ms4a1*, *Napsa*, and *Psap*−all genes overrepresented in Eμ-PRMT5/TCL1 spleen cells cluster 5.6. Top upregulated genes in cluster LN.7 and LN.2 converged on cell cycle control signaling, EIF2 and mTOR signaling, B-cell development, oxidative phosphorylation, and NFAT-related immune response (Supplemental Fig. 5F).

Overall, a high degree of overlap in the most enriched genes from Eμ-PRMT5/TCL1-dominant lymph node cluster LN.7, Eμ-PRMT5/TCL1-dominant spleen cluster 5.6, and Eμ-PRMT5-dominant spleen clusters 3.4 and 3.6, was observed with cells from human RT lymph node tumors with abundant *PRMT5* expression (RT clusters 3 and 11). This included leukemogenic genes including *Myc*, *Cd69*, *Cd83*, AP-1 components including *Fos*, *Jun*, and *Junb*, tubulin-encoding genes *Tuba1b* and *Tubb5*, genes encouraging lymphoid homing including *Ccr7* and *Cxcr5*, and mediators of immune suppression including *Il-10*, *Ctla4*, and *Cd274* (*Pd-L1*), cumulatively providing supporting evidence for a role for PRMT5 in the advanced progression of CLL.

### Identification of a selective inhibitor of PRMT5

The emerging role for PRMT5 activity in both solid and hematologic tumors has spurred the development of several drug candidates with uncompetitive (i.e., compound EPZ015666[39]) and competitive (i.e., compounds JNJ-64619178[40], LLY-283[41], PF-06855800[42]) inhibition of SAM-mediated PRMT5 enzymatic activity. However, these compounds have been reported with limited specificity for PRMT5 among other methyltransferases, restricting the potential for clinical translatability as off-target effects disrupting processes essential in normal hematopoiesis are of significant concern.

Thus, with the intent to therapeutically target PRMT5 in the context of CLL at risk of RT, we developed tool compound PRT382 (Fig. 6a), a class of potent, selective PRMT5 inhibitors. Structural similarity between PRT382 and adenosine-backbone inhibitors JNJ-64619178, LLY-283, and PF-06855800 suggests a comparable SAM-competitive mechanism, however, PRT382 is distinguishable by a 3,4-dichlorophenyl at the 5' position of the ribose and a methoxyimine

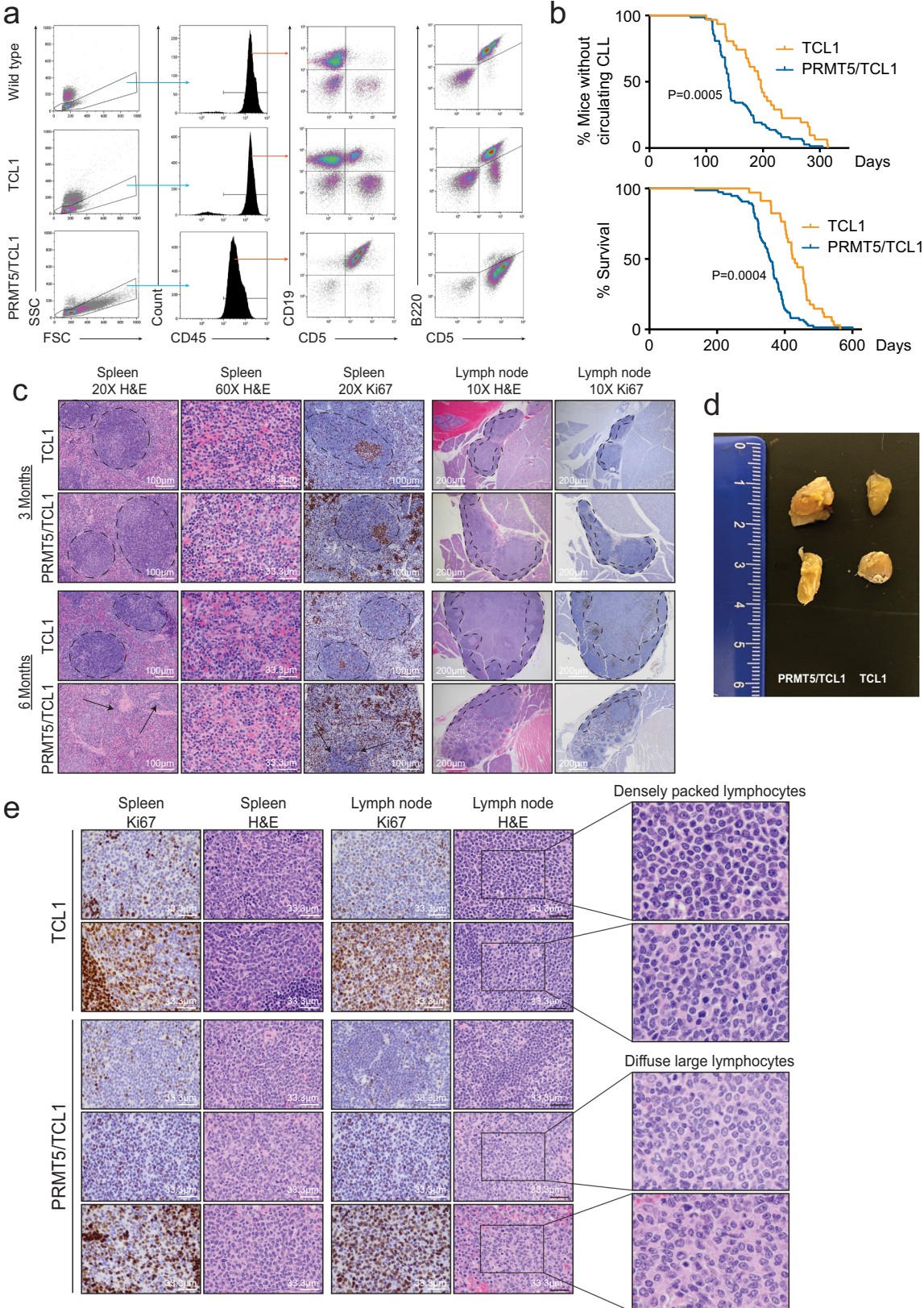

at the 4' position of the pyrrolo pyrimidin moiety. PRT382 is selective for PRMT5 alone, demonstrating an $IC_{50} > 1\,\mu M$ against 38 other methyltransferases (e.g DNMTs, MLL1-4 complexes, EZH1 and EZH2 complexes, NRMTs, NSD, SUVs, SMYs, SETs, PRDM9), and other members of the PRMT family (Fig. 6b). PRT382 demonstrated optimal enzymatic kinetics in vitro, producing an $IC_{50}$ of 2.8 nM in filtration binding assays with recombinant human PRMT5/MEP50 and histone H2A as the protein substrate (Fig. 6c), and an $IC_{50}$ of 27 nM in a reducing sDMA assay in the Granta-519 lymphoma cell line (Fig. 6d). Anti-proliferative efficacy was demonstrated in the leukemia and lymphoma cell lines CCMCL-1, JVM-2, U-2932, RL, and KARPAS-422 with $IC_{50}$ values ranging between 0.01 and $0.5\,\mu M$ (Fig. 6e, f).

**Fig. 4 | Overexpression of PRMT5 contributes to the progression of CLL in vivo and results in an aggressive phenotype. a** Eμ-PRMT5/TCL1 (n = 78) and Eμ-TCL1 (n = 36) mice were followed monthly for spontaneous disease expansion by flow cytometry of peripheral blood. Representative flow cytometry plots are shown in 8-month-old mice. CLL-like disease development was determined by the expansion of Cd19+/Cd5+/B220dim cell populations. Gating strategy to visualize single Cd45+ cells as indicated. **b** Eμ-PRMT5/TCL1 and Eμ-TCL1 mice (from **a**) were followed monthly and assessed for disease development and survival. CLL-like disease onset (>20% Cd19+ /Cd5+ cells in peripheral blood; top panel) and survival (bottom panel) comparisons were visualized via Kaplan–Meier plot, and statistical analysis was completed using the log-rank (Mantel–Cox) test. Median time to disease onset: 140 days, Eμ-PRMT5/TCL1; 196 days Eμ-TCL1. Median survival: 356 days, Eμ-PRMT5/TCL1; 426 days Eμ-TCL1. Disease onset: P = 0.0005, hazard ratio = 0.5, 95% CI (0.34−0.74); survival: P = 0.0004, hazard ratio = 0.13, 95% CI (0.07−0.23). Source data are provided as a Source Data file. **c** Representative histopathology analysis via H&E and Ki67 staining analysis of spleen and lymph node tissues from Eμ-TCL1 and Eμ-PRMT5/TCL1 mice at 3 and 6 months of age (Eμ-TCL1: n = 3, n = 2, respectively; Eμ-PRMT5/TCL1: n = 4, n = 4, respectively). Dashed lines encircle proliferative white pulp areas in the spleen, which are absent in 6-month Eμ-PRMT5/TCL1 mice. Splenic lymphoid tissue with marked effacement of normal lymphoid architecture is indicated with arrows. Ki67 staining of splenic and lymph node tissue highlights the germinal centers. Proliferative lymph node germinal centers are also highlighted by dashed lines. **d** Gross histology examination of cervical lymph nodes collected from Eμ-PRMT5/TCL1 (left) and Eμ-TCL1 (right) mice (n = 2 each). Enlarged lymph nodes were frequently observed in Eμ-PRMT5/TCL1 animals. **e** Representative histopathology analysis via H&E and Ki67 staining of spleen and lymph node tissues from Eμ-PRMT5/TCL1 (n = 4) and Eμ-TCL1 (n = 2) mice euthanized upon the development of a disease phenotype meeting predefined removal criteria. Scale bars as indicated. Cut out of lymph node H&E images demonstrate transformation to a diffuse large cell phenotype in Eμ-PRMT5/TCL1 mice compared to densely packed lymphocytes in Eμ-TCL1 mice.

In vivo, PRT382 demonstrated low clearance and high oral bioavailability in mice, rats, and dogs (F% = 70–80%), and high AUC levels (1175 h*kg*ng/mL/mg) in mice. Delivery of PRT382 at 10, 30 and 100 mg/kg was well-tolerated in WT CD-1 mice, leading to peak plasma levels of 12, 32, and 75 μM, respectively, without observable toxicity after 5 daily oral doses (Fig. 6g).

## PRMT5 blockade displays antileukemic properties in aggressive CLL

Cas9-mediated doxycycline-inducible knockout of PRMT5 blocked cell proliferation in the CLL cell line Mec-1 (Fig. 7a, b). Further, in CLL cell lines HG3 and Mec-1, 72 h PRT382 treatment resulted in a decrease in global symmetric dimethyl arginine (SDMA) residues at doses as low as 100 nM, indicating potential efficacy for PRT382 against CLL cells (Fig. 7c). Consistent with the reduction in global SDMA, PRT382 treatment induced time and dose-dependent growth inhibition beginning at 100 nM in the HG3 and Mec-1 cell line (Fig. 7d). No changes in growth potential were observed using the SAM non-competitive PRMT5 inhibitor EPZ015666[39], which coincided with only a modest decrease in global symmetric dimethyl arginine residues at higher concentrations (Supplemental Fig. 6A). PRT382 demonstrated similar-to-reduced anti-proliferative potency in HG3 and Mec-1 cells when compared to SAM-competitive PRMT5 inhibitor compounds JNJ-64619178, LLY-283, and PF-06855800, although this result was expected due to the wider selectivity profile reported for these compouds[40–42]. In addition, in primary CLL cells (n = 12 IGHV-U patients) with and without TLR9 stimulation using synthetic CpG oligodeoxynucleotide, 72 h PRT382 treatment resulted in a dose-dependent reduction in proliferative potential in vitro (Supplemental Fig. 6B).

To evaluate the in vivo anti-tumor effect of PRT382 we first confirmed that the Eμ-PRMT5/TCL1 bulk tumor population was amenable to adoptive transfer into immunocompetent C57BL/6J syngeneic mice. This established a circulating malignancy at a median time of 21 days after injection, accumulating a B220lowCd19+Cd5+ CLL-like population in the blood of all recipients (n = 7 mice; Supplemental Fig. 6C). Left untreated, engrafted mice succumbed to expanding leukemia/lymphoma with a median overall survival of 67 days (range 66–75 days, Supplemental Fig. 6D), and tumor histology was similar to that of transgenic Eμ-PRMT5/TCL1 mice with spleen and lymph node involvement and bone marrow infiltration (Supplemental Fig. 6E).

A cohort of similarly engrafted mice were followed and randomly enrolled into treatment groups (10 mg/kg PRT382, 50 mg/kg EPZ015666, vehicle control; Fig. 7e) upon evidence of circulating disease (>10% Cd19+Cd5+ cells) to evaluate PRT382 efficacy in vivo and to further compare efficacy against the non-competitive PRMT5 inhibitor EPZ015666. PRT382 treatment reduced the rate of peripheral disease early in disease progression (weeks 2-7 post engraftment) compared to mice treated with EPZ015666 or vehicle (Fig. 7f). PRT382-induced delay in leukemic progression led to a significant extension in survival compared to mice treated with EPZ015666 or vehicle (Fig. 7g). Spleens from PRT382-treated mice had significantly reduced weight and volume compared to those treated with either EPZ015666 or vehicle (Fig. 7h and Supplemental Fig. 6F). Immunoblot analysis of splenic B cells from mice within each treatment group confirmed PRMT5 target modulation in all PRT382 but not EPZ015666-treated mice (Fig. 7i).

Histopathologic assessment revealed normal lymph node architecture was effaced by sheets of large neoplastic lymphocytes with occasional mitotic figures, and expansion of splenic white pulp by neoplastic lymphocytes was apparent in all treatment groups (Fig. 7j). PRT382-treated mice displayed a decrease in leukemic expansion across non-lymphoid tissues compared with vehicle or EPZ015666-treated mice, notably demonstrated by reduced numbers of sinusoidal (arrows) and portal/periportal (P) lymphocytes accumulating in the liver. There was no observable treatment-related toxicity, bone marrow depletion, or GI effects in any treatment group. Taken together, our findings suggest PRMT5 is a promising therapeutic target in CLL/RT and warrants clinical investigation in patients with aggressive CLL and RT.

## Discussion

A number of recurrent clonal and subclonal genetic mutations of potential leukemogenic relevance (e.g., TP53, Notch, SF3B1, XPO1) have been identified, however, these mutations do not entirely explain the molecular pathogenesis of CLL or transformation to RT. Animal models mimicking large B-cell lymphomas that arise from CLL are completely missing, posing a major hurdle in RT research. Importantly, given the poor survival seen in patients with RT, a model for testing therapeutics in the setting of both CLL and aggressive lymphoma is of high interest.

PRMT5 regulates oncogenes such as NOTCH1, c-MYC, P53, and MLL–AF9[43] that are often dysregulated in RT; thus, we hypothesized PRMT5 may be a unifying driver contributing to CLL progression to RT. We showed that while PRMT5 is minimally expressed in most CLL cases, both in circulating and lymph node-based cells, RT tumors express abundant PRMT5. Circulating CLL cells from patients who eventually develop RT expressed abundant PRMT5 months to one year prior to RT diagnosis. Transcriptomic analysis in RT LN tissue with elevated PRMT5 revealed enrichment for genes including BIRC5, CDK1, TCL1A, FOXM1, TOP2A, HMGB1/2, UBE2C/S, and TUBA1A/B/C. Upregulation of these factors with PRMT5 may be necessary to overcome the negative cell-cycle regulatory signals from CDKN2A likely stemming from enhanced BCR stimulation[44]. These suggest PRMT5 overexpression may be an early event predisposing CLL patients to ultimately progress and transform even in the absence of identifiable genetic lesions.

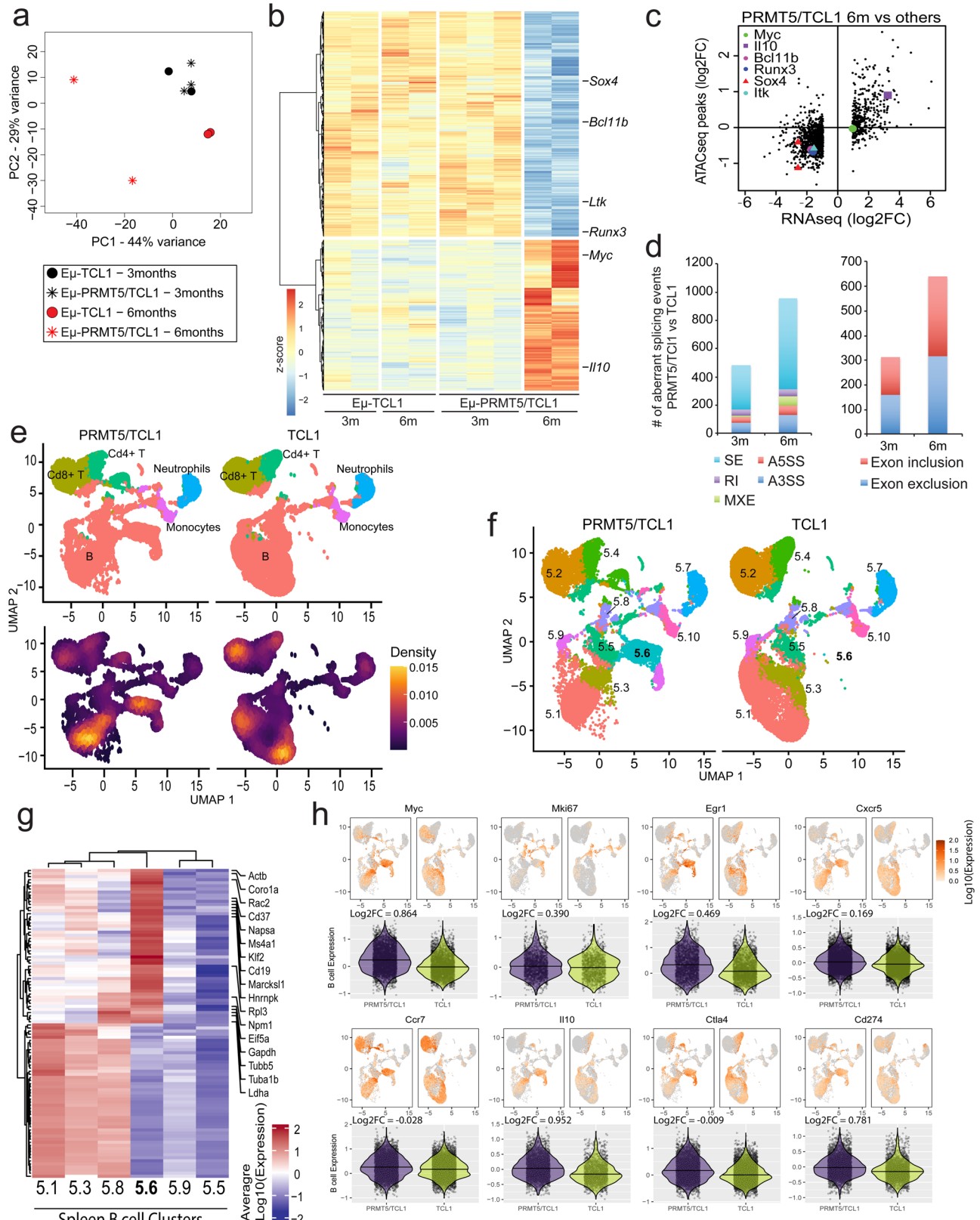

While ablation of PRMT5 has been shown to prevent MYC and NOTCH1-driven lymphomagenesis[21], the in vivo effects of B-cell-specific PRMT5 upregulation are unknown. We show here that mice overexpressing human PRMT5 in immature and mature B cells develop a pre-neoplastic expansion of Cd5+ B-cell clones, with elder mice eventually developing CLL. Survival of Eµ-PRMT5 mice was significantly reduced compared to WT littermates, predominantly due to an abnormal expansion of clonal lymphoid populations invading the lymph nodes, lung, and spleen. The onset of a clonally related B-cell-lymphoproliferative disorder with a CLL-like gene expression profile supports a leukemogenic driving role for PRMT5.

**Fig. 5 | Leukemogenic signature in Eμ-PRMT5/TCL1 splenic B cells differs from Eμ-TCL1 mice and resembles RT tumors. a** Bulk RNA-sequencing performed on leukemic cells from the spleen of Eμ-TCL1 mice at 3 months (n = 2) and 6 months (n = 2) of age compared with Eμ-PRMT5/TCL1 mice at 3 months (n = 3) and 6 months (n = 2) of age. Variance is visualized via a principal component analysis plot. **b** DEG analysis from bulk RNA-sequencing in cells from (**a**) displayed via heatmap. **c** Comparative gene expression analysis of spleen cells from Eμ-PRMT5/TCL1 mice at 6 months of age compared to all other groups (from **a**) by log2FC in RNA-seq (x axis) and log2FC in ATAC-seq (y axis). Points represent gene-peak pairs. The plot represents genes exceeding a cutoff of log2 fold change in RNA-seq > |1|. **d** Differential alternative splicing events between Eμ-PRMT5/TCL1 and Eμ-TCL1 cells (from **a**) at 3 and 6 months. SE skipped exon, RI retained intron, MXE mutually exclusive exons, A5SS alternative 5' splice site, A3SS alternative 3' splice site. **e** ScRNA-seq analysis of spleen cells in Eμ-PRMT5/TCL1 (n = 4) and Eμ-TCL1 (n = 4) mice, visualized via UMAP and clustered according to K-means (n = 10). B cell

(Cd19+, Ms4a1+, Cd79a+), T cell (Cd3+, Cd4+, Cd8+), Mono: monocyte (Cd14+, Itgam), and neutrophil (Cd177, Itgam) clusters were assigned as indicated, Supplemental Fig. 4E. Cell density is calculated as the 2d kernel density estimate mapped to color scale. **f** K-means clusters stratified by genotype. Clusters are distinguished by color and number. **g** Heatmap highlighting the top 50 enriched genes in clusters 5.6 and 5.1 in mice (from **e**), visualizing expression relative to other B-cell clusters. Gene expression across 100 genes is shown as the average Log10-transformed expression in splenic B-cell clusters. Genes associated with leukemia and lymphoma are highlighted. Source data are provided as a Source Data file. **h** Relative gene expression of CLL and RT-related genes in Eμ-PRMT5/TCL1 and Eμ-TCL1 spleen cells (from **e**). Violin plots demonstrate the relative distribution of expression in B cells with non-zero expression stratified by genotype and shown as the Log10-transformed expression. Log2-transformed fold changes were calculated from pseudobulk differential expression analysis comparing B-cell expression between models. Points represent individual cells.

While the overall transcriptome between Eμ-PRMT5 and Eμ-TCL1 tumors were broadly similar, the expansion of proliferating clones characterized by activation of oncogenic pathways including *Myc*, *ATF4*, *Il-4*, *Ccr7*, *Cd69*, *Cxcr5*, *Npm1*, *Birc5*, and AP-1 transcription factor complex member *Jun-b* were observed in Eμ-PRMT5 mice. Our data also suggest it is the accumulation of these transcriptional changes within discrete clonal populations that drive disease initiation and progression. Manipulation of MYC, NPM1, and BIRC5 present well-defined roles in leukemogenic transformation, and CD69 expression on CLL cells has been discussed as a critical marker for predicting CLL prognosis[45]. MYC activation has also been observed to upregulate *ATF4*, where ATF4 plays a critical role in supporting cell adaptation and survival during MYC-dependent tumor growth[46], suggesting an additional cooperative axis between PRMT5, MYC, and ATF4 in RT. Further, the chemokine receptor Ccr7 plays a pivotal role in directing tumor cells to lymphoma-supporting niches in the Eμ-Myc mouse model[47], suggesting PRMT5 hyperactivity may also encourage honing and accumulation in these secondary lymphoid organs. Moreover, significantly enriched genes in distinct Eμ-PRMT5 populations overlapped with *PRMT5*-expressing cells from human RT tumors, establishing a PRMT5-driven gene signature in aggressive mouse and leukemia.

A CLL-to-RT phenotype is rarely observed in the Eμ-TCL1 mouse model, even in the presence of conditional B-cell-specific Trp53 deficiency or with simultaneous co-expression of *Myc*[35]. Given *PRMT5* is consistently overexpressed in RT tissues and in CLL eventually undergoing RT, we developed the Eμ-PRMT5/TCL1 model to evaluate a role for PRMT5 in CLL progression to RT. Concurrent *PRMT5* and *TCL1* expression resulted in a more rapid disease onset with increased lymph node involvement and overt lymphomagenesis that appeared to recapitulate disease behavior seen in human RT, representing a model to mimic large B-cell lymphomas that arise from CLL. Cells from the Eμ-PRMT5/TCL1 model again featured a gene expression signature that is concordant with RT tumors and the Eμ-PRMT5 model, where dysregulated PRMT5 and upregulation of genes including *Ctla4*, *Cd274* (*Pd-L1*), and *Il-10* suggest aberrant activation of PRMT5 in B lymphocytes predisposed to CLL may further manipulate the surrounding environment to facilitate tumorigenic outgrowth in secondary lymphoid niches. Importantly, DEG analysis in Eμ-PRMT5/TCL1 mice at 6 months of age revealed dysregulation of genes including *Myc*, *Il-10*, *Sox4*, *Ltk*, *Bcl11b*, and *Runx3* compared to Eμ-TCL1 mice at this age and both Eμ-PRMT5/TCL1 and Eμ-TCL1 mice at 3 months of age. Interrogating the differential transcriptome at this early time point, while the CLL-like disease in Eμ-PRMT5/TCL1 mice is evolving through the pre-RT phase to morphologically resemble a large B-cell lymphoma, provides insight to mechanisms driving this progression. Understanding the dysregulated gene signature in the pre-RT phase, like that identified herein, can assist in the early detection of evidence indicating CLL-to-RT is likely in patients monitored in the clinic, and

further characterization of this pre-RT phase gene signature should be prioritized to better define this phenomenon.

A major hurdle in understanding the genomic complexity of Richter's Transformation and the molecular divergence from clonally related CLL is the rarity and aggressiveness of RT, resulting in limited samples available for evaluation. This is further compounded by the reality that RT tumors are primarily nodal-based and often not recommended for surgical extraction due to high risk to the patient. Thus, genomic studies evaluating CLL-to-RT evolution are often limited in regard to the number of samples analyzed. The high throughput genomic studies presented herein following longitudinal CLL-to-RT samples were similarly impacted by the lack of available tissue, limiting our analysis (Fig. 1a, d). However, by plotting the aggregate gene expression described in a recent RT genomic study[31] against the scRNA-seq from RT patients data in this study, we observed a high degree of overlap between the gene signature in PBMC and LN samples from RT patients described here with those described previously, notably varying from samples from the CLL phase in our study and in previous studies (Fig. 1e).

These limitations were also evident in recent publications evaluating gene expression from paired circulating CLL vs nodal RT tumors[48,49], reporting data from n = 12 and n = 1 longitudinal samples, respectively. Although limited, these analyses provide valuable insight into the poorly defined molecular landscape of RT and have widened our understanding of these highly aggressive tumors. Using these studies for further comparison, Klintman and colleagues[48] observed enrichment of *IL-10* in RT tumors compared to circulating CLL, similar to our evaluation in Eμ-PRMT5/TCL1 spleen and lymph node tissue (Fig. 5g and Supplemental Fig. 5D), indicating a prominent role for mediators of immune suppression in CLL-to-RT evolution. Further, among the top enriched genes in RT tumors from the Klintman study [Log2FC > 0.4, -Log10(P value)>2] were *CCNA2*, *MYC*, and *NR4A1*; each identified as genes also enriched in B-cell clusters predominantly in the spleen and lymph nodes of Eμ-PRMT5 and Eμ-PRMT5/TCL1 mice evaluated in this study (Figs. 3g and 5g and Supplemental Fig. 5E). Overlap of these genes cumulatively supports the notion that PRMT5 acts as a significant factor in evolution to RT and that RT tumors are highly active compared to classically indolent circulating CLL, where enrichment in mediators of cellular activity and cell cycle control (such as *CCNA2*, *MYC*) may be drivers of this phenotype.

Interestingly, the *NR4A1* gene encodes a member of the nuclear receptor subfamily 4 (Nur77) that behaves as a ligand-activated transcription factor and nuclear co-factor. NR4A1 is reported to be largely absent in leukemia and lymphoma yet highly expressed in a myriad of solid tumors, specifically in solid tumors with enhanced metabolic rate, with elevated *NR4A1* expression acting as a negative prognostic marker in affected patients[50]. ChIP analysis has demonstrated NR4A1 interactions bound to the GC-rich promoter sequence of *BIRC5* (*survivin*), and *Nr4a1* knockout mice developed expansions of myeloid

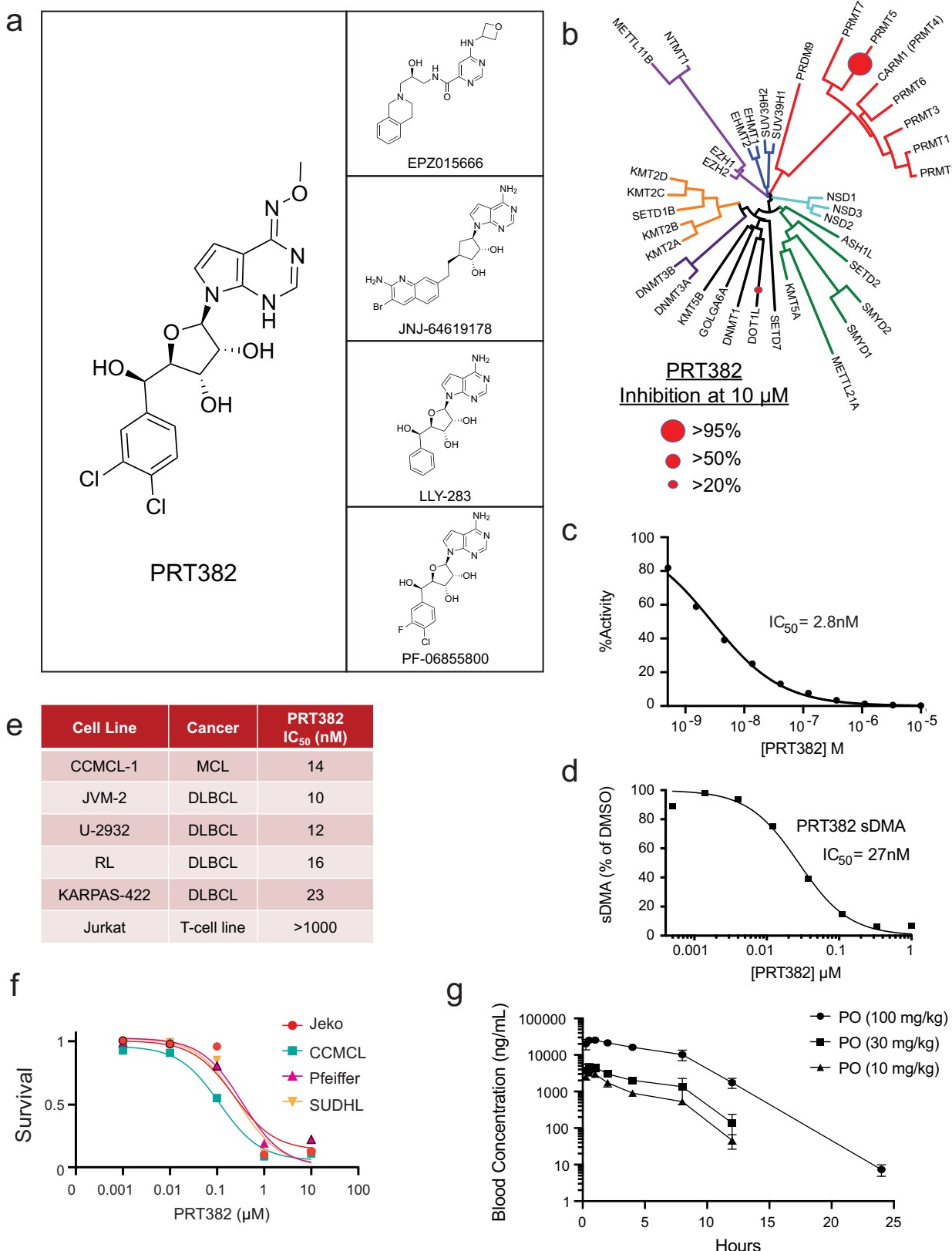

progenitors and hemopoietic stem cells with decreased expression of *c-Jun* and *Jun-B*[51,52]. In our studies, *Birc5* and *Junb* were repeatedly observed among the most enriched genes in B-cell clusters predominantly in the spleen and lymph nodes of Eµ-PRMT5 and Eµ-PRMT5/TCL1 mice, possibly implicating a PRMT5-NR4A1 axis as a major mechanism supporting the CLL-to-RT evolution. Further work to

describe this interaction will be necessary to appropriately define the oncogenic potential of this relationship. Overall, comparative evaluation of Eµ-PRMT5 and Eµ-PRMT5/TCL1 mouse tumors and human RT lymph nodes in our limited cohort, in addition to similar recently described human RT cohorts, validates the use of the Eµ-PRMT5 mouse as a promising laboratory model of Richter's Transformation

**Fig. 6 | PRT382, a selective inhibitor of PRMT5. a** Chemical structure of tool compound PRT382, an in vitro and in vivo PRMT5 inhibitor. Structural comparison is shown between recently developed PRMT5 inhibitors EZP015666 (SAM non-competitive) and JNJ-64619178, LLY-283, and PF-06855800 (SAM-competitive). **b** Selectivity of PRT382 for PRTM5 in contrast to other methyltransferases and PRMT family members. Circle size represents percent inhibition at 10 μM. **c** Filtration binding assays in the presence of recombinant human PRMT5/MEP50 and histone H2A. Plot is representative of $n = 3$ experiments. **d** Reducing sDMA assay in the Granta-519 lymphoma cell line. Plot is representative of $N = 3$ experiments. **e** Anti-proliferative activity, measured as IC$_{50}$, of PTR382 against a leukemia and lymphoma immortalized cell lines. **f** Dose-dependent proliferation decrease

with a panel of B-lymphoma cell lines ($n = 3$ independent experiments) upon PRMT5 inhibition with increasing doses of PRT382 for 72 h as measured by MTS assay. **g** PRT382 was formulated in 0.5% carboxymethylcellulose sodium salt + 0.5% Tween80 as a 1 mg/mL suspension for 10 mg/kg and was administered in a 10 mL/kg dose volume by oral gavage to CD-1 mice. Blood was collected at indicated time points and analyzed by LC-MS/MS. In vivo delivery of PRT382 at 10, 30, and 100 mg/kg was well-tolerated in WT CD-1 mice and led to peak plasma levels of 12, 32, and 75 μM, respectively ($n = 3$ biologically independent animals per group). Blood concentrations are reported as the mean +/−SD. Source data are provided as a Source Data file.

and warrants further evaluation with additional current and ongoing studies in RT.

Critically, we show that genomic ablation of *PRMT5* in CLL cells decreased cell proliferation, supporting the inhibition of PRMT5 as a promising therapeutic target in CLL. Recently, a suite of SAM-competitive PRMT5 inhibitors have been developed[40–42], however, the diverse array of targets affected by these compounds may limit their progression in the clinical setting due to unwanted cytotoxicity in healthy tissue. To address this issue, we herein report here the development of a small molecule PRMT5 inhibitor with a predicted SAM-competitive mechanism of action, PRT382, which is highly selective for PRMT5 against a wide range of other methyltransferases. PRT382 demonstrated in vitro and in vivo efficacy against CLL cell lines, isolated CLL patient cells, and the CLL-like cells from the Eμ-PRMT5/TCL1 mouse model of CLL/RT, overall providing strong evidence supporting the use of targeted PRMT5 inhibition in CLL with elevated *PRMT5* expression at high risk of RT. As expected, due to restricted selectivity, PRT382 demonstrated reduced cytotoxic potency against CLL cells compared to compounds with a similar mechanism of action. However, when compared with the SAM non-competitive PRMT5 inhibitor EPZ015666, PRT382 demonstrated superior in vitro anti-tumor activity and improved survival in the Eμ-PRMT5/TCL1 mouse model of CLL/RT in vivo.

The ability of PRMT5 inhibitors to influence multiple cellular pathways and restore the function of tumor suppressors via demethylation may be attractive as a single agent in CLL to induce durable responses and prevent or delay acquired resistance, however, care must be taken evaluating this target in the clinical setting as PRMT5 is essential in normal hematopoiesis[53]. Studies to understand mechanisms by which PRMT5 overexpression may cooperate with recurrent genomic lesions to contribute to CLL pathogenesis and progression are also necessary as the critical targets for response to PRMT5 inhibition are likely context-dependent for each cancer type[54].

Upregulation of PRMT5 is documented in lymphoma and is herein observed in CLL-to-RT progression, yet mechanisms by which this occurs remain unknown. We previously reported inhibition of bromodomain-containing protein-4 (BRD4) to down-modulate *PRMT5* along with other key genes (i.e., *cMYC, BTK*)[55] in CLL and DLBCL, suggesting BRD4 activity may control *PRMT5* gene transcription. To explore the *PRMT5* gene interactome, we employed a targeted 4C-seq approach in a CLL cell line (HG3) using bait primers specific for the *PRMT5* gene promoter region. In addition to multiple trans interactions with this site (Supplemental Fig. 7A), we identified a distinct cys genomic region ~500 kb downstream from the PRMT5 start codon on chromosome 14 featuring a large enhancer (Supplemental Fig. 7B) we have previously reported having active histone marks (i.e., H3K27Ac) and BRD4 occupancy in activated CLL cells but not normal B cells[55]. These active histone marks were reduced following pharmacologic inhibition of BRD4, along with reduced PRMT5 mRNA and protein expression in *IGHV-U* CLL cells (Supplemental Fig. 7C), suggesting *PRMT5* may be regulated by BRD4-mediated priming of this enhancer and providing an additional therapeutically targetable mechanism for RT patients and for CLL at high risk of progression to RT alone or in

combination with PRMT5i. Although suggestive, these data are very preliminary, and further studies are warranted to validate a role of BRD4 in *PRMT5* overexpression in CLL/RT, and to resolve the numerous trans interactions of unexplored significance, where irregular activity at any number of these locations may play a significant role in promoting *PRMT5* expression.

In summary, our data indicate that PRMT5 promotes CLL development, facilitates CLL progression to aggressive lymphoma through altered epigenetic programming, and provides a preclinical model and a therapeutic candidate to study CLL-to-RT progression. Upregulated PRMT5 thus presents as a biomarker for CLL patients at risk of transformation several months before a definitive RT diagnosis can be made. As patient mortality rates post-RT diagnosis are reported among the most rapid of all cancer types[5], prospective screening for diagnostic markers such as PRMT5 may be pivotal for improving the bleak outcomes associated with this disease.

## Methods

### Patient sample, cell lines, and cell culture
Our research complies with all relevant ethical regulations. De-identified human CLL cells were isolated as previously described after obtaining informed consent on protocols approved by The Ohio State University Cancer Institutional Review Board in accordance with the Declaration of Helsinki. All patients examined had CLL as defined by the 2008 IWCLL criteria[56]. Sex and age of the patients who provided samples was not available. Human peripheral blood mononuclear cells (PBMC) were isolated from CLL patients using Ficoll density gradient centrifugation[57]. (Ficoll-Paque Plus; Amersham Biosciences Inc, Piscataway, NJ). Enriched B-cell fractions were prepared using the RosetteSep™ B-cell kit (STEMCELL Technologies, Vancouver, Canada) according to the manufacturer's instructions. Isolated cells were incubated in RPMI 1640 media (Invitrogen, Grand Island, NY) supplemented with 10% heat-inactivated fetal bovine serum (Sigma-Aldrich, St. Louis, MO), 2 mM L-glutamine (Invitrogen), and 100 U/mL penicillin/100 μg/mL streptomycin (Sigma-Aldrich) at 37 °C in an atmosphere of 5% CO$_2$. The purity of enriched populations of CLL was routinely checked using CD19-PE (BD Pharmingen, Billerica, MA) staining by flow cytometry.

MEC-1 and HG3 lines were obtained from DSMZ (Braunschweig, Germany). MEC-1 cell identity was reconfirmed via fluorescence in situ hybridization (FISH) using CLL probe panel, i.e., CEP12/13q14/13q34, ATM/p53, BCL6, MYC, IGH/CCND1 (Abbott Molecular, Des Plaines, IL), and 6q21 (Kreatech, Buffalo Grove, IL) (results for 200 analyzed cells were reported). Lymphoma cell lines JVM-2, U-2932, RL, KARPAS-422, Jeko, Jurkat, Pfeiffer, SUDHL were obtained from commercial sources (DSMZ and ATCC [Manassas, Virginia, USA]). CCMCL-1 was provided by Dr. Selina Chen-Kiang from the Department of Pathology and Laboratory Medicine, Weill Cornell Medicine. All cell lines were authenticated by short tandem repeat DNA profiling. All cell lines were confirmed to be mycoplasma negative using the MycoAlert™ Mycoplasma Detection Kit from Lonza (Rockland, ME) according to manufacturer instructions and used within 3–4 weeks from thawing. For experiments, cells were resuscitated, cultured in RPMI (Invitrogen,

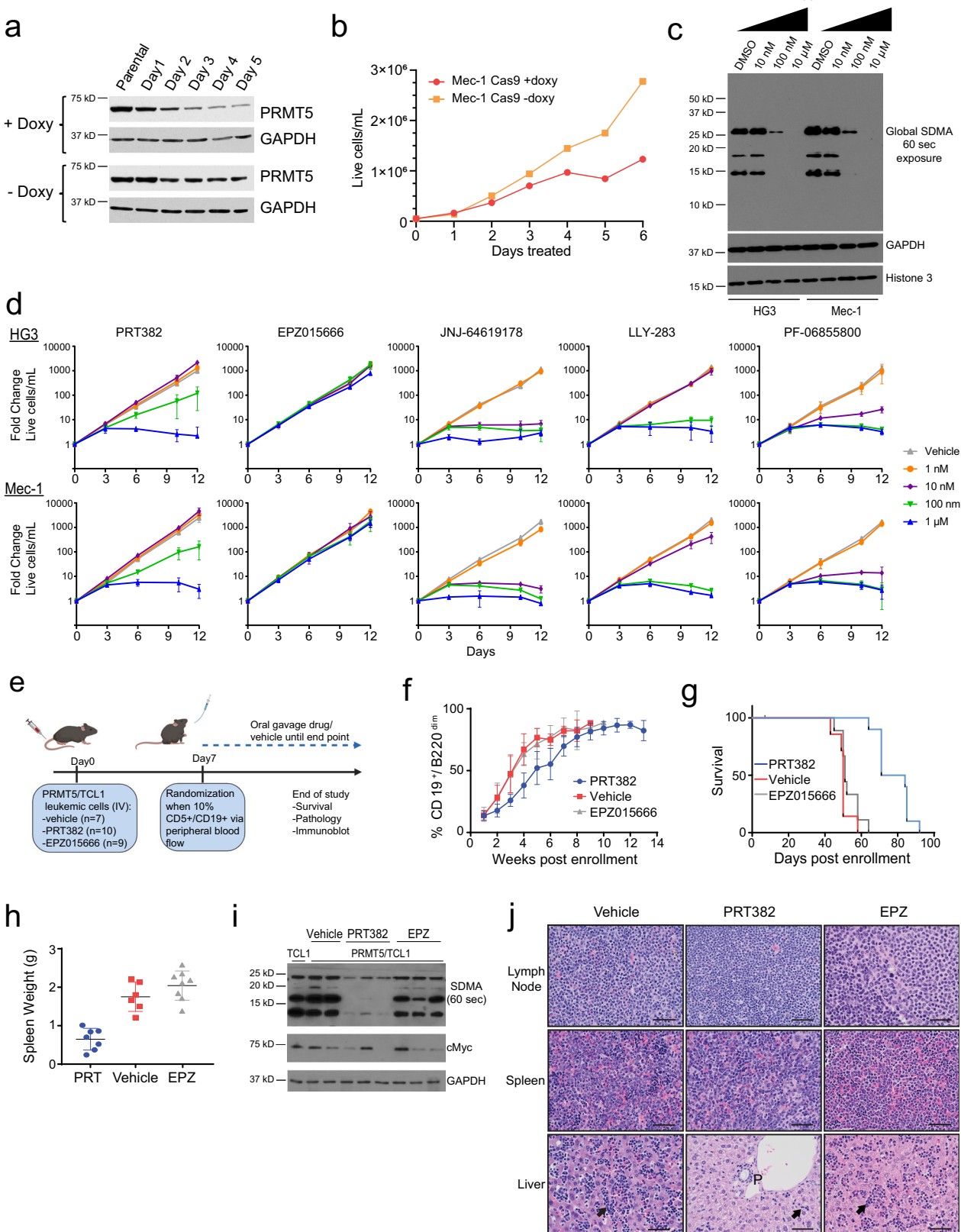

Grand Island, NY) supplemented with 10% fetal bovine serum (FBS), and used within 3-4 weeks from thawing.

Proteins extracted from whole-cell lysates were resolved by SDS-PAGE and transferred on nitrocellulose membrane as previously described[55]. CpG685 oligonucleotide was obtained from the OSU Cytogenetics Core Lab. Antibodies used for immunoblots included anti-human PRMT5 (Abcam #ab109451), GAPDH (Millipore Sigma #MAB374), SDMA (Cell Signaling Technology #13222), c-Myc (Cell Signaling Technology #5605S), and β-Actin (Cell Signaling Technology #4967S). Cas9-mediated doxycycline-inducible knockout of PRMT5 utilized a targeted sgRNA (guide sequence: AGTAGGTCTGATCGTGT CTG, PAM: GGG).

**Fig. 7 | Selective targeting of PRMT5 in vitro and in vivo with PRT382 displays antileukemic activity in an aggressive model of CLL/RT. a** Cas9-mediated dox-ycycline-inducible knockout of PRMT5 in Mec-1 cell line confirmed via western blot ($n = 3$). **b** Knockout of PRMT5 in Mec-1 cells (from **a**) blocked cell proliferation. Plot represents the mean of $n = 2$ independent experiments. **c** Global symmetric dimethyl arginine (SDMA) residues via western blot ($n = 3$) as a marker of PRMT5 activity in HG3 and Mec-1 cell lines treated with PRT382 for 72 h. **d** Dose-dependent proliferative potential as measured by cell growth in CLL cell lines upon treatment with PRT382 (n = 3), EPZ015666 (n = 4), JNJ-64619178 (n = 2), LLY-283 (n = 2), and PF-06855800 (n = 2). Cell growth is plotted as fold change relative to vehicle-treated cells ±SD. Fresh culture media and PRMT5 inhibitor at indicated concentrations were supplied every 3 days for continuous exposure over 12 days. **e** Viable $Cd19^+Cd5^+$ cells ($5 \times 10^6$) Eμ-PRMT5/TCL1 spleen cells were engrafted by tail vein into immunocompetent C57/BL6J mice. Engrafted mice were randomly assigned to treatment conditions: PRT382 10 mg/kg ($n = 10$), EPZ015666 50 mg/kg ($n = 9$), or vehicle ($n = 7$) at 1-week post engraftment. All treatments were administered for 4 contiguous days per week. Illustration created with BioRender.com. **f** Engrafted

mice weekly peripheral blood flow cytometry for the percentage of CLL-like ($Cd5^+Cd19^+B220^{dim}$) cells ±SD. PRT382 10 mg/kg ($n = 10$), EPZ015666 50 mg/kg ($n = 9$), or vehicle (0.5% methylcellulose, 0.1% Tween; $n = 7$). **g** Kaplan–Meier plot of engrafted mice survival post-enrollment. Median survival: PRT382, 77 days; EPZ015666, 51 days; vehicle, 50 days. Mantel–Cox test [PRT382 vs vehicle $P < 0.0001$, hazard ratio = 0.21 95% CI (0.051–0.87); PRT382 vs EPZ015666 $P < 0.0001$, hazard ratio=0.65 95% CI (0.26–1.6)]. **h** Average splenic weight at ERC ± SD. PRT382 vs vehicle $P = 0.0013$; PRT382 vs EPZ015666 $P = 0.0004$, paired two-tailed $t$ test. PRT382 ($n = 7$), EPZ015666 ($n = 6$), vehicle ($n = 6$). **i** PRMT5 activity evaluated by immunoblot analysis of splenic B cells from engrafted mice. Vehicle ($n = 2$), PRT382 ($n = 3$), or EPZ015666 ($n = 3$). Cells from Eμ-TCL1 ($n = 1$) shown as comparison. **j** Representative H&E histopathology evaluation at ERC in lymph node, spleen, and liver tissues of engrafted mice (from **e**) treated with vehicle ($n = 6$), PRT382 ($n = 7$), or EPZ015666 ($n = 8$). Sinusoidal lymphocytes are high-lighted with arrows. The scale bar is 33.3 μm. Source data are provided as a Source Data file.

## Flow cytometry

All gates were set using fluorescent antibody controls prepared for each individual experiment using strategies following published data and technical resource publications[58,59]. CLL populations in mouse peripheral blood were determined as $Cd45^+/Cd5^+/Cd19^+$ and $Cd45^/Cd19^+/B220^{dim}$ populations. Immunophenotyping of tumor cells in peripheral blood and spleen of Eμ-PRMT5, Eμ-PRMT5/TCL1, and Eμ-TCL1 mice by flow cytometry was performed as follows: APC rat anti-mouse Cd45 (1/50 dilution; BD Biosciences, Cat #559864), FITC rat anti-mouse Cd45R/B220 (1/25 dilution; BD Biosciences, Cat #553088), BV421 rat anti-mouse Cd19 (1/25 dilution; BD Biosciences, Cat #562701), PE rat anti-mouse Cd5 (1/50 dilution; BD Biosciences, Cat #553023), BV510 hamster anti-mouse Cd3e (1/50 dilution; BD Biosciences, Cat #563024), BV650 rat anti-mouse Cd11b (1/200 dilution; BD Biosciences, Cat #653402), BB515 rat anti-mouse Cd19 (1/200 dilution; BD Biosciences, Cat #564509), BUV737 rat anti-mouse Cd5 (1/150 dilution; BD Biosciences, Cat #612809). Immunophenotyping was conducted using a Beckman-Coulter Gallios 3-laser-10-color (B5-R3-V2) cell analyzer and BD LSRFortessa Cell Analyzer (Cat #649225). Peripheral blood from Eμ-PRMT5, Eμ-PRMT5/TCL1 and Eμ-TCL1 transgenic mice was collected monthly via check punch. Cells from the spleens of Eμ-PRMT5, Eμ-PRMT5/TCL1 and Eμ-TCL1, having met predefined euthanasia criteria, were processed and isolated as single-cell suspensions. Mouse B cells were isolated from whole spleen suspensions using EasySepTM mouse pan B-cell isolation kit (STEMCELL Technologies; Cat #19844). All flow cytometry data were analyzed with Kaluza v2.0 software (Becton Dickinson).

## Proliferation assay and flow cytometric studies

Cell proliferation was assessed via MTS (3-[4,5-dimethylthiazol-2-yl]-5-[3-carboxymethoxyphenyl]-2-[4-sulfophenyl]-2H-tetrazolium) assay (Sigma), using the tetrazolium dye 3′(4,5-dimethylthiazol-2-yl)-2,5-diphenyl-tetrazolium bromide[60]. Cell apoptosis was evaluated by Annexin V-FITC or PE and propidium iodide (PI) staining using FC500 cytometer (Beckman-Coulter, Indianapolis, IN) following the manu-facturer's procedure (BD Pharmingen, Billerica, MA)[55]. For cell cycle analysis, cells were fixed with cold ethanol overnight and then stained before analysis with 30 μg/mL propidium iodide (Invitrogen) and 200 μg/mL RNase (Sigma-Aldrich) in 0.1 % v/v Triton X-100 (Sigma-Aldrich). Cell cycle data analysis was performed using ModFit LT software (Verity Software House, Topsham, ME).

## Animal studies

Experiments were carried out under protocols approved by OSU Institutional Animal Care and Use Committee (IACUC). Eμ-PRMT5 mice: A cDNA fragment possessing the entire human *PRMT5* coding region was generated by PCR and cloned into pBluescript II KS (pEμ)

plasmid containing a mouse $V_H$ promoter (V186.2) and the $Ig_H$-μ enhancer along with the 3′ untranslated region and the poly(A) site of the human β-globin gene. The pBluescript II KS (pEμ) plasmid con-taining full length of human *PRMT5* cDNA were injected into mouse embryos at the single-cell stage and implanted into pseudopregnant B6 females. Mice were screened for the presence of the transgene by PCR on tail DNAs. Three founders were obtained (E4474 and M0806, and M0807) and bred. Eμ-PRMT5 mice were genotyped using the following primer sequences: (fwd: CGGGGAGTACACTTCCTTTC; rev: TCTCAAAGTAGCCGGCAAAG). Transgenic heterozygote mice issued from these founders were studied and compared with nontransgenic siblings raised in identical conditions. Generation of the Eμ-PRMT5/TCL1 double transgenic leukemia model was achieved by crossing Eμ-PRMT5 mice with homozygous Eμ-TCL1 mice on the C57BL/6 J background[33,61]. Disease progression was monitored by $Cd45^+/Cd5^+/Cd19^+$ flow cytometry of peripheral blood and manual assessment of peripheral blood smears, spleen size, and weight loss. Whole blood was collected via cheek punch at time of euthanasia, as well as at monthly intervals for each transgenic colony and at weekly intervals for each arm of the transplant model. Manual assessment of peripheral blood smears were confirmed by two independent pathology experts. Organs were collected at time of euthanasia and preserved in 10% formalin. Formalin-fixed tissues from mice were paraffin-embedded, sectioned onto glass slides at 4 μM, then deparaffinated, hydrated with distilled water, and stained with hematoxylin and eosin (HE), or Ki67. Mixed female and male populations were used across all experiments unless otherwise stated.

For adoptive transfer, viable $Cd19^+Cd5^+$ cells ($5 \times 10^6$) from the spleen of an Eμ-PRMT5/TCL1 transgenic mouse with active CLL-like leukemia (>80% circulating in peripheral blood) and palpable spleno-megaly were engrafted by tail vein into immunocompetent C57/BL6J female mice. Engrafted mice were randomly assigned to treatment conditions: PRT382 10 mg/kg, EPZ015666 50 mg/kg, or vehicle (0.5% methylcellulose, 0.1% tween) at 1-week post engraftment. All treatments were administered by oral gavage 4 contiguous days per week (Mon-Thurs). Predefined euthanasia criteria for mice in all transgenic colonies and murine transplant models were followed and included lethargy, impaired motility, splenomegaly, enlarged lymph nodes and/or super-ficial lymphoid tumors greater than 1.6 cm in diameter (or cumulative diameter of 1.6 cm if multiple tumors), decrease in body weight (>20%), ruffled fur, hunched back, failure to nest, and loss of appetite.

## RNA extraction, RNA-seq, and low-RNA-seq

**RNA-seq, featured in Fig. 5a, b.** Mouse B cells were collected using the EasySep™ mouse pan B-cell isolation kit (STEMCELL Technologies). Total RNA was isolated from TRIzol suspensions using a chloroform/ethanol extraction method and quantified via Qubit RNA HS Assay kit

(Invitrogen). Total RNA was quantified using the Nanodrop 2000 Spectrophotometer (ThermoFisher Scientific). Sequencing libraries were prepared using Illumina TruSeq stranded mRNA library preparation kit and sequenced on HiSeq 4000 instrument targeting ~60–80 × 10⁶ read pairs per sample. The 150 bp paired-end reads were trimmed for adapters and low-quality bases with Cutadapt and mapped to the mouse reference genome GRCm38 with STAR aligner v2.6.0[62]. The featureCounts utility[63] from the subread package[64] (v1.4.6) was used to generate gene counts based on Ensembl gene annotation release 95[65]. Pairwise comparisons were performed to identify DEGs using the DESeq2 package (v1.24.0)[66].

**Low-input RNA-seq, featured in Fig. 2i, Supplementary Figs. 2C, D, and 4B–D.** B cells were isolated via EasySep Mouse Pan-B Cell Isolation Kit (Stemcell Technologies) from spleen suspensions of diseased Eμ-PRMT5 ($n = 4$), Eμ-TCL1 ($n = 4$), and Eμ-PRMT5xTCL1 ($n = 3$) mice meeting euthanasia criteria. In total, $1 \times 10^6$ isolated B cells were washed with PBS, and RNA was extracted using the Monarch Total RNA Miniprep Kit (New England Biolabs) according to manufacturer instructions. Total RNA was quantified using the Nanodrop 2000 Spectrophotometer (ThermoFisher Scientific). The Clontech SMARTer v4 kit (Takara Bio USA, Inc.) was used for global preamplification. Illumina sequencing libraries were derived from the resultant cDNA using the Illumina Nextera XT DNA Library Prep Kit following the manufacturer's instructions. Libraries were sequenced using an Illumina HiSeq 4000 sequencer paired-end 150 bp protocol to approximately 12 million passed filter clusters per sample. Data processing was performed according to the CLEAR workflow, which identifies reliably quantifiable transcripts in low-input RNA-seq for differentially expressed gene (DEG) transcripts using gene coverage profiles. BCR reads were binned by specific heavy (H) and light-chain V genes. First, we quantified the percentage of reads with heavy, kappa, and lambda chains compared to the number of total sequencing reads. MiXCR[67] (v3.0.5) was used with default parameters to identify preprocessed reads containing CDR3 regions from B-cell heavy, kappa, and lambda chains, generating a list of CDR3 sequences associated with their relative abundances and specific V(D)J gene usage. Global DEG analysis was performed using DESeq2 (v1.20.0)[66]. The count's table containing CLEAR transcripts from all replicates was used for comparisons between groups. The resultant data were then used for gene-specific analysis for a targeted list of oncogenes.

## ATAC-seq and data processing

Accessible chromatin mapping was performed using the ATAC-seq method[68,69]. Briefly, in each experiment, $1 \times 10^5$ cells were washed once in 50 μl PBS, resuspended in 50 μl ATAC-seq lysis buffer (10 mM Tris-HCl pH 7.4, 10 mM NaCl, 3 mM MgCl₂ and 0.1% IGEPAL CA-630) and centrifuged for 10 min at 4 °C. On centrifugation, the pellet was washed briefly in 50 μl MgCl₂ buffer (10 mM Tris pH 8.0 and 5 mM MgCl₂) before incubating in the transposase reaction mix (12.5 μl 2 × TD buffer, 2 μl transposase (Illumina) and 10.5 μl nuclease-free water) for 30 min at 37 °C. After DNA purification with the MinElute kit, 1 μl of the eluted DNA was used in a qPCR reaction to estimate the optimum number of amplification cycles. Library amplification was followed by SPRI size selection to exclude fragments larger than 1200 bp. DNA concentration was measured with a Qubit fluorometer (Life Technologies). Library amplification was performed using Nextera primers. The libraries were sequenced by the Genomics Services Laboratory at Nationwide Children's Hospital using an Illumina HiSeq 4000 platform. The 150 bp paired-end reads were trimmed for adaptors and low-quality bases with Cutadapt and mapped to the mouse reference genome GRCm38 using the Bowtie2 algorithm[70] with a maximum fragment length of 1000 and no-mixed, no-discordant and very-sensitive options. After the removal of duplicate reads, deepTools2[71] were used to generate normalized genome coverage files. ATAC-seq peak locations were determined using

the MACS2 algorithm (v2.1.1.)[72] with a cutoff $P$ value of 1E-5 with "–broad–nomodel–shift -100–extsize 200" options. Consensus peak regions from all samples were generated as the union of the overlapping peaks using R GenomicRanges package[73], then annotated using R ChIPpeakAnno package[74]. Using annotatePeakInBatch function from ChIPpeakAnno package, each ATAC peak was assigned to closets/overlapping gene within 5000 bp.

## Circularized chromosome conformation capture (4C)

HG3 cells were fixed with paraformaldehyde, and chromosome segments in physical proximity were cross-linked. Chromatin was digested by DpnII and ends of interacting chromatin regions were ligated with T4 ligase. Cross-linking was removed and secondary digestion was done with Csp6I primers (CCCAACCGACTCAGGATC, TCGGCAGCA AGGATGGAG) to specifically amplify interacting partners with the bait for library construction[75]. The library was subjected to next-generation sequencing on the Illumina platform. The sequencing reads were analyzed with the *R.4Cker* pipeline and visualized with the *RCircos* package[76] mapped to the UCSC.HG19.Human.CytoBandIdeogram (http://genome.ucsc.edu, https://rdrr.io/cran/RCircos/man/UCSC.HG19.Human.CytoBandIdeogram.html).

## Histology and immunohistochemistry

Formalin-fixed spleens, lungs, hearts, and lymph nodes from study mice were paraffin-embedded, sectioned onto glass slides at 4 μM, then deparaffinated, hydrated with distilled water, and stained with hematoxylin and eosin (HE), or Ki67 as previously described[77]. For Ki67 staining, following hydration, sections were treated with citrate buffer pH 6.0 at 125 °C for 15 min. In all, 3% hydrogen peroxide with methanol was used to block endogenous peroxidase. A serum-free protein block was applied, incubated for 10 min and rinsed. Staining was performed by sequential application and incubation with anti-Ki67 primary antibody (Thermo Scientific, Waltham, MA) using 1:100 dilution at room temperature, biotinylated goat anti-rabbit secondary antibody (Vector Labs, Burlingame, CA, USA) for 30 min at room temperature, ABC biotinylated antibody linker (Vector Labs) for 30 min at room temperature, and DAB substrate (Dako, Carpinteria, CA) for 5 min. Hematoxylin was used as a counter-stain. All photographs were taken using an Olympus SC30 camera with an Olympus BX53 microscope.

## Single-cell library preparation and sequencing

A 10X Genomics Chromium machine was used for 3000–5000 single-cell capture. Cells were washed and partitioned into nanoliter-scale Gel Bead-In-EMulsions (GEMs), where all generated cDNA share a common 10x Genomics Barcode but uses a pool of ~750,000 barcodes to separately index each cell's transcriptome. The silane magnetic beads and Solid Phase Reversible Immobilization beads were used to clean up the GEM reaction mixture and the barcoded cDNA was then amplified in a PCR step. For mouse scRNA-seq, libraries were prepared using the Chromium Single Cell 3′ Reagent Kits (v2): Single Cell 3′ Library & Gel Bead Kit v2 (PN-120237), Single Cell 3′ Chip Kit v2 (PN-120236) and i7 Multiplex Kit (PN-120262) (10x Genomics). Murine libraries were sequenced on an Illumina Nextseq 500 to achieve 75 bp reads. Optimal cDNA amplicon size was achieved using Covaris machine prior to library construction. The P7 and R2 primers were added during the GEM incubation and the P5, and R1 during library construction via end repair, A-tailing, adapter ligation and PCR. The final libraries contain the P5 and P7 primers used in Illumina bridge amplification. Following the Single Cell 3′ Reagent Kits (v2) user guide (manual part no. CG00052 Rev A.), data was processed by the Cell Ranger pipeline (v2.0.1; 10x Genomics). Count matrices were analyzed using the Seurat package (v2.2.1) in R. Read counts were normalized to library size, scaled by 10,000, log-transformed, and filtered.

Human scRNA-seq libraries were prepared using the Chromium Next GEM Single Cell 5′ Kit (v2) (PN-1000263): Library Construction kit (PN-1000190), Chromium Next GEM Chip K Single Cell Kit (PN-1000286), Chromium Single Cell Human BCR Amplification Kit (PN-1000253), and Dual Index Kit TT Set A (PN-1000215) (10x Genomics). Human libraries were sequenced on a NovaSeq 6000 with 300 cycles and paired 150 bp reads and resulted in a minimum of >10,000 reads per cell. The data were processed by the Cell Ranger pipeline (v1.0.1; 10x Genomics). Quality control removed low-quality cells with a high percentage of reads mapped to mitochondrial genes (mito counts >2 median absolute deviations above the median) and/or low numbers of genes detected (feature counts >2 median absolute deviations below the median) using functions from *blaseRtools* (https://github.com/blaserlab/blaseRtools). The *Doubletfinder* package[78] was used to remove high-likelihood doublets. UMAP dimensionality reduction, clustering, and top marker analysis was achieved using functions from *Monocle3*[79,80]. Leiden clustering[81] and k-Means10 clustering was utilized for human and murine analyses, respectively. BCR V(D)J single-cell transcriptome analysis was analyzed using *blaseRtools* R package. Gene expression profiles for stated clusters (Fig. 1h) were aggregated for pseudobulk differential expression analysis using *Monocle3* and *DESeq2* functions[66]. UMAP plots were generated utilizing *blaseRtools*. Aggregate gene expression UMAP plots utilized the sum of the normalized per cell UMI counts for each gene set to generate the aggregate gene expression score[79].

## PRT382 studies

PRT382, by IUPAC nomenclature: (2 R,3 S,4 R,5 R)-2-[(R)-(3,4-dichlorophenyl)-hydroxy-methyl]-5-[(4Z)-4-methoxyimino-1H-pyrrolo[2,3-d]pyrimidin-7-yl]tetrahydrofuran-3,4-diol hydrochloride, was synthesized for this study by Prelude Therapeutics (Wilmington, DE; Supplemental Figs. 8, 9). Synthesis of PRT382 followed methods described in ref. 82. PRT382 drug stocks used in these studies were confirmed by Prelude Therapeutics to contain >99% purity via LC-MS and HPLC methods (Supplemental Figs. 10 and 11). Chemical identity for compound PRT382 is as follows: HNMR: 1H NMR (400 MHz, DMSO-d6) δ 8.13 (s, 1H), 7.68−7.53 (m, 3H), 7.38 (dd, J = 8.3, 2.0 Hz, 1H), 6.67 (s, 1H), 6.04 (d, J = 7.6 Hz, 1H), 4.79 (d, J = 5.5 Hz, 1H), 4.49 (dd, J = 7.6, 5.0 Hz, 1H), 4.08 (dd, J = 5.1, 1.5 Hz, 1H), 3.97 (dd, J = 5.4, 1.4 Hz, 1H), 3.82 (s, 3H). 1H NMR was carried our under ambient temperature. LC-MS (m/z): [M]+ calcd. for C18H18Cl2N4O5, 441.3; found, 443.3 (Supplemental Figs. 8–11).

EPZ015666 (cat #HY-127270, JNJ-64619178 (cat #HY-101564), LLY-283 (cat #HY-107777), and PLX-51107 (cat #HY-111422) were obtained from MedChemExpress (Monmouth Junction, NJ). PF-06855800 (cat #PZ0401) was obtained from Sigma-Aldrich Inc. (Milwaukee, WI).

For in vitro assays, the compound was stored in a 10 mM solution in DMSO. For in vivo assays, PRT382 was dissolved in a 0.5% methylcellulose, 0.1% tween solution, and administered as oral gavage. For Pharmacokinetic studies, PRT382 was formulated in 0.5% carboxymethylcellulose sodium salt + 0.5%Tween80 as a 1 mg/mL suspension for 10 mg/kg and was administered in a 10 mL/kg dose volume by oral gavage to CD-1 mice. Blood was collected at various time points after dosing into EDTA-K2 tubes, placed on ice, and stored at −80 °C prior to analysis by LC-MS/MS. For biochemical potency and selectivity, PRT382 was solubilized in 100% DMSO and diluted serially threefold. For assessment of selectivity, enzyme activity against a panel of 37 methyltransferases was evaluated using 3H-SAM as a radioligand probe with filtration binding. PRT382 was tested in duplicate at a final concentration of 10 μM. The percent of enzyme activity relative to DMSO control condition was determined for each enzyme.

For biochemical potency and selectivity, PRT382 was solubilized in 100% DMSO and diluted serially threefold. The diluted compound was further diluted in assay buffer (20 mM Tris-HCl, pH 8.0, 50 mM NaCl, 0.002% Tween20, 1 mM TCEP, 1% DMSO) for 10-point IC$_{50}$ curve

at a concentration tenfold greater than the desired assay concentration. Standard reactions were performed in a total volume of 30 μl in assay buffer, with 300 nM histone H4-based AcH4-23 (Anaspec: AS-65002) as substrate. To this was added the PRMT5/MEP50 complex diluted to provide a final assay concentration of 2.5 nM, and the compounds were allowed to preincubate for 20 min at 37 °C. The reaction was initiated by adding S-[3 H-methyl]-adenosyl-L-methionine (PerkinElmer: NET155001MC) to a final concentration of 1 μM. Following a 30 min of incubation at 37 °C, the reaction was stopped by adding 25 μL of 8 M Guanidine HCl. To each reaction, 150 μL of SPA beads suspension (Perkinelmer: RPNQ0012) was added and incubated while shaking at room temperature for 30 min. The plate was centrifuged at 100 xg for 30 s before reading in a scintillation counter. IC$_{50}$ values were determined by fitting the data to with Hill Slope using GraphPad Prism software (v9.2.0).

For western blot analysis of Cellular sDMA in Granta-519, one day before the experiment, Granta-519 cells were passaged to a density of 0.5 × 106 cells/ml. The next day, Granta-519 cells were spun down at 1500 rpm for 4 min, resuspended in fresh medium at 0.5E6 cells/ml and 3 mL of culture (1.5E6 cells) were seeded into a six-well plate. Eight-point, threefold serial dilutions of compound working stocks were added to cells (3 μl, 1:1000 dilution, DMSO concentration was 0.1%; final top concentration at 1 μM) and incubated for 3 days. Cells incubated with DMSO was used as vehicle control. Cells were harvested 3 days later, resuspended in 15 μL PBS, lysed in 4% SDS, and homogenized by passing through homogenizer column (Omega Biotek, Catalog #: HCR003). Total protein concentrations were determined by BCA assay (ThermoFisher Scientific, Catalog #: 23225). Lysates were mixed with 5× Laemmli buffer and boiled for 5 min. Forty micrograms of total protein was separated on SDS-PAGE gels (Bio-Rad, catalog #: 4568083, 4568043), transferred to PVDF membrane, and blocked with 5% dry milk (Bio-Rad, Catalog #: 1706404) in TBS with 0.1% v/v Tween20 (TBST) for 1 h at room temperature (RT), and incubated with primary antibodies (sDMA: Cell signaling, Catalog #: 13222, 1:3,000; β-actin: sigma, Catalog #: A2228, 1:5000) in 5% dry milk in TBST at 4 °C overnight. The next day, membranes were washed with TBST, 5 × 5 min, and incubated with HRP conjugated seconded antibody (GE Healthcare; Catalog #: NA9341 ML, NA931-1ML; 1:5000) for 2 hours at RT, followed by 5 × 5 min washes with TBST, and incubation with ECL substrates (Bio-Rad, Catalog #: 1705061, 1705062). The chemiluminescent signal was captured with Fluochem HD2 imager (Proteinsimple). SmD3me2s bands were quantified by ImageJ. Signals were normalized to β-Actin and DMSO control. IC$_{50}$ values were calculated using Graphpad Prism (v9.2.0) ([Inhibitor] vs. normalized response − Variable slope).

For cell growth curves in HG3 and Mec-1 cells, 3E5 cells were plated in 2 mL culture media with indicated dose PRT382 or comparator compound. Live cells/mL were counted on days 3, 6, 10, and 12. At indicated time points, cells were pelleted and 3E5 live cells were removed and resuspended in 2 mL fresh culture media with fresh dose of PRT382 or comparator compound. The split factor was calculated at indicated time points by dividing the counted cells/mL by 3E5. Cell growth was extrapolated over time by multiplying the subsequent cell count by the previous split factor. Cell growth is plotted as fold change relative to DMSO-treated cells.

## Statistical analysis

All analyses were performed by the OSU Center for Biostatistics using previously described models[57] using SAS/STAT software (v9.4; SAS Institute, Inc., Cary, NC). For survival experiments, survival curve estimates were calculated using the Kaplan−Meier method, and differences in curves were assessed using the log-rank test. Spleen volume was compared between groups using ANOVA, and mixed effects models were used to assess changes in %CD19$^+$/CD5$^+$ cells over time. Differences in PBL at different time points were assessed using a mixed effects model, and a two-sample *t* test assuming unequal

variances was used to compare spleen weight. Where applicable, data were log-transformed to reduce skewness. *P* values were adjusted for multiple comparisons using Holm's stepdown procedure or Dunnett's test, where appropriate. Chemical structures were generated using ChemDraw JS for the web (PerkinElmer Informatics).

## Reporting summary
Further information on research design is available in the Nature Portfolio Reporting Summary linked to this article.

## Data availability
The RNA-sequencing, scRNA-seq, and ATAC-seq data generated in this study have been deposited in the GEO database under accession code #GSE183432. PRT382 is a preclinical tool compound that it is not commercially available but can be obtained from OSU or Prelude therapeutics upon request. CLL and RT gene expression sets derived from Nadeu and colleagues[31] (https://doi.org/10.1038/s41591-022-01927-8) in Supplemental Table 11b. The GRCm38 mouse reference genome is available at (https://www.ncbi.nlm.nih.gov/assembly/GCF_000001635.20/). The UCSC.HG19.Human.CytoBandIdeogram is available at https://rdrr.io/cran/RCircos/man/UCSC.HG19.Human.CytoBandIdeogram.html. Source data are provided with this paper.

## Code availability
ScRNA-seq analysis scripts are provided at https://github.com/blaserlab/lapalombella_whipp. A companion R data package is available at zenodo.org[83].

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

## Acknowledgements

We are grateful to the patients who provided blood for the above studies and to the OSU Comprehensive Cancer Center Leukemia Tissue Bank (supported by NCI P30 CA016058) for sample procurement. X-ray diffraction data were collected at beamline 8.3.1 at the Advanced Light Source (Lawrence Berkeley National Laboratory), beamline 08ID-1 at Canadian Light Source and beamline 7.1 at the Stanford Synchrotron Radiation Lightsource (a directorate of the SLAC National Accelerator Laboratory). Authors would also like to acknowledge Alan Flechtner, HTL and The Ohio State University Veterinary Histology Laboratory for their assistance in immunohistochemical studies, also supported by NCI P30 CA016058, and Amy Porter of the Michigan State University Investigative Histopathology Laboratory. This study makes use of RNA-seq data generated by the Blueprint Consortium. A full list of the investigators who contributed to the generation of the data is available from www.blueprint-epigenome.eu. This work was supported by the National Cancer Institute R01 CA177292 (R.L.), R01 CA214046 (R.L.), CCC core grant P30 CA016058, P01CA214274 (R.L., R.B., and S.C.-K.), NIGMS T32 GM068412 (J.S.W.), NCATS TL1 TR002735 (J.S.W.), Eli Lilly Fellowship in Pharmaceutics (E.C.W.), a Research Scholar grant 129863-RSG-16-158-01-CDD (R.L.) from the American Cancer Society, The OSU Comprehensive Cancer Center using Pelotonia funds, and further research support to the EHL Laboratory from the Four Winds Foundation, the D. Warren Brown Foundation, the Connie Brown CLL Research fund, and the Sullivan CLL Research Foundation.

## Author contributions

Z.A.H., J.S.W., and E.C.W. designed and conducted experiments, generated data and figures, analyzed the data, interpreted results, and wrote the manuscript. L.B., M.C., P.Z., S.S., F.B., P.L.S., C.A., S.A.P., J.N.S., K.W., C.B.C., H.P., J.S., B.K.H., P.S., M.W., P.P., C.X., and K.V. conducted experiments and generated figures. L.P.B. conducted mouse experiments. Z.A.H., K.W., V.C., and R.L. contributed to the generation of the mouse model. A.P., K.S., C.T.G., G.H.O., S.A.Y., A.L.M., and L.Y. conducted the statistical analysis. P.Y., S.C.-K., J.W., J.S.B., L.A., and Y.Y. interpreted the results and reviewed the manuscript. J.C.B., R.A.B., B.W.B., and R.L. planned the project, acquired funding, supervised the study, interpreted results, and reviewed the manuscript. These senior authors (B.W.B. and R.L.) contributed equally. All authors read and approved the final version of the manuscript.

## Competing interests

R.B. has received research funding from Prelude Therapeutics. P.S., M.W., P.P., C.X., and K.V. are employees of Prelude Therapeutics. All the remaining authors declare no competing interests. This does not alter our adherence to *Nature Communications* policies on sharing data and materials.

## Additional information

¹Division of Hematology, Department of Internal Medicine, The Ohio State University, Columbus, OH, USA. ²Haematopathology Unit, IRCCS Azienda Ospedaliero-Universitaria di Bologna, Bologna, Italy. ³European Institute of Oncology, Istituto di Ricovero e Cura a Carattere Scientifico (IRCCS), Milan, Italy. ⁴Department of Specialized, Experimental and Diagnostic Medicine, University of Bologna, Bologna, Italy. ⁵Department of Biomedical Informatics, The Ohio State University, Columbus, OH, USA. ⁶Department of Veterinary Biosciences, The Ohio State University, Columbus, OH, USA. ⁷Department of Pathobiology and Diagnostic Investigation, College of Veterinary Medicine, Michigan State University, East Lansing, MI, USA. ⁸Center for Biostatistics, The Ohio State University, Columbus, OH, USA. ⁹Department of Cancer Biology and Genetics, The Ohio State University, Columbus, OH, USA. ¹⁰Prelude Therapeutics, Wilmington, DE, USA. ¹¹Department of Pathology and Laboratory Medicine, Weill Cornell Medicine, New York, NY, USA. ¹²Department of Internal Medicine, University of Cincinnati, Cincinnati, OH, USA. ¹³These authors contributed equally: Zachary A. Hing, Janek S. Walker, Ethan C. Whipp, Bradley W. Blaser, Rosa Lapalombella. ✉e-mail: rosa.lapalombella@osumc.edu

