## [Peer Review File · Nature Communications]

Dysregulation of PRMT5 in Chronic Lymphocytic Leukemia Promotes Progression with High Risk of Richter's TransformationREVIEWER COMMENTS

Reviewer #1 (Remarks to the Author); expert in PRMT inhibitors:

Manuscript by Hing et al is an extensive analysis of the role PRMT5 plays in CLL, its progression to B cell lymphoma and downstream disease driver oncogenic pathways. The authors utilized several animal models investigating gene expression, splicing, chromatin openness and architecture to determine gene signatures in CLL and RT. Overall the manuscript provides interesting insights into CLL and RT in the context of PRMT5.

Major issues

1. The statistics in the manuscript are rather murky. The N is not stated, the statistical analysis methods are mostly discussed for RNAseq. The replicate numbers and statistical significance should be stated in each figure as per Nature Communications standards.
2. The authors claim the superior properties of PRT382. However, they compare it to the early EPZ-GSK compound that has a different MOA and a very different structure. PRT382 should be compared to other SAM-mimetic compounds such as LLY-283, PF-06939999, PF-0685580, JNJ-64619178. That would help understand where PRT382 came from as there is no discovery or SAR described or references provided, so it looks like it just got discovered. Especially interesting would be to compare PRT382 to LLY-283 as there is only a methoxy and 2 Cl groups that are different between these compounds. And the cellular IC50 is very similar: 20nM vs 25nM. Thus, claiming that PRT382 is superior maybe misleading. The authors should also refer to original publications for this compound series development and compound use in vivo. This will help the scientific community understand the results that have been surfacing by using the EPZ015666 compound ie reported lack of activity (PMID32641491).
3. Fig 6B selectivity data should be performed with 1uM and 100nM of the compound as the activity in panel C is 2nM IC50, approx. 100nM IC90. The number of replicates in panel C is not stated.
4. The number of replicates and the assay methods for Fig 6D are not provided. Suppl Fig 6A has a type in SMDA, should be SDMA, and its not clear what the western band represents. The SDMA is found on many proteins thus, the gels should look like Fig 7J.

Minor comments

1. Given the potency of PRT382, why 1uM is needed to have cell inhibitory effect in Fig 7C over 12 days with 100nM showing only minimal effects? Since the methods for PRT382 use were described very poorly, it is not clear if the compound was added once in 12d.
2. The assay in Fig 7E and Suppl 6B needs a better description.
3. Was PRMT5 not in the top 50 regulated genes in Fig 1E?
4. Why ATF3 and ATF4 are considered cluster 4 genes in Fig 3D but UPR and ER stress pathways they regulate are enriched in cluster 3 (Suppl 2E)?
5. Line 392 states optimal target binding while no binding but enzymatic assays were performed
6. Line 165 typo in activating
7. Titles for Suppl Figures would be helpful
8. Amidoxime in Fig 6B legend needs to be explained

Reviewer #2 (Remarks to the Author); expert in scRNAseq, ATAC-seq, 4C and CLL:

Hing and colleagues propose in this study a role for PRMT5 in the transformation of CLL to Richter's disease. As the molecular mechanisms of this transformation are poorly understand and there is a high clinical need for a better understanding of this aggressive disease, the study and findings are highly interesting and of clinical importance.

By using bulk and single-cell RNA-sequencing of patient and murine samples of CLL and RT, the study provides important insights into disease biology. Transgenic PRMT5 mouse models and pre-clinical treatment studies with a novel PRMT5 inhibitor provide evidence for an involvement of this methyltransferase in disease progression and suggest it as a novel target for therapy. In addition, the

generation of a new mouse line that models RT and can be used for further preclinical analyses is of high value for follow-up studies.

The manuscript is clearly written and the data presentation and discussion mostly valid and comprehensible.

I have only a few minor comments to improve the display and description of data:

Lines 259-261: Is the difference in B220dim vs bright signal mainly due to the difference in size of the cells? Or do the authors claim a differential expression of this molecule in the two types of B cells?

Line 283: There is no Figure 4G, this needs to be changed to 4C.

Lines 291-294: "... the extent of the neoplastic infiltrates appeared more pronounced in Eμ-PRMT5/TCL1 animals..." This is not clearly visible in Figure 4E and should be shown in a better way.

Lines 294-298: Where can one see what is stated in this paragraph? The display of these observations needs to be improved.

Figure 5A: It would be helpful to add a principle component analysis of this data to see the clustering of the 4 different groups analysed here, instead of only showing a heatmap of the DEG.

Lines 158 and 463: The authors describe a likely contribution of IL4I1 to CLL-to-RT progression via promoting immune suppression. The original article describing a role for IL4I1 in immune suppression in CLL needs to be cited here (Sadik et al., Cell 2020).

Reviewer #3 (Remarks to the Author); expert in clinical and preclinical CLL:

This paper by Hing et al described that mice model with overexpressing hPRMT5 developed a B-lymphoid expansion. The mixed double transgenic mice with Eu-PRMT5/TCL-1 developed a more aggressive lymphoid disease resembling Richter transformation of CLL. The authors further studied the oncogenic pathways regulated by this transgenic model. They also reported the development of a PRMT5 inhibitor, PRT382. They argued that further clinical test of PRMT5 inhibitor, e.g, PRT382 maybe potentially explored for clinical therapeutic trial for RT.

Even though this above finding of this mice model seems interesting, but many questions remain unclear.

1. In the introduction in Page 3, the author describe RT as the most common complication of CLL. This is not correct. RT is a disease evolution from CT, meaning disease evolved into more aggressive form of lymphoma. This needs to correct. In, addition, RT is chemotherapy-resistant, would not use therapy-resistant. This is because there are some experiences using variable targeted, immune checkpoint, as well CART therapy for RT.

2. It seems rather weak to link PRMT5 with RT. The authors cited several references in 20-27 about the important role of PRMT5 in cancer or regulating MYC and Cyclin D1, however, lack specific reason or logic to focusing on this particular protein/gene PRMT5 in RT pathogenesis, making it less logic to study and test this protein.

3. In Figure 1, PRMT5 expression were shown in CLL and pre-RT samples by western blot. However, it is unclear whether any of these samples were consecutive samples, meaning from classic CLL to pre-RT phase. If they are sequential samples, what is the time interval or therapy pattern between these samples?

Also, it will make sense to see the PRMT5 expression level with MYC rearrangement/protein expression relationship. This part of study need to provide more details.

4. For figure 1C staining, it will be important to provide costaining of PRMT5 with a B cell or T cell markers to know the relative expression in different lineages.

5. In figure 2, given the documented mixed cellular population in PRMT5 mice (in 2D), it is important

to show the specific B/T/Myeloid markers in these organs shown in 2G.

6. PRMT5 mice seems to have more diverse clonal population compared to TCL1 mice, how will author explain these results in 2H?

7. Staining the specific B, T and myeloid markers will be critical to show the disease in this mouse model (figure 4C and 4E).

8. How would author explain the PRMT5/TCL-1 mice lymphoid atrophy developed in 6 months? How is this similar to human RT tumor? Careful histology comparison will be needed from pathology point of view.

9. While the DEG analysis of 3 vs. 6 months of PRMT5/TCL-1 mice was informative, it is unclear how this is resemble human RT transcriptome analysis or purely representing genes regulated by PRMT5/TCL-1 overexpression.

10. What is EPZ15666?

Below, we have addressed all reviewer remarks in a point-by-point fashion:

REVIEWER COMMENTS

Reviewer #1 (Remarks to the Author); expert in PRMT inhibitors:

Manuscript by Hing et al is an extensive analysis of the role PRMT5 plays in CLL, its progression to B cell lymphoma and downstream disease driver oncogenic pathways. The authors utilized several animal models investigating gene expression, splicing, chromatin openness and architecture to determine gene signatures in CLL and RT. Overall the manuscript provides interesting insights into CLL and RT in the context of PRMT5.

Major issues

1. The statistics in the manuscript are rather murky. The N is not stated, the statistical analysis methods are mostly discussed for RNAseq. The replicate numbers and statistical significance should be stated in each figure as per Nature Communications standards.

- We thank the reviewer for suggesting we improve our reporting of experiment details in this manuscript. In the track-changes version of this resubmission you will find the figure legends have been updated to include these details and the methods section has been improved to further describe the experiments in this study.

2. The authors claim the superior properties of PRT382. However, they compare it to the early EPZ-GSK compound that has a different MOA and a very different structure. PRT382 should be compared to other SAM-mimetic compounds such as LLY-283, PF-06939999, PF-0685580, JNJ-64619178. That would help understand where PRT382 came from as there is no discovery or SAR described or references provided, so it looks like it just got discovered. Especially interesting would be to compare PRT382 to LLY-283 as there is only a methoxy and 2 Cl groups that are different between these compounds. And the cellular IC50 is very similar: 20nM vs 25nM. Thus, claiming that PRT382 is superior maybe misleading. The authors should also refer to original publications for this compound series development and compound use in vivo. This will help the scientific community understand the results that have been surfacing by using the EPZ015666 compound ie reported lack of activity (PMID32641491).

- We agree with these suggestions and thank the reviewer for inviting us to expand our analysis of our newly presented PRMT5 inhibitor compound PRT382. To expand our analysis of PRT382 with other newly developed PRMT5 inhibitors we have added several comparisons that we believe enhances our manuscript.
- In **Figure 6A**, we have added the chemical structures for compounds JNJ-64619178, LLY-283, and PF-06855800 to demonstrate to the reader the difference in chemical structure between compounds. Indeed, PRT382 is

distinguishable by a 3,4-dichlorobenzene at the 5' position of the ribose and a methoxyamine group at the 6' position of the purine.

- As suggested, we then compared cytotoxic efficacy of PRT382 against these other PRMT5 inhibitors in addition to EPZ015666, which has a different mechanism of action, in **Figure 7D**. We observed compounds JNJ-64619178, LLY-283, and PF-06855800 to contain similar anti-proliferative potency to PRT382 at high doses (1 μ M), but other compounds were more potent at lower doses (10nM, 100nM). This result was not surprising, as these compounds have been reported with broad specificity also targeting other methyltransferases. Altogether, as PRT382 is specific for targeting PRMT5 and demonstrates anti-tumor efficacy in CLL cell lines HG3 and Mec-1, these results support our claims for PRMT5 as a critical factor in aggressive CLL and is encouraging that blocking PRMT5 activity alone can produce such evident anti-proliferative effects.
- We agree with the reviewer that claiming PRT382 is broadly superior is misleading. We have amended the text to remove this language from the manuscript. We further demonstrate in **Figure 7** that *in vivo* treatment with the SAM-competitive PRMT5 inhibitor PRT382 results in superior anti-tumor effects when compared to the SAM non-competitive inhibitor EPZ015666, demonstrating molecules with competitive mechanism of action should be considered for these tumor types and that targeting PRMT5 alone influences tumor progression in *in vivo* murine models.

Figure 6

Figure 7

Abstract:

Richter's Transformation (RT) is a poorly understood and fatal progression of chronic lymphocytic leukemia (CLL) manifesting histologically as diffuse large B-cell lymphoma. PRMT5 is implicated in lymphomagenesis, but its role in CLL or RT progression is unknown. We demonstrate herein that tumors uniformly overexpress PRMT5 in patients with progression to RT. Furthermore, mice with B-specific overexpression of hPRMT5 developed a B-lymphoid expansion with increased risk of death, and E μ -PRMT5/TCL1 double transgenic mice developed a highly aggressive disease **with transformation** that histologically resembles RT; where large-scale transcriptional profiling identified unique oncogenic pathways mediating PRMT5-driven disease progression. Lastly, we report the development of a novel **SAM-competitive** PRMT5 inhibitor, PRT382, **with exclusive selectivity and optimal** *in vitro* and *in vivo* activity compared to available PRMT5 inhibitors. Taken together, the discovery that PRMT5 drives oncogenic pathways promoting RT provides a compelling rationale for clinical investigation of PRMT5 inhibitors such as PRT382 in aggressive CLL/RT cases.

Introduction:

We herein demonstrate that RT tumors overexpress PRMT5, and PRMT5 upregulation may be observed in some cases up to one year prior to RT diagnosis. We then developed and characterized the first murine model of B cell-specific expression of human PRMT5 (E μ -PRMT5), producing an overt B-lymphoid expansion eventually developing a CLL-like phenotype. When crossed with the E μ -TCL1 model, double transgenic mice uniformly developed a highly aggressive disease with **select** focal lymphoid tumors **transforming to** histologically resemble RT, altogether proposing a direct cause-effect relationship for PRMT5 in the pathogenesis of RT and identifying an attractive therapeutic target for RT and CLL at risk of RT. Lastly, we report development of a novel inhibitor (PRT382) with enhanced PRMT5 selectivity, improved tolerability, and **optimal** *in-vitro* and *in-vivo* anti-tumor activity compared to currently available inhibitors.

Results:

Identification of a novel selective inhibitor of PRMT5

The emerging role for PRMT5 activity in both solid and hematologic tumors has spurred development of several drug candidates with uncompetitive (i.e. compound EPZ015666⁴⁵) and competitive (i.e. compounds JNJ-64619178⁴⁶, LLY-283⁴⁷, PF-06855800⁴⁸) inhibition of SAM-mediated PRMT5 enzymatic activity. However these compounds have been reported with limited specificity for PRMT5 among other methyltransferases, restricting potential for clinical translatability as off-target effects disrupting processes essential in normal hematopoiesis are of significant concern.

Thus, **with** the intent to therapeutically target PRMT5 **in the context of** CLL at risk of RT, we developed **tool compound** PRT382 (**Figure 6A**), a novel class of potent, selective PRMT5 inhibitors. **Structural similarity between PRT382 and adenosine-backbone inhibitors JNJ-64619178, LLY-283, and PF06855800 suggests a comparable SAM-competitive mechanism, however PRT382 is distinguishable by a 3,4-dichlorobenzene at the 5' position of the ribose and a methoxyamine at the 6' position of the purine.** PRT382 is selective for PRMT5 alone, demonstrating an IC₅₀ >1 μ M against 38 other methyltransferases (e.g DNMTs, MLL1-4 complexes, EZH1 and EZH2 complexes, NRMTs, NSD, SUVs, SMYs, SETs, PRDM9), and other members of the PRMT family (**Figure 6B**). PRT382 demonstrated optimal enzymatic **kinetics** *in-vitro*, producing an IC₅₀ of 2.8 nM in filtration binding assays with recombinant human PRMT5/MEP50 and histone H2A as the protein substrate (**Figure 6C**), and an IC₅₀ of 27 nM in a reducing sDMA assay in the Granta-519 lymphoma cell line (**Figure 6D**). Anti-proliferative efficacy was demonstrated in the leukemia and lymphoma cell lines CCMCL-1, JVM-2, U-2932, RL, and KARPAS-422 with IC₅₀ values ranging between 10-100nM (**Figure 6E, F**).

In-vivo, PRT382 demonstrated low clearance and high oral bioavailability in mice, rats, and dogs (F%=70-80%), and high AUC levels (1175 hr*kg*ng/mL/mg) in mice (data not shown). Delivery of PRT382 at 10, 30 and 100mg/kg was well tolerated in wild type C57BL/6 mice, leading to peak plasma levels of 12, 32, and 75 μ M, respectively, without observable toxicity after 5 daily oral doses (**Figure 6G**).

PRMT5 blockade displays antileukemic properties in aggressive CLL

Cas9-mediated doxycycline inducible knockout of PRMT5 blocked cell proliferation in the CLL cell line Mec-1 (**Figure 7A, B**). Further, in CLL cell lines HG3 and Mec-1, 72h PRT382 treatment resulted in a decrease in global symmetric dimethyl arginine (SDMA) residues at doses as low as 100nM, indicating potential efficacy for PRT382 against CLL cells (**Figure 7C**). Consistent with reduction in global SDMA, PRT382 treatment induced time and dose-dependent growth inhibition beginning at 100nM in the HG3 and Mec-1 cell line (**Figure 7D**). No changes in growth potential were observed using the SAM non-competitive PRMT5 inhibitor EPZ015666⁴⁵, which coincided with only a modest decrease in global symmetric dimethyl arginine residues at higher concentrations (*Supplemental Figure 6A*). PRT382 demonstrated similar-to-reduced anti-proliferative potency in HG3 and Mec-1 cells when compared to newly developed SAM-competitive PRMT5 inhibitor compounds JNJ-64619178, LLY-283, and PF-06855800, although this result was expected due to the wider selectivity profile reported for these compounds⁴⁶⁻⁴⁸. Additionally, in primary CLL cells (n=12 IGHV-U patients) with and without TLR9 stimulation using synthetic CpG oligodeoxynucleotide, 72h PRT382 treatment resulted in dose-dependent reduction in proliferative potential *in-vitro* (*Supplemental Figure 6B*).

To evaluate the *in vivo* anti-tumor effect of PRT382 we first confirmed that the E μ -PRMT5/TCL1 bulk tumor population was amenable to adoptive transfer into immunocompetent C57BL/6J syngeneic mice. This established a circulating malignancy at a median time of 21 days after injection, accumulating a B220lowCD19+CD5+ CLL like population in the blood of all recipients (n=7 mice; *Supplemental Figure 6C*). Left untreated, engrafted mice succumbed to the expanding leukemia/lymphoma with a median overall survival of 67 days (range 66-75 days, *Supplemental Figure 6D*), and tumor histology was similar to that of transgenic E μ -PRMT5/TCL1 mice with spleen and lymph node involvement and bone marrow infiltration (*Supplemental Figure 6E*).

A cohort of similarly engrafted mice were followed and randomly enrolled into treatment groups (10 mg/kg PRT382, 50 mg/kg EPZ015666, vehicle control; **Figure 7E**) upon evidence of circulating disease (>10% CD19+CD5+ cells) to evaluate PRT382 efficacy *in vivo* and to further compare efficacy against the non-competitive PRMT5 inhibitor EPZ015666. PRT382 treatment reduced the rate of peripheral disease early in disease progression (weeks 2-7 post engraftment) compared to mice treated with EPZ015666 or vehicle (**Figure 7F**). PRT382-induced delay in leukemic progression led to a significant extension in survival compared to mice treated with EPZ015666 or vehicle (**Figure 7G**). Spleens from PRT382 treated mice had significantly reduced weight and volume compared to those treated with either EPZ015666 or vehicle (**Figure 7H**, *Supplemental Figure 6F*). Immunoblot analysis of splenic B cells from mice within each treatment group confirmed PRMT5 target modulation in all PRT382 but not EPZ015666 treated mice (**Figure 7I**).

Histopathologic assessment revealed normal lymph node architecture was effaced by sheets of large neoplastic lymphocytes with occasional mitotic figures, and expansion of splenic white pulp by neoplastic lymphocytes was apparent in all treatment groups (**Figure 7J**). PRT382 treated mice displayed a decrease in leukemic expansion across non-lymphoid tissues compared with vehicle or EPZ015666 treated mice, notably demonstrated by reduced numbers of sinusoidal (arrows) and portal/periportal (P) lymphocytes accumulating in the liver. There was no observable treatment related toxicity, bone marrow depletion, or GI effects in any treatment group. Taken together, our findings suggest PRMT5 is a promising therapeutic target in CLL/RT and warrants clinical investigation in patients with aggressive CLL and RT.

Discussion:

Critically, we show that genomic ablation of PRMT5 in CLL cells decreased cell proliferation, supporting inhibition of PRMT5 as a promising therapeutic target in CLL. Recently, a suite of SAM-competitive PRMT5 inhibitors have been developed⁴⁶⁻⁴⁸ however the diverse array of targets affected by these compounds may limit their progression in the clinical setting due to unwanted cytotoxicity in healthy tissue. To address this issue, we herein report here the development of a novel small molecule PRMT5 inhibitor with predicted SAM-competitive mechanism of action, PRT382, which is highly selective for PRMT5 against a wide range of other methyltransferases. PRT382 demonstrated *in vitro* and *in vivo* efficacy against CLL cell lines, isolated CLL patient cells, and the CLL-like cells from the E μ -PRMT5/TCL1 mouse model of CLL/RT, overall

providing strong evidence supporting the use of targeted PRMT5 inhibition in CLL with elevated PRMT5 expression at high risk of RT. As expected, due to restricted selectivity, PRT382 demonstrated reduced cytotoxic potency against CLL cells compared to compounds with similar mechanism of action. However, when compared with the SAM non-competitive PRMT5 inhibitor EPZ015666, PRT382 demonstrated superior *in-vitro* anti-tumor activity and improved survival in the Eμ-PRMT5/TCL1 mouse model of CLL/RT *in-vivo*.

3. Fig 6B selectivity data should be performed with 1uM and 100nM of the compound as the activity in panel C is 2nM IC50, approx. 100nM IC90. The number of replicates in panel C is not stated.

- In Figure 6B we demonstrate selectivity of PRT382 at 10 uM for PRMT5 over numerous other tested methyltransferases. At this concentration PRT382 did not hit other tested methyltransferases more than 20% inhibition, suggesting efficacy of PRT382 at this concentration, and even more so at lower concentrations (i.e. 2.8 nM – IC50 as shown in Fig6C), would primarily be due to on target PRMT5 inhibition. We chose to report specificity at 10 uM as the most informative data to the reader as 10 uM is the highest concentration PRT382 used in subsequent evaluation in the panels following Fig 6B. Thus the authors do not believe additional selectivity plots at lower concentrations would benefit this manuscript.
- Additionally, publications describing selectivity data for compounds JNJ-64619178 (Ref # - 10.1158/1535-7163.MCT-21-0367), LLY-283 (Ref # - 10.1021/acsmedchemlett.8b00014), and PF-0685580 (Ref # - 10.1158/1535-7163.MCT-21-0620) do so at 10uM, thus allowing further clarity as the reader is able to compare selectivity of compound PRT382 with other PRMT5 inhibitors.
- We thank the reviewer for the comment regarding Figure 6C. The legend has been updated to display the number of replicates used in this study.

“C) Filtration binding assays in presence of recombinant human PRMT5/MEP50 and histone H2A. Plot is representative of N=3 experiments.”

4. The number of replicates and the assay methods for Fig 6D are not provided. Suppl Fig 6A has a typo in SMDA, should be SDMA, and its not clear what the western band represents. The SDMA is found on many proteins thus, the gels should look like Fig 7J.

- We again thank the reviewer for the comment regarding Figure 6D. The legend has been updated to display the number of replicates used in this study.

“D) Reducing sDMA assay in the Granta-519 lymphoma cell line. Plot is representative of N=3 experiments.”

- We have amended the western blot in Suppl. Figure 6A to expand the range of proteins evaluated for SDMA.

Minor comments

1. Given the potency of PRT382, why 1uM is needed to have cell inhibitory effect in Fig 7C over 12 days with 100nM showing only minimal effects? Since the methods for PRT382 use were described very poorly, it is not clear if the compound was added once in 12d.

- We thank the reviewer for this question. Indeed, filtration binding assay and reducing sDMA assay demonstrated efficacy for PRT382 at lower concentrations than 100nM, which the lowest dose observed to reduce cell growth in HG3 and Mec-1 cell lines in Figure 7D. The authors believe these in-vitro assays are informative of the on-target efficacy of PRT382, but are not entirely predictive of the effect on cell growth and survival upon exposure to this compound. In Figure 7C, we demonstrate dramatic loss of global sDMA was only observed at doses 100nM and greater. When paired with data from Figure 7D, it is then suggested 100nM dose PRT382 is likely required to induce anti-proliferative effect on treated cells.
- We additionally thank the reviewer for addressing the need for us to improve the methods reported for the data in Figure 7C. The axis label, methods section, and legend for Fig7C has been updated to better reflect the methods used in this study.

“For cell growth curves in HG3 and Mec-1 cells, 3E5 cells were plated in 2mL culture media with indicated dose PRT382 or comparator compound. Live cells/mL were counted on days 3, 6, 10, and 12. At these indicated time points, cells were pelleted and 3E5 live cells were removed and resuspended in 2mL fresh culture media with fresh dose of PRT382 or comparator compound. Split factor was calculated at indicated time points by dividing the counted cells/mL by 3E5. Cell growth was extrapolated over time by multiplying the subsequent cell count by previous split factor. Cell growth is plotted as fold change relative to vehicle treated cells.”

“D) Dose-dependent decrease in proliferative potential as measured by cell growth in CLL cell lines HG3 and Mec-1 upon PRMT5 inhibition with PRT382 (n=3 independent experiments), EPZ015666 (n=4), JNJ-64619178 (n=2), LLY-283 (n=2), and PF-06855800 (n=2). Cell growth is plotted as fold change relative to vehicle treated cells \pm SD. Live cells/mL were counted on days 3, 6, 10, and 12. Fresh culture media with fresh dose of PRMT5 inhibitor at indicated concentrations were supplied at days 3, 6, 10, and 12 to continuous exposure over 12 day treatment period.”

2. The assay in Fig 7E and Suppl 6B needs a better description.

- We thank the reviewer for this suggestion to improve reporting of these data. This data has been moved to Suppl. Figure 6B. The Figure legend Suppl. Fig 6B have been updated to better describe this data.

“E) Dose-dependent decrease in proliferation, measured via MTS assay, for primary IGVH-U CLL cells (n=12 patients) upon 72 hr PRMT5 inhibition with increasing doses of PRT382 with and without TLR9 stimulation using 21.7 μ g/mL synthetic CpG oligodeoxynucleotide. Indicated doses of PRT382 were given to cells once at initiation of this study. Circles represent individual patients.”

- Suppl Figure 6B have been updated to better reflect this assay and now include all points shown.

3. Was PRMT5 not in the top 50 regulated genes in Fig 1E?

- The reviewer is correct *PRMT5* was not found among the top 50 enriched genes in Figure 1E. Figure 1E is demonstrating Log2FC of genes enriched in cluster 5 or cluster 8 compared with other clusters in these patients. As evident in Figure 1D, *PRMT5* is expressed to a lesser extent in multiple clusters from these patients, including clusters 2, 3, and 6. While indeed cells with highest *PRMT5* expression were predominantly identified in clusters 5 and 8, other genes were observed with elevated Log2FC comparing Cluster 5 or Cluster 8 to other cell clusters, thus *PRMT5* is not displayed in Figure 1E.

4. Why ATF3 and ATF4 are considered cluster 4 genes in Fig 3D but UPR and ER stress pathways they regulate are enriched in cluster 3 (Suppl 2E)?

- The reviewer is correct is discerning ATF3 and ATF4 were observed as upregulated genes in cluster 4 yet unfolded protein response (UPR) and endoplasmic reticulum stress pathways (ER stress pathways) were enriched in cluster 3. Cluster 3 was nearly exclusively found in Eu-TCL1 mice and cluster 4 was enriched in Eu-*PRMT5* mice. Enrichment of genes including *EDEM1*, *HSP90B1*, *SEL1L*, *XBP1* contributed to identification of UPR and ER Stress pathways enriched in cluster 3. Thus, multiple dysregulated genes appear to be converging on UPR and ER stress pathways and are driving this result.
- We do appreciate the reviewer bringing the point of ATF3/4 to our attention, and we have identified this as an area where increased discussion would improve the scope of our manuscript. High expression of ATF3 has been shown as an important element controlling the growth of ABC DLBCL (10.1182/blood-2015-07-655647 – New ref #49). It has also been shown that Myc induces expression of ATF4, and expression of ATF4 has been shown to play a critical role in supporting cell adaptation and survival during MYC-dependent tumor growth and progression (10.1038/s41556019-0347-9 – New ref #50). Further discussion of ATF3/4 has been added to the discussion with these citations.

“While the overall transcriptome between Eμ-*PRMT5* and Eμ-TCL1 tumors were broadly similar, a unique expansion of proliferating clones characterized by activation of oncogenic pathways including *Myc*, *ATF3/4*, *Il-4*, *Ccr7*, *Cd69*, *Cdk4*, *Cxcr5*,

Npm1, *Birc5*, and **AP-1 transcription factor complex members *Jun* and *Fos*** were observed in E μ -PRMT5 mice. Our data also suggest it is the accumulation of these transcriptional changes within discrete clonal populations that drive disease initiation and progression. Manipulation of *Myc*, *Npm1*, and *Birc5* present well-defined roles in leukemogenic transformation, and Cd69 expression on CLL cells has been discussed as a critical marker for predicting CLL prognosis⁴⁸. **Elevated ATF3 expression has been identified as a hallmark of samples from patients with activated B-cell (ABC) type DLBCL⁴⁹, particularly relevant to this study as an overwhelming majority of RT cases transform from CLL to ABC-type DLBCL.** *Myc* activation has also been observed to upregulate ATF4, where ATF4 plays a critical role in supporting cell adaptation and survival during MYC-dependent tumor growth⁵⁰, suggesting an additional cooperative axis between PRMT5, MYC, and ATF4 in RT. Further, the chemokine receptor *Ccr7* plays a pivotal role in directing tumor cells to lymphoma-supporting niches in the E μ -*Myc* mouse model⁵¹, suggesting PRMT5 hyperactivity may also encourage honing and accumulation in these secondary lymphoid organs. Moreover, significantly enriched genes in distinct E μ -PRMT5 populations overlapped with PRMT5-expressing cells from human RT tumors, establishing a PRMT5-driven gene signature in aggressive mouse and leukemia.”

5. Line 392 states optimal target binding while no binding but enzymatic assays were performed

- We thank the reviewer for notifying us of this error. We have updated the text to reflect this change:

“PRT382 demonstrated optimal **enzymatic** kinetics in-vitro, producing an IC50 of 2.8 nM in filtration binding assays with recombinant human PRMT5/MEP50 and histone H2A as the protein substrate (**Figure 6C**), and an IC50 of 27 nM in a reducing sDMA assay in the Granta-519 lymphoma cell line (**Figure 6D**).”

6. Line 165 typo in activating

- We thank the reviewer for catching this error. This has been fixed.

“Ingenuity Pathway Analysis of top enriched genes in clusters C5 and C8 also reflected these immunomodulatory and **activating** signaling pathways stemming from the BCR axis (Supplementary Figure 1C).”

7. Titles for Suppl Figures would be helpful

- We thank the reviewer for this suggestion. Titles for Suppl figures have been added.

“**Supplemental Figure 1: Analysis of CLL-to-RT disease burden reveals expansion of clonally related cells containing tumor subsets with distinct transcript profile**”

“**Supplemental Figure 2: CLL-like phenotype in the E μ -PRMT5 mouse model**”

“**Supplemental Figure 3: Accelerated tumor kinetics and time to death in the E μ -PRMT5/TCL1 mouse model**”

“**Supplemental Figure 4: CLL-like phenotype in the E μ -PRMT5/TCL1 mouse model**”

“**Supplemental Figure 5: scRNA-seq reveals differential tumor transcriptome in lymph node tissue from E μ -PRMT5/TCL1 and E μ -TCL1 mouse models**”

“**Supplemental Figure 6: In vitro and in vivo efficacy of targeted PRMT5 inhibition in CLL and CLL-like cells**”

“**Supplemental Figure 7: BRD4 occupying promoter region reveals role in PRMT5 expression**”

8. Amidoxime in Fig 6B legend needs to be explained

- We thank the reviewer for this suggestion. We have updated the figure 6B legend.

“B) Selectivity of PRT382 for PRMT5 in contrast to other **methyltransferases** and PRMT family members. Circle size represents percent inhibition at 10 μ M.”

Reviewer #2 (Remarks to the Author); expert in scRNAseq, ATAC-seq, 4C and CLL:

Hing and colleagues propose in this study a role for PRMT5 in the transformation of CLL to Richter's disease. As the molecular mechanisms of this transformation are poorly understood and there is a high clinical need for a better understanding of this aggressive disease, the study and findings are highly interesting and of clinical importance.

By using bulk and single-cell RNA-sequencing of patient and murine samples of CLL and RT, the study provides important insights into disease biology. Transgenic PRMT5 mouse models and pre-clinical treatment studies with a novel PRMT5 inhibitor provide evidence for an involvement of this methyltransferase in disease progression and suggest it as a novel target for therapy. In addition, the generation of a new mouse line that models RT and can be used for further preclinical analyses is of high value for follow-up studies.

The manuscript is clearly written and the data presentation and discussion mostly valid and comprehensible.

I have only a few minor comments to improve the display and description of data:

1. Lines 259-261: Is the difference in B220dim vs bright signal mainly due to the difference in size of the cells? Or do the authors claim a differential expression of this molecule in the two types of B cells?

- We thank the reviewer for identifying this opportunity to improve the clarity of our manuscript. We have adjusted the results section of the manuscript, adding Ref #41 (10.1158/1078-0432.CCR-19-0273) to support this claim:

“While the large cells in E μ -PRMT5/TCL1 mice maintained the CD19+/CD5+ ‘CLL-like’ phenotype, these cells were predominantly B220bright, a feature consistent with murine models of large cell lymphoma⁴¹, in contrast with B220dim expression observed on the ‘CLL-like’ small cells in both E μ -PRMT5/TCL1 and E μ -TCL1 mice.”

2. Line 283: There is no Figure 4G, this needs to be changed to 4C.

- We thank the reviewer for catching this mistake. It has been fixed:

“Similarly, lymph nodes from E μ -PRMT5/TCL1 mice were more severely affected, demonstrating hyperplasia and increased B-cell proliferation in lymphatic germinal centers at 3 months of age which progressed to atrophy by 6 months, as evident by scant regions of cortex with few germinal centers (Figure 4C dashed circles).”

3. Lines 291-294: “... the extent of the neoplastic infiltrates appeared more pronounced in E μ -PRMT5/TCL1 animals... “ This is not clearly visible in Figure 4E and should be shown in a better way.

- We thank the reviewer for this suggestion to improve our manuscript, and we agree our reporting of this data was misleading. This line referenced by the reviewer is a continuation of a previous idea that we did not sufficiently explain. At the time animals in the E μ -PRMT5/TCL1 and E μ -TCL1 animals meet removal criteria due to advanced disease, the observable phenotype by histopathology is largely similar. We have amended the text to reflect this:

“With aging, key clinical characteristics of E μ -PRMT5/TCL1 mice at ERC included pronounced cervical lymphadenopathy (Figure 4D) and palpable splenomegaly. Mandibular, mesenteric, and thoracic lymph nodes were markedly enlarged in E μ -PRMT5/TCL1 mice, a hallmark of human RT. By contrast, even mildly enlarged mandibular or other peripheral, thoracic and abdominal lymph nodes were rarely observed in E μ -TCL1 mice. In general, histopathology analysis of tissues from E μ -TCL1 and E μ -PRMT5/TCL1 mice reaching endpoint criteria demonstrated a similar histologic phenotype, observing intermediate sized lymphocytes with central nuclei containing stippled chromatin and scant basophilic cytoplasm

infiltrated the spleen, liver, and lung (Figure 4E). Notably, a partial or complete transformation to a diffuse large cell phenotype in the lymph node was observed in 2/5 analyzed Eμ-PRMT5/TCL1 mice, in contrast to the densely packed lymphocytes observed in the lymph nodes of Eμ-TCL1 mice. In involved lymph nodes and salivary glands, 3-dimensional structures were completely destroyed and not histologically recognizable.”

4. Lines 294-298: Where can one see what is stated in this paragraph? The display of these observations needs to be improved.

- We again thank the reviewer for suggesting we improve reporting of the data in Figure 4E. We have amended the text to this paragraph, displayed in our response to comment #3 shown above. We have also amended the figure to better describe what is stated in this paragraph, shown above.

5. Figure 5A: It would be helpful to add a principal component analysis of this data to see the clustering of the 4 different groups analysed here, instead of only showing a heatmap of the DEG.

- We thank the reviewer for this suggestion and agree this would improve the manuscript. We have added a principal component analysis of this data to Figure 5A, and adjusted the main text and figure legend as necessary. Acronyms for panel D have been added to the legend:

“To understand the downstream pathways contributing to a more aggressive disease phenotype in the Eμ-PRMT5/TCL1 model, we performed RNA-seq in B-cells isolated from spleens of Eμ-PRMT5/TCL1 and Eμ-TCL1 mice at 3 months and 6 months of age – prior to accumulation of a lethal circulating disease (Figure 5A, B).”

A) Bulk RNA-sequencing performed on leukemic cells from the spleen of Eμ-TCL1 mice at 3 months (n=2) and 6 months (n=2) of age compared with Eμ-PRMT5/TCL1 mice at 3 months (n=3) and 6 months (n=2) of age. Variance in transcription profile is visualized via principal component analysis plot. Distinct variance is observed in 6 month Eμ-PRMT5/TCL1 mice along the PC1 axis.

B) Bulk RNA-sequencing in cells from **A** displayed via heatmap. Notable differentially expressed genes (fold change cutoff of |2| and minimum FDR cutoff of 5%) observed in Eμ-PRMT5/TCL1 mice at 6 months of age compared to other groups are indicated. This comparison identified 2412 DEGs between 3 month and 6 month Eμ-PRMT5/TCL1 samples whereas no DEGs between 3 month and 6 month Eμ-TCL1 samples were identified. No DEGs were identified between Eμ-TCL1 and Eμ-PRMT5xTCL1 samples at 3 months. In contrast, a total of 2601 genes (498 upregulated and 2103 downregulated) were dysregulated in Eμ-PRMT5/TCL1 vs Eμ-TCL1 samples at 6 months.

C) Comparative gene expression analysis of spleen cells from Eμ-PRMT5/TCL1 mice at 6 months of age compared to all other groups by log2 fold change in RNA-seq (x-axis) and log 2-fold change in ATAC-seq (y-axis). Black dots represent individual genes. The plot represents genes exceeding a cutoff of log2 fold change in RNA-seq >|1|. Notable genes identified in (A) are highlighted by symbol.

D) Analysis with rMATS revealed an increase in frequency of aberrant splicing events between Eμ-PRMT5/TCL1 and Eμ-TCL1 cells at 6 months compared to 3 months. Alternative splicing events are dominated in cassette exons with higher rates of exon inclusion at 6 months compared to 3 months. Skipped exon (SE); Retained intron (RI); Mutually exclusive exons (MXE); Alternative 5' splice site (A5SS); Alternative 3' splice site (A3SS).

E) scRNA-seq analysis of spleen cells in Eμ-PRMT5/TCL1 (n=4) and Eμ-TCL1 (n=4) mice, visualized via UMAP and clustered according to K-means (n=10). Clustering of B (Cd19+), T (Cd3+, Cd4+, Cd8+) and myeloid (Cd11b+) cells as indicated.

F) Relative gene expression of CLL and RT-related genes in E μ -PRMT5/TCL1 and E μ -TCL1 spleen cells. Violin plots demonstrate the relative distribution of expression between analyzed cells.

G) Heatmaps highlighting the top 50 enriched and the top 50 depleted genes in cluster C6, visualizing expression relative to other B cell clusters.”

6. Lines 158 and 463: The authors describe a likely contribution of IL4I1 to CLL-to-RT progression via promoting immune suppression. The original article describing a role for IL4I1 in immune suppression in CLL needs to be cited here (Sadik et al., Cell 2020).

- We thank the reviewer for noting this omission. This has been added as reference #32:

“PRMT5-enriched cluster C5 also presented upregulated *BCL21A*, *IL4I1*, *CD83*, *TCL1A*, and *CCL3/4*, likely contributing to CLL-to-RT progression by stimulating anti-apoptotic signaling³⁰ and promoting immune suppression^{31,32} with enhanced activity along the BCR-signaling axis^{33,34} (Figure 1E).”

Reviewer #3 (Remarks to the Author); expert in clinical and preclinical CLL:

This paper by Hing et al described that mice model with overexpressing hPRMT5 developed a B-lymphoid expansion. The mixed double transgenic mice with Eu-PRMT5/TCL-1 developed a more aggressive lymphoid disease resembling Richter transformation of CLL. The authors further studied the oncogenic pathways regulated by this transgenic model. They also reported the development of a PRMT5 inhibitor, PRT382. They argued that further clinical test of PRMT5 inhibitor, e.g, PRT382 maybe potentially explored for clinical therapeutic trial for RT.

Even though this above finding of this mice model seems interesting, but many questions remain unclear.

1. In the introduction in Page 3, the author describe RT as the most common complication of CLL. This is not correct. RT is a disease evolution from CT, meaning disease evolved into more aggressive form of lymphoma. This needs to correct. In, addition, RT is chemotherapy-resistant, would not use therapy-resistant. This is because there are some experiences using variable targeted, immune checkpoint, as well CART therapy for RT.

- We thank the reviewer for the suggestion to improve the introduction of our manuscript. Accordingly, we have updated this paragraph of the introduction to better introduce this topic:

“Richter’s Transformation (RT) is the morphologic **evolution** from CLL/small lymphocytic lymphoma to a high grade diffuse large B-cell (DLBCL) or immunoblastic lymphoma, occurring in up to 10% of CLL patients. **RT is the most common disease progression in CLL patients within the first 18 months of receiving targeted therapies (i.e. ibrutinib, venetoclax), with overall incidence expected to rise as more patients receive these treatments in the front line and relapse/refractory setting^{1,2}.** Unlike de-novo DLBCL, generally chemotherapy-responsive and potentially curable, RT is **chemotherapy-resistant** and associated with a dreadful prognosis³.”

2. It seems rather weak to link PRMT5 with RT. The authors cited several references in 20-27 about the important role of PRMT5 in cancer or regulating MYC and Cyclin D1, however, lack specific reason or logic to focusing on this particular protein/gene PRMT5 in RT pathogenesis, making it less logic to study and test this protein.

- We thank the reviewer for this suggestion to clarify the rationale for this project. Accordingly, we have updated the paragraph of the introduction to better present the rationale for our study, adding ref #28 (*Schnormeier et al, 2020. Sci Rep*) to support this claim:

“Arginine methylation regulates critical cellular processes including proliferation and differentiation^{3,12-18}. The **type II** protein arginine methyltransferase 5 (PRMT5) mediates symmetric dimethylation of arginine residues on histones (i.e H3 and H4) and non-histone proteins [e.g. retinoblastoma (Rb), p53, and Sm proteins of the spliceosome] thereby influencing gene transcription, splicing¹⁹, and protein activity. PRMT5 is essential for the eIF4E-mediated 5'-cap-dependent translation of c-MYC and CYCLIN D1²⁰ and is required for the establishment and maintenance of c-Myc and NOTCH1 driven lymphomas^{21,22}. Moreover, co-deletion of MTAP and CDKN2A as a result of del(9)(p21.3), seen in up to 30% of RT tumors^{23,24}, has been shown to confer vulnerability to PRMT5-targeted therapies, **suggesting dependence on PRMT5 for proliferation of these tumors²⁵⁻²⁷. While literature strongly supports a requisite role for PRMT5 in the context of large cell lymphomas, emerging evidence now suggests aberrant PRMT5 expression additionally disrupts critical regulatory mechanisms and contributes to tumor progression in CLL²⁸. Accordingly, we then hypothesized mechanisms stemming from dysregulated PRMT5 activity to play a major role in facilitating CLL-to-RT evolution.**”

3. In Figure 1, PRMT5 expression were shown in CLL and pre-RT samples by western blot. However, it is unclear whether any of these samples were consecutive samples, meaning from classic CLL to pre-RT phase. If they are sequential samples, what is the time interval or therapy pattern between these samples?

Also, it will make sense to see the PRMT5 expression level with MYC rearrangement/protein expression relationship. This part of study need to provide more details.

- We thank the reviewer for this suggestion. We have updated the figure legend to make more clear these were not sequential samples, rather samples from 16 individual CLL patients.

“**B**) PRMT5 expression in CLL B-cells via western blot. Samples were collected **from 16 individual patients** during the CLL-phase and retrospectively identified as **those eventually** progressing (pre-RT) or not (CLL) to RT within one year from **the time of** collection.”

- We also agree the addition of MYC rearrangement relationship details in the patients studied would benefit figure 1. We have updated the results section of the study to reflect this:

“To evaluate the role for PRMT5 in RT we again conducted scRNA-seq in LN biopsies from two patients collected at the time of RT diagnosis (**Figure 1D**). **Notably, clinical processing of ‘RT Pt-2’ observed a rearrangement with the MYC locus by FISH.** Transcriptional heterogeneity was observed in B cells between samples; noting that PRMT5 was not abundantly expressed in all B cells, but was instead restricted to distinct subpopulations also displaying strong MYC expression. **Clustering of cells with strong PRMT5 and MYC expression was observed in both patients regardless of identifiable MYC rearrangement.**”

4. For figure 1C staining, it will be important to provide co-staining of PRMT5 with a B cell or T cell markers to know the relative expression in different lineages.

- We agree with the reviewer that additional staining for T cell marker would improve the evaluation of these tumors. Accordingly, we have amended the manuscript to include CD3 staining by tissue microarray in Suppl. Figure 1B. This analysis demonstrates intermittent distribution of CD3+ cells within the LN tumors of CLL patients, with CD3+ cells displaying a differential overall morphology compared to the PRMT5+ expressing cells. Unfortunately, we were not able to obtain further RT LN tissue for this analysis. Thus, images for CLL LN from PRMT5 low and PRMT5 weakly positive cases are presented.
- Visualized at 400x, we further observe small CD3+ cells in the LN of CLL tumors are drastically morphologically different compared to the strongly PRMT5+ large cells of RT LN tumors.

- Co-Author Claudio Agostinelli, world-renowned expert pathologist, provided this analysis of the tissue microarray data:
 - *Included are:*
 - *a photo of CD3 expression on small lymphocytes in the PRMT5 negative case (I would observe how you do not see small PRMT5⁻ lymphocytes in the typical CLL of photo 1C)*
 - *a photo of the CD3 in the weakly positive PRMT5 case, where expression is seen in small lymphocytes (cytologically different from the medium-sized PRMT5 elements in photo 1C)- unfortunately, we no longer have any material from the strongly PRMT5⁻ RT cases, however the cytology of the PRMT5⁻ elements clearly indicates that the positive cells are tumor blasts and not small lymphocytes from the microenvironment.*
- We have updated Suppl. Figure 1B, and amended the text to include this data:

“We next evaluated the PRMT5 protein expression in CLL cells from the blood of patients that eventually did (CLL pre-RT) or did not (CLL) transform to RT within one year from the time the sample was collected. PRMT5 expression was variable in CLL patients who did not undergo transformation, ranging from not expressed to weakly expressed, whereas PRMT5 was consistently overexpressed in pre-RT patients (**Figure 1B**). We then performed tissue microarray studies with lymph node biopsies from 70 CLL cases – classified by proliferation center (PC)-rich or typical-PC distribution^{29,30} – and 15 RT cases. PC-rich CLL cases were defined as those with confluent PCs whereas typical CLL cases showed scattered, small, ill-defined PCs in a monotonous background of small, relatively round lymphocytes (small PC). The small lymphocytes within typical CLL cases stained negative for PRMT5 expression, however, small PC areas showed weak PRMT5 expression (**Figure 1C** – pink stain). PC-rich CLL cases contained weak to moderate positivity for PRMT5 in prolymphocytes (PL) and paraimmunoblasts (PI). CLL LN tumors were intermittently populated with small CD3+ T cells with distinct cytologic morphology from small-to-medium sized tumor blasts with PRMT5 positivity (*Supplementary Figure 1B*). In stark contrast from CLL tumors, all 15 RT cases showed strong positivity for PRMT5 (**Figure 1C**). Further evident at 400x magnification, the strongly PRMT5+ medium sized tumor blasts in RT LN tumors demonstrate cytology explicitly divergent from CLL LN cases with limited penetrance of small lymphoid cells, indicating these tumors were largely comprised of PRMT5+ large lymphocytes and not small lymphocytes from the tumor microenvironment (*Supplementary Figure 1C*).”

Supplemental Figure 1:

“B) Representative CD3 staining by tissue microarray in lymph node biopsies from CLL LN cases negative or weakly positive for PRMT5. LN tumors in both cases are intermittently populated with CD3+ T cells with distinct cytologic morphology from small-to-medium sized tumor blasts with PRMT5 positivity. Visualized at 400x.”

5. In figure 2, given the documented mixed cellular population in PRMT5 mice (in 2D), it is important to show the specific B/T/Myeloid markers in these organs shown in 2G.

- We thank the reviewer for inviting us to expand the characterization of the E μ -PRMT5 mouse. We agree with this comment, and have conducted further IHC staining in E μ -PRMT5 mice to show B cells (B220+), T cells (CD3+), and myeloid markers (F4/80+ macrophages) in affected organs. We have updated Figure 2 to show representative staining in the E μ -PRMT5 model compared to WT mice, and have updated the results section and figure legend to reflect these changes:

“Upon euthanization of symptomatic mice, nodal and extranodal tissues were collected and examined. Histopathologic analysis demonstrated evidence of hematopoietic neoplasia in 4/11 E μ -PRMT5 mice (**Figure 2F**). Overall, inter-specimen variability in morphology of the neoplastic cells was observed, ranging from anaplastic large lymphocytes to small lymphocytes commonly invading the spleen, lymph nodes, lung, liver and kidney, although not all tissues were affected in every mouse. Mice numbers L31, L35, and L36 contained anaplastic large lymphocytes (**L**, appear blue-purple) and bland histiocyte infiltrates (**H**, light pink cytoplasm). Mouse L32 showed sheets of small lymphocytes (**S**) in the lymph nodes and spleen, consistent with a small lymphocytic lymphoma/chronic lymphocytic leukemia. **Immunohistochemistry in E μ -PRMT5 mice further demonstrated these tumors were largely composed of a background of large B220+ B-cells with scattered CD3+ T-cells and F4/80+ macrophages (Figure 2G).** Overall, this mixture of cell types is consistent with the diagnosis of histiocyte associated lymphoma, which is the murine counterpart of the human disease T-cell and Histiocyte-rich B-cell lymphoma.”

“Figure 2: Overexpression of PRMT5 induces lymphoid proliferation in-vivo

- A) Schematic representation of the construct used to generate the E μ -PRMT5 transgenic mouse. The entire human PRMT5 coding region was cloned into pBluescript II KS (pE μ) plasmid containing a mouse VH promoter (V186.2) and the IgH- μ enhancer along with the 3' untranslated region and the poly(A) site of the human β -globin gene. hPRMT5 gene transcripts were observed via RT-qPCR at detectable threshold cycles (Ct) in B cells of E μ -PRMT5 mice but not wildtype littermates. mTbp1 shown as control. Circles indicate individual mice. Bars indicate mean \pm SD.
- B) E μ -PRMT5 mice (n=67) and wildtype littermates (n=10) were followed for survival. Comparisons were visualized via Kaplan-Meier plots and statistical analysis was completed using the log rank (Mantel-Cox) test [p=0.0004, hazard ratio=0.12 (95% CI: 0.069 to 0.23)].
- C) Cause of death (COD) observations for E μ -PRMT5 transgenic mice (n=75). Categorical COD was recorded as either spontaneous death or meeting predefined euthanasia criteria. Unresolved cause of death cases were grouped as “other”.
- D) A cohort of E μ -PRMT5 (n=26), E μ -TCL1 (n=35), and wildtype littermate mice (n=9) were assessed monthly for spontaneous disease expansion by flow cytometry of peripheral blood. Expansive populations of T cells (CD45+/CD5+), B cells (CD19+/CD5+/B220^{bright}), CLL-like cells (CD19+/CD5+/B220^{dim}), or myeloid cells (FSC^{bright} & CD19-/CD5-) were observed in varying ratios within groups.
- E) Rate of CLL-like cell accumulation over time in E μ -PRMT5 mice. Dotted line indicates 10% CD5+/B220+ cells in the blood observed via flow cytometry.
- F) Histopathology analysis via hematoxylin and eosin (H&E) staining of E μ -PRMT5 mice (L31, L32, L35, L36) reaching ERC and representative age-matched wildtype littermate. L, anaplastic large lymphocytes; H, bland histiocytes; S, sheets of small lymphocytes. Scale bars as indicated.
- G) **Representative immunohistochemistry of tumors from E μ -PRMT5 (n=5) mice demonstrates tumors are largely composed of a background of large B220+ B-cells with scattered CD3+ T-cells and F4/80+ macrophages. Arrows indicate scattered CD3+ and F4/80+ cells. This mixture of cell types is consistent with the diagnosis of histiocyte associated lymphoma, which is the murine counterpart of the human disease T-cell and Histiocyte-rich B-cell lymphoma. B220 and CD3 shown in spleen, F4/80 shown in liver. Images shown at 40x magnification. Scale bar is 50 μ m.**
- H) B cell receptor variable gene usage in splenic CLL-like cells from E μ -PRMT5 (n=4) and E μ -TCL1 (n=1) mice evaluated via RNA-seq. BCR reads were binned by specific heavy (H) and light-chain V genes, and percentage of each were compared to the total reads to determine gene usage. Each color represents a unique BCR gene.
- I) Bulk RNA-seq in splenic CLL-like cells from E μ -PRMT5 and E μ -TCL1 mice from H. Volcano plot representing transcript abundance enrichment in E μ -PRMT5 (orange/green) vs E μ -TCL1 cells (blue/purple).”
- To accommodate for this addition, data in former Figure 2F has been moved to Supplemental Figure 2A, and the associated text and legend has been updated to reflect this change:

“Supplemental Figure 2: CLL-like phenotype in the Eμ-PRMT5 mouse model

A) Peripheral blood smears from Eμ-PRMT5 and Eμ-TCL1 mice. Scale bar is 60 μm.

B) Representative flow cytometry plots of spleen cells derived from Eμ-PRMT5 (n=5) and Eμ-TCL1 (n=3) mice at ERC. A CLL-like phenotype (CD5+/CD19+, CD5+/B220dim) is observed in Eμ-PRMT5 and Eμ-TCL1 mice, but not in WT mice.

C) CDR3 sequence abundances presented as pie-charts, reflecting clonal relation between the overall B cell populations of the spleen in Eμ-PRMT5 (n=4) and Eμ-TCL1 (n=1) mice at ERC. Top 20 most abundant CDR3 sequences are shown by mouse; shared sequences have the same color across all samples. Sequences not in the top 20 are grouped as “Others” (forest green sector).

D) Percentage of IgVH Mutations in the dominant clone of Eμ-PRMT5 (n=4) and Eμ-TCL1 (n=4) mice. The IgHV gene sequence of the dominant clone in each sample was mapped against its corresponding germline sequence. The total number of mutation events were divided by the total number of nucleotide reads to calculate the mutation percentage. IgVH mutation percentage below the 2% threshold is indicative of an ‘unmutated IgHV’ classification.

E) Visualization of variation in gene expression (Log2 normalized expression) of select leukemogenic genes in Eμ-PRMT5 (n=5) and Eμ-TCL1 (n=4) splenic B cells distributed by UMAP criteria. Violin plots illustrate gene expression of select leukemogenic genes in the B-cell populations of Eμ-PRMT5 and Eμ-TCL1 spleen. Dots represent expression level in individual cells.

F) Ingenuity Pathway Analysis (IPA) of the top upregulated genes in clusters C3, C4, and C6 from the spleens of Eμ-PRMT5 (n=5) and Eμ-TCL1 (n=4) mice. Color scale reflects the range of enrichment significance [-Log2(p-value)] determined via Fisher’s exact t-test.

G) Ingenuity Pathway Analysis (IPA) of the top depleted genes in clusters C4 and C6 from the spleens of Eμ-PRMT5 (n=5) and Eμ-TCL1 (n=4) mice. Color scale reflects the range of enrichment significance [-Log2(p-value)] determined via Fisher’s exact t-test.

6. PRMT5 mice seems to have more diverse clonal population compared to TCL1 mice, how will author explain these results in 2H?

- We thank the reviewer for this opportunity to expand the interpretation of our results. We have updated the results section to include this:

“We next performed RNA sequencing to evaluate the transcriptome and BCR gene usage³⁸ in CD5+CD19+ cells (*Supplemental Figure 2A*) isolated from the spleen of Eμ-PRMT5, Eμ-TCL1 and WT mice. Predominant usage of distinct kappa and heavy chain genes in each Eμ-PRMT5 and Eμ-TCL1 mouse, but not WT mice, was observed (**Figure 2H**, *Supplemental Figure 2B*), indicating an expansion of cells with minimal clonal diversity. **In concordance with the observed frequency of CLL-like development in Eμ-PRMT5 mice, aberrant PRMT5 expression as the sole transgenic event appeared as a less-robust murine leukemogenic driver as a greater degree of clonal diversity was observed in Eμ-PRMT5 mice when directly compared to the B cell compartment of Eμ-TCL1 mice.** A low mutational burden in the *IgHV* gene sequence (<2%) was observed in the dominant clones across all samples, consistent with classification as *unmutated IgHV* (*Supplemental Figure 2C*).”

7. Staining the specific B, T and myeloid markers will be critical to show the disease in this mouse model (figure 4C and 4E).

- We thank the reviewer for the opportunity to expand characterization of the Eμ-PRMT5/TCL1 mouse model. We agree further discussion of the relationship between B, T and myeloid cell population in this model would benefit the

manuscript. We have conducted further flow cytometry analysis of blood and spleen cells in Eu-PRMT5/TCL1 and Eu-TCL1 mice at 3 and 6 months of age, and added this to the manuscript as Supplemental Figure 3D. The text has been amended to reflect this addition.

- Similarly, Flow cytometry analysis of cells from the spleen in Eu-PRMT5/TCL1 and Eu-TCL1 mice meeting removal criteria due to advanced CLL is already present in the manuscript as Supplemental Figure 4A. These data show similar trends to what is observed in 6 month Eu-PRMT5/TCL1 mice with significant CLL-like disease, where the CLL-like cells outcompete CD5+ and CD5-/CD19- myeloid cells at this disease stage and comprise the majority of the resulting tumor.

“To interrogate the timeline of disease progression, we conducted interim analysis of 3 and 6-month old μ -PRMT5/TCL1 and μ -TCL1 mice. By flow cytometry analysis of the blood and spleen at 3 months of age both μ -PRMT5/TCL1 and μ -TCL1 mice were observed with Cd11b-/CD3-/CD19+ B cells comprising the majority of these compartments (*Supplemental Figure 3D*). Varying CD3+ T cell and CD11b+ myeloid populations were found in both models, with limited evidence of CD19+/CD5+ ‘CLL-like’ disease emerging at this age. By 6 months of age, prominent expansion of ‘CLL-like’ cells could be identified in both blood and spleen of μ -PRMT5/TCL1 mice but not in μ -TCL1 mice. The ‘CLL-like’ expansion appeared to completely dominate the blood and spleen compartment, outcompeting both T cell and myeloid populations. Similar T cell and myeloid populations were observed in μ -TCL1 mice at 6 months compared to 3 month μ -PRMT5/TCL1 and μ -TCL1 mice.

Similarly by histopathology analysis, at 3 months both $\text{E}\mu\text{-PRMT5/TCL1}$ and $\text{E}\mu\text{-TCL1}$ mice showed evidence of emerging germinal center hyperplasia of the spleen (Figure 4C dashed circles; and Supplemental Figure 3E). By 6 months of age, effacement of normal architecture including loss of germinal centers and indistinct splenic white pulp with rare foci of neoplastic lymphocytes encroaching upon splenic white pulp (termed 'lymphoid atrophy')⁴² was exclusively present in the spleens of $\text{E}\mu\text{-PRMT5/TCL1}$ mice (arrows), whereas $\text{E}\mu\text{-TCL1}$ mice revealed persistent hyperplasia in these regions (Supplemental Figure 3F). This finding was confirmed via Ki67 staining, which highlights the proliferative germinal centers (Figure 4C). These data suggested leukemic cells penetrating the spleen of the $\text{E}\mu\text{-PRMT5/TCL1}$ animals produced a more aggressive phenotype, replacing normal B and T lymphocytes with rapidly progressing leukemic cells."

"D) Representative flow cytometry analysis of the peripheral blood and spleen from $\text{E}\mu\text{-PRMT5/TCL1}$ mice at 3 months (n=4) and 6 months (n=4) of age compared with $\text{E}\mu\text{-TCL1}$ mice at 3 months (n=3) and 6 months (n=2) of age. Staining for markers including Cd11b and CD3 indicate myeloid and T cell populations. 'CLL-like' CD19+/CD5+ population is identified from [CD11b-/CD3-] population, and is more pronounced in $\text{E}\mu\text{-PRMT5/TCL1}$ mice at 6 months of age."

Supplemental Figure 4A, already present in the manuscript:

8. How would author explain the PRMT5/TCL-1 mice lymphoid atrophy developed in 6 months? How is this similar to human RT tumor? Careful histology comparison will be needed from pathology point of view.

- We thank the reviewer for this suggestion to improve our manuscript. The term 'lymphoid atrophy' in this manuscript is used to describe the advanced pathologic progression from germinal center hyperplasia to total effacement of normal tissue architecture with loss of identifiable germinal center regions, which is observed in the LN of human RT tumors. We understand the term atrophy is also used to describe necrotic, shrinking, and apoptotic tissues, and is confusing in this context. Thus we agree that the use of this term may be misleading to the reader, and we have removed it from the manuscript.

"Similarly by histopathology analysis, at 3 months both $\text{E}\mu\text{-PRMT5/TCL1}$ and $\text{E}\mu\text{-TCL1}$ mice showed evidence of emerging germinal center hyperplasia of the spleen (Figure 4C dashed circles; and Supplemental Figure 3E). By 6 months of age, effacement of normal architecture including loss of germinal centers and indistinct splenic white pulp with rare foci of neoplastic lymphocytes encroaching upon splenic white pulp was exclusively present in the spleens of $\text{E}\mu\text{-PRMT5/TCL1}$

mice (arrows), whereas E μ -TCL1 mice revealed persistent hyperplasia in these regions (*Supplemental Figure 3F*). This finding was confirmed via Ki67 staining, which highlights the proliferative germinal centers (Figure 4C). These data suggested leukemic cells penetrating the spleen of the E μ -PRMT5/TCL1 animals produced a more aggressive phenotype, replacing normal B and T lymphocytes with rapidly progressing leukemic cells.

Additionally, lymph nodes from E μ -PRMT5/TCL1 mice were more severely affected, demonstrating hyperplasia and increased B-cell proliferation in lymphatic germinal centers at 3 months of age which progressed to **effacement of lymphoid architecture** by 6 months, as evident by scant regions of cortex with few germinal centers (Figure 4C dashed circles). Hyperplastic involvement of lymph node germinal center tissue was infrequently observed in E μ -TCL1 mice at 3 months, and regularly observed in analyzed mice at 6 months of age.”

9. While the DEG analysis of 3 vs. 6 months of PRMT5/TCL-1 mice was informative, it is unclear how this resembles human RT transcriptome analysis or purely representing genes regulated by PRMT5/TCL-1 overexpression.

- We thank the reviewer for this suggestion to improve discussion of this topic in our manuscript. The authors believe interrogating the transcriptome at early time point in the PRMT5/TCL-1 mouse is important to understand progression to RT in early phase (pre-RT phase). Understanding early changes like those described in this manuscript can assist in early detection of evidence indicating CLL-to-RT is likely in patients. We have updated our discussion of this topic to better reflect this idea:

“A CLL-to-RT phenotype is rarely observed in the E μ -TCL1 mouse model even in presence of conditional B cell-specific TRP53-deficiency or with simultaneous co-expression of Myc⁵². Given PRMT5 is consistently overexpressed in RT tissues and in CLL eventually undergoing RT, we developed the E μ -PRMT5/TCL1 model to evaluate a role for PRMT5 in CLL progression to RT. Concurrent *PRMT5* and *TCL1* expression resulted in a more rapid disease onset with increased lymph node involvement and overt lymphomagenesis that appeared to recapitulate disease behavior seen in human RT, representing the first model to mimic large B-cell lymphomas that arise from CLL. Cells from the E μ -PRMT5/TCL1 model again featured a unique gene expression signature in concordance with RT tumors and the E μ -PRMT5 model, with pathway analysis underscoring activation of immune signaling, immunosuppressive activity, MAPK signaling, and enrichment for cell cycle checkpoint pathways. Upregulation of genes including *Ctla4*, *Pd-L1*, and *Il-10* suggest aberrant activation of PRMT5 in B lymphocytes predisposed to CLL may further manipulate the surrounding environment to facilitate tumorigenic outgrowth in secondary lymphoid niches.

Importantly, DEG analysis in E μ -PRMT5/TCL1 mice at 6 months of

age revealed dysregulation of genes including *Myc*, *Il-10*, *Sox4*, *Ltk*, *Bcl11b*, and *Runx3* compared to E μ -TCL1 mice at this age and both E μ -PRMT5/TCL1 and E μ -TCL1 mice at 3 months of age. Interrogating the differential transcriptome at this early time point, while the CLL-like disease in E μ -PRMT5/TCL1 mice is evolving through the pre-RT phase to morphologically resemble a large B cell lymphoma, provides insight to mechanisms driving this progression. Understanding the dysregulated gene signature in the pre-RT phase, like that identified herein, can assist in early detection of evidence indicating CLL-to-RT is likely in patients monitored in the clinic, and further characterization of this pre-RT phase gene signature should be prioritized to better define this phenomenon.”

10. What is EPZ15666?

- We thank the reviewer for inviting better description of the PRMT5 inhibitors we compare with PRT382, the compound established in our study. A more robust description of EPZ015666 can be found in our response to comment 2 from reviewer #1.

Miscellaneous Revisions:

1. A file titled 'Source Data' has been added to this submission including raw data used to generate plots and uncropped blots.
2. A section titled "Data availability" has been added to the manuscript, after the methods section but before the references.
3. Authors Fiona Brown¹, Porsha Smith¹, Hannah Phillips¹ and Min Wang¹⁰ were added to reflect contribution to this manuscript.
4. Figure 2A and Supplemental Figure 6B were updated to show individual points

0. Figure 3D, Cks1b was incorrectly listed twice. Fixed to list Ifitm3.
1. An acknowledgement to Amy Porter of the Michigan State University Investigative Histopathology Laboratory has been added to the manuscript.
2. Amended typing errors identified throughout the manuscript.

REVIEWER COMMENTS

Reviewer #1 (Remarks to the Author):

Concerns have been addressed

Reviewer #2 (Remarks to the Author):

All my comments were adequately addressed by the authors.

Reviewer #3 (Remarks to the Author):

The authors have almost addressed all of my concerns from the initial reviews.

The only one concern/suggestion I still have now and can be better addressed is to include more recent references including Blood (2021) 137 (20): 2800–2816, The human richter transcriptomic data in this paper appears to be very limited. Authors either can do more human RT transcriptomic study to increase the number of human RT or carefully review the recent data published for human RT transcriptome (likely more than the one paper mentioned above). The goal is to compare the human data with mouse data to make sure the mouse data is not deviated from human data for bias.

Reviewer #4 (Remarks to the Author):

Disclaimer

Reviewer is not an expert in immunology or the etiology of the disease. As such, this review pertains mostly to the sequencing experiments described in the manuscript.

Major points:

About RNA-seq:

- We recommend the authors to use a single differential expression package for all of their RNA-seq analysis. Specifically, we recommend they replace their limma-based analyses with DESeq2, as the latter makes use of the statistical properties of count data.
- In L223-L225, authors conclude minimal differential expression between Eμ-PRMT5 and Eμ-TCL1 tumours. However, there is only 1 Eμ-TCL1 sample without replicates, which is problematic for differential expression analysis (as can be seen in figure 2I, by the large number of genes with adjusted p-values near 1). This highly suggests that the authors are not in a position to detect most differentially expressed genes, which prohibits the conclusion that there is minimal differential expression. We'd recommend to redo the experiment with replicates, or omit this analysis from the manuscript altogether.

About ATAC-seq:

- It is a missed opportunity to not look into the sequence motifs of differentially accessible regions and how they relate to transcription factors. The correlation between DEGs and chromatin are somewhat suggestive that a pioneer factor might be orchestrating differential expression. Since the authors appear to be familiar with Bioconductor packages, we suggest that they try a package like chromVAR to investigate motifs.

About scRNA-seq:

- Because cells in UMAPs tend to partially overlap, it is hard to judge how many cells have a property that is described in the text. For differential abundance conclusions, it would be good to include a table or plot wherein it is easy to compare cell numbers/proportions by sample/cluster.
- The manuscript describes at least 5 scRNAseq comparisons, all with their own clusters. However, some clusters share names between comparisons (e.g. 'cluster 1' appears in every one), which makes it slightly harder to unambiguously refer to a particular cluster (perhaps exemplified best in

L406-L409). It would be a good idea to uniquely name each cluster (or even better, harmonise cluster assignment between related samples). To illustrate the confusion, cluster 1 in figure 3A are indicated as Cd8+ T cells, whereas cluster 1 in figure 5E are Cd79a+ B cells. Both these figures use the TCL1 spleen samples, so the same cell may be in two clusters depending on the context.

- The methods section seems unclear on the following, but from the figure 1 description it appears as if the authors have used K-means clustering on the two-dimensional UMAP coordinates to segregate cells into clusters. In the methods section, it is stated that graph-based clustering on PC1-10 is performed. The graph based clustering on PC1-10 should be preferred over the K-means clustering on UMAP coordinates, due to UMAP distorting distances between cells. In addition, the authors should mention both the graph algorithm (e.g., KNN-graph) and graph clustering algorithm, such as the Louvain or Leiden algorithm (the latter reported with parameters). This applies to other scRNAseq experiments as well.
- In the methods, the following is written: “Cell-to-cell variation in gene expression driven by the number of detected molecules and mitochondrial gene expression was regressed out using linear regression”. The authors should either provide a reference for this approach, report the function + arguments + package(s) used, or motivate this step themselves, as this seems a bit of an atypical step in the analysis for scRNAseq data.
- When making claims about the significance of genes, such as in L376, the methods should include what testing procedure was used. More generally, null hypothesis testing for differential gene expression in single cell data can be problematic due to the clusters having been defined on the exact data that is used for testing.

About 4C:

- The chosen restriction enzyme strategy for 4C induces what are called ‘blind fragments’ [Van de Werken 2012, <https://doi.org/10.1038/nmeth.2173>], which are unreliable. We suggest either omitting the 4C part from the manuscript or repeating the experiment with Mbol/DpnII as a first cutter and Csp6I as a second cutter. Furthermore, to draw any meaningful conclusions template replicates should be included.

Minor points:

About RNA-seq:

- The manuscript and GEO records are currently unclear which samples are considered low-input RNA-seq and regular RNA-seq. Consequently, it is unknown which part of the methods applies to which part of the figures. The authors should clarify this.
- In figure 5A, it is recommended to display the variance explained of the first two PCs as a percentage (next to the axis titles).
- In figure 5B, indicate what the colours encode (likely fold-change or row-wise z-score).

About ATAC-seq:

- The methods describe mapping the data to the human genome, whereas the data appear to be from mice (based on text and GEO records). This needs to be corrected.
- In figure 5C it is unclear from either the legend description or the methods what association rule was used to link genes to accessible regions.
- In figure 5C, it is suggested in the legend description that every dot is an individual gene, whereas Sox4 appears to be displayed twice, hinting that perhaps every dot is a peak-gene link instead of a gene.

About scRNA-seq:

- It is generally hard to follow when two UMAP plots share the same coordinates. For example, in figure 1A, it appears as if the data share coordinates due to their overlap, whereas this is much clearer in figure 1D, where the overlap between Pt1 and Pt2 appears smaller. One way this might be remedied is that when presenting UMAPs faceted by sample/subset, it might be a good idea to show cells that are not members of the subset in muted/dimmed colours/grey in the background. Otherwise, it is visually hard to judge whether populations overlap between samples.
- In figure 1A, it is hard to judge whether the increase in PRMT5 is driven primarily by cluster 4 cells or is additionally driven by cluster 1.
- The dotplot in figure 1A might better be replaced with a labelled scatterplot, as it is easier to read xy-coordinates than two separate size and colour scales.

- In figure 1D, please report effect sizes for PRMT5, MYC and MKI67 as for example (log) fold changes.
- In figure 1E, clusters 3, 4, and 9 are missing and it is unclear from the description or the methods how this enrichment was calculated. We suggest to display all the clusters, but highlight the relevant ones.
- In figure 3A/B/C, it appears as if the aspect ratio of UMAP coordinates is distorted. It is generally recommended to let one unit of dimension 1 be of equal length of one unit of dimension 2.
- In figure 3B, it is unclear how the CD18+/CD5+ cells were identified, or to be more precise, which thresholds were used.
- In figure 3C, the legend description mentions 'relative gene expression', whereas the colour bar indicates absolute (normalized and log-transformed) gene expression.
- In figure 3C, please report effect sizes for the displayed genes.
- In figure 3D, the same comments as for figure 1E apply, but clusters 1, 5, 7, 9 and 10 are missing.
- In figure 3D, the colour bar for the middle heatmap is missing a minus sign. Moreover, it would probably be more intuitive to have the values in left-to-right order go from negative to positive instead of from positive to negative, in all the heatmaps.
- In figure 5F, please report effect sizes for the displayed genes.
- In figure 5G, the same comments as for figure 1E apply, but clusters 2, 4, 7 and 10 are missing.

Below, we have addressed all reviewer remarks in a point-by-point fashion:

REVIEWER COMMENTS

Reviewer #1 (Remarks to the Author); expert in PRMT inhibitors:

Concerns have been addressed.

- We thank this reviewer for carefully reviewing our manuscript and the overall positive remarks.

Reviewer #2 (Remarks to the Author); expert in scRNAseq, ATAC-seq, 4C and CLL:

All my comments were adequately addressed by the authors.

- We thank this reviewer for carefully reviewing our manuscript and the overall positive remarks.

Reviewer #3 (Remarks to the Author); expert in clinical and preclinical CLL:

The authors have almost addressed all of my concerns from the initial reviews.

The only one concern/suggestion I still have now and can be better addressed is to include more recent references including Blood (2021) 137 (20): 2800–2816, The human richter transcriptomic data in this paper appears to be very limited. Authors either can do more human RT transcriptomic study to increase the number of human RT or carefully review the recent data published for human RT transcriptome (likely more than the one paper mentioned above). The goal is to compare the human data with mouse data to make sure the mouse data is not deviated from human data for bias.

- We thank the reviewer for the additional suggestion to expand comparison of our human RT transcriptomic data with others currently available in the literature, and we agree this addition would strengthen our study. As such, we have obtained a human RT and CLL RNA-seq dataset available from a recent study by Nadeu et al (doi.org/10.1038/s41591-022-01927-8), and plotted the aggregate gene expression signature from CLL and RT patient tumors against the human samples analyzed in our study. Briefly, genes in Supplemental Table 11b from Nadeu et al with $\text{padj} < 0.05$ and direction “Up” were identified as “RT genes” and those with $\text{padj} < 0.05$ and direction “Down” were identified as “CLL genes”. Normalized per-cell UMI counts for each of these gene sets were summed to generate the aggregate gene expression score (Ref: Trapnell et al 2014. *Nature biotechnology*). In this analysis, we observed the CLL aggregate gene signature from Nadeu et al was enriched in CLL PBMCs and a subset of RT PBMCs from patients in our study. Importantly, the RT aggregate gene signature from Nadeu et al was enriched in RT LN cells in our study. This analysis demonstrates that, while limited by small sample size, the RT scRNA-seq data generated by our group is consistent with RT transcriptomic data in the literature and the results discussed further in this manuscript may be broadly applicable to further studies evaluating RT. We have added this data to our manuscript as Figure 1E and amended the associated text in the results.

- As the reviewer suggested, we have also reviewed the RT transcriptomic data in Blood (2021) 137 (20): 2800–2816 and compared to our data generated here. We have amended the discussion to include further comparison of our data with this study, further complementing our own RT transcriptomic data and that discussed in the Nadeu et al study.
- The reviewer also suggested we make further effort to compare human data with the mouse data in our study. Despite inherent limitations of using mouse models to study human disease, we observed considerable overlap in dysregulated genes in the E μ -PRMT5 mouse models and human RT transcriptomic data both here and in recent RT transcriptomic studies. We have accordingly amended the results sections to provide more focus on this comparison, and have expanded the discussion to include further comparison between human and mouse data. Overall, the comparative evaluation of E μ -PRMT5 and E μ -PRMT5/TCL1 mouse tumors to human RT lymph nodes from our limited cohort and the recently described human RT cohorts (Nadeu et al and Klintman et al) supports the use of the E μ -PRMT5/TCL1 mouse as a promising laboratory model of Richter's Transformation and warrants further evaluation with additional current and ongoing studies in RT.

Reviewer #4 (Remarks to the Author):

Major points:

About RNA-seq:

We recommend the authors to use a single differential expression package for all of their RNA-seq analysis. Specifically, we recommend they replace their limma-based analyses with DESeq2, as the latter makes use of the statistical properties of count data.

- We agree with this comment referring to the RNA-seq analysis presented in Figure 5A & 5B that was conducted using a limma-based package. As the reviewer suggested, we replaced data with analysis using DESeq2. The updated PCA plot associated with this analysis can be found in Figure 5A, and the updated heatmap of associated DEGs can be found in Figure 5B.
- The methods have been adjusted to reflect this change.

“RNA-seq, featured in Figure 5A, B: Mouse B cells were collected using the EasySep™ mouse pan B cell isolation kit (STEMCELL Technologies). Total RNA was isolated from TRIzol suspensions using a chloroform/ethanol extraction method and quantified via Qubit RNA HS Assay kit (Invitrogen). Total RNA was quantified using the Nanodrop 2000 Spectrophotometer (Thermo Fisher Scientific). Sequencing libraries were prepared using Illumina TruSeq stranded mRNA library preparation kit and sequenced on HiSeq 4000 instrument targeting approximately 60-80x10⁶ read pairs per sample. The 150 bp paired-end reads were trimmed for adaptors and low-quality bases with Cutadapt and mapped to the mouse reference genome GRCm38 with STAR aligner v2.6.0c⁶⁶. Then, featureCount function⁶⁷ from the subread package was used to generate gene counts based on Ensembl gene annotation release 95⁶⁸. Pairwise comparisons were performed to identify DEGs using DESeq2 R package (v1.20.0)⁶⁹.”

In L223-L225, authors conclude minimal differential expression between Eμ-PRMT5 and Eμ-TCL1 tumours. However, there is only 1 Eμ-TCL1 sample without replicates, which is problematic for differential expression analysis (as can be seen in figure 2I, by the large number of genes with adjusted p-values near 1). This highly suggests that the authors are not in a position to detect most differentially expressed genes, which prohibits the conclusion that there is minimal differential expression. We’d recommend to redo the experiment with replicates, or omit this analysis from the manuscript altogether.

- The reviewer brings up a great point. As suggested, we are now showing data derived from 4 Eμ-PRMT5 and 3 Eμ-TCL1 samples. Visualization of this experiment is presented in Figure 2I, and the text has been updated to reflect discussion of DEGs from this experiment that is more appropriately powered.

“Gene expression analysis showed minimal differentially expressed genes (DEGs; n=288 in Eμ-PRMT5, n=222 in Eμ-TCL1) between Eμ-PRMT5 and Eμ-TCL1 splenic B cells, suggesting a similar transcriptome profile within CLL-like cells in these two models (Figure 2I). Among these few DEGs we observed upregulation of *Arg2*, *Cxcl16*, *Elane*, *Il1r2*, and *Zap70* in the Eμ-PRMT5 tumors, and enrichment of *Plk2* and *Fos/Jun* family members in Eμ-TCL1 tumors. Transcriptional silencing of *PLK2* is a frequent event in B-cell malignancies³⁶, and associated with increased cell proliferation and decreased apoptosis in B cell lymphoma.”

About ATAC-seq:

It is a missed opportunity to not look into the sequence motifs of differentially accessible regions and how they relate to transcription factors. The correlation between DEGs and chromatin are somewhat suggestive that a pioneer factor might be orchestrating differential expression. Since the authors appear to be familiar with Bioconductor packages, we suggest that they try a package like chromVAR to investigate motifs.

- We agree with the reviewer and evaluation of motifs and transcription factor footprinting is part of larger ongoing work but outside the scope of this current manuscript.

About scRNA-seq:

Because cells in UMAPs tend to partially overlap, it is hard to judge how many cells have a property that is described in the text. For differential abundance conclusions, it would be good to include a table or plot wherein it is easy to compare cell numbers/proportions by sample/cluster.

- We thank the reviewer for this suggestion and agree this would enhance the clarity of our manuscript. As suggested, we have added a UMAP plot demonstrating cell density in UMAP space to figures in the manuscript in which we discuss scRNA-seq data, including Figure 1A, Figure 1D, Figure 3A, Figure 5E, and Sup. Figure 5A.

The manuscript describes at least 5 scRNAseq comparisons, all with their own clusters. However, some clusters share names between comparisons (e.g. 'cluster 1' appears in every one), which makes it slightly harder to unambiguously refer to a particular cluster (perhaps exemplified best in L406-L409). It would be a good idea to uniquely name each cluster (or even better, harmonise cluster assignment between related samples). To illustrate the confusion, cluster 1 in figure 3A are indicated as Cd8+ T cells, whereas cluster 1 in figure 5E are Cd79a+ B cells. Both these figures use the TCL1 spleen samples, so the same cell may be in two clusters depending on the context.

- We thank the reviewer for this suggestion and agree this may bring confusion to the reader. We have amended the manuscript to include unique names for each cluster discussed. Clusters from scRNA-seq with human data now have names RT.LN.#, and clusters from scRNA-seq with mouse data now have names according to figure they are presented (i.e. cluster 3.1, 5.1 etc).

The methods section seems unclear on the following, but from the figure 1 description it appears as if the authors have used K-means clustering on the two-dimensional UMAP coordinates to segregate cells into clusters. In the methods section, it is stated that graph-based clustering on PC1-10 is performed. The graph based clustering on PC1-10 should be

preferred over the K-means clustering on UMAP coordinates, due to UMAP distorting distances between cells. In addition, the authors should mention both the graph algorithm (e.g., KNN-graph) and graph clustering algorithm, such as the Louvain or Leiden algorithm (the latter reported with parameters). This applies to other scRNAseq experiments as well.

- We thank the reviewer for this suggestion. We have corrected the methods and included information on the graph and cluster algorithms.

“UMAP dimensionality reduction, clustering, and top marker analysis was achieved using functions from *Monocle3*^{81,82}. Leiden clustering⁸³ and k-Means10 clustering was utilized for human and murine analyses, respectively.”

In the methods, the following is written: “Cell-to-cell variation in gene expression driven by the number of detected molecules and mitochondrial gene expression was regressed out using linear regression”. The authors should either provide a reference for this approach, report the function + arguments + package(s) used, or motivate this step themselves, as this seems a bit of an atypical step in the analysis for scRNAseq data.

- We thank the reviewer for their suggestion and have adjusted the text to accurately reflect our analysis.

“Quality control removed low quality cells with a high percentage of reads mapped to mitochondrial genes (mito counts >2 median absolute deviations above the median) and/or low numbers of genes detected (feature counts >2 median absolute deviations below the median) using functions from *blaserTools* (<https://github.com/blaserlab/blaseRtools>).”

When making claims about the significance of genes, such as in L376, the methods should include what testing procedure was used. More generally, null hypothesis testing for differential gene expression in single cell data can be problematic due to the clusters having been defined on the exact data that is used for testing.

- We thank the reviewer for this comment and agree with the reviewer. We have changed the wording of text in L376 and other similar lines of text elsewhere.

About 4C:

The chosen restriction enzyme strategy for 4C induces what are called ‘blind fragments’ [Van de Werken 2012, <https://doi.org/10.1038/nmeth.2173>], which are unreliable. We suggest either omitting the 4C part from the manuscript or repeating the experiment with MboI/DpnII as a first cutter and Csp6I as a second cutter. Furthermore, to draw any meaningful conclusions template replicates should be included.

- We thank the reviewer for bringing this point to our attention. We performed a new experiment using DpnII as the first cutter and Csp6I as the second cutter as the reviewer suggested. We have replaced the circos plot in Sup. Figure 7A and the associated peaks plot demonstrating overlap between CHIP and 4C in Sup. Figure 7B to reflect this change. Nevertheless, the new data show similar conclusions as for the previous analysis.

Minor points:

About RNA-seq:

The manuscript and GEO records are currently unclear which samples are considered low-input RNA-seq and regular RNA-seq. Consequently, it is unknown which part of the methods applies to which part of the figures. The authors should clarify this.

- We apologize for the lack of clarity in our methods. We agree with this comment and have subsequently updated the methods and GEO records to enhance the distinction between these two experiments.

“RNA extraction, RNA-seq and Low-RNA-seq

RNA-seq, featured in Figure 5A, B: Mouse B cells were collected using the EasySep™ mouse pan B cell isolation kit (STEMCELL Technologies). Total RNA was isolated from TRIzol suspensions using a chloroform/ethanol extraction method and quantified via Qubit RNA HS Assay kit (Invitrogen). Total RNA was quantified using the Nanodrop 2000 Spectrophotometer (Thermo Fisher Scientific). Sequencing libraries were prepared using Illumina TruSeq stranded mRNA library preparation kit and sequenced on HiSeq 4000 instrument targeting approximately 60-

80x10⁶ read pairs per sample. The 150 bp paired-end reads were trimmed for adaptors and low-quality bases with Cutadapt and mapped to the mouse reference genome GRCm38 with STAR aligner v2.6.0c⁶⁶. Then, featureCount function⁶⁷ from the subread package was used to generate gene counts based on Ensembl gene annotation release 95⁶⁸. Pairwise comparisons were performed to identify DEGs using DESeq2 R package (v1.20.0)⁶⁹.

Low-input RNA-seq, featured in Figure 2H, Sup. Figure 2C, D, and Sup. Figure 4B-D: B-cells were isolated via EasySep Mouse Pan-B Cell Isolation Kit (Stemcell Technologies) ...”

In figure 5A, it is recommended to display the variance explained of the first two PCs as a percentage (next to the axis titles).

- We agree with the reviewer. We have calculated the percent variance for the 6-month Eu-PRMT5/TCL1 group and have added this information to the axis titles of figure 5A, and accordingly adjusted the figure legend.

“A) Bulk RNA-sequencing performed on leukemic cells from the spleen of Eμ-TCL1 mice at 3 months (n=2) and 6 months (n=2) of age compared with Eμ-PRMT5/TCL1 mice at 3 months (n=3) and 6 months (n=2) of age. Variance in transcription profile is visualized via principal component analysis plot. **Variance of 44% along the PC1 axis was observed in 6 months Eμ-PRMT5/TCL1 mice compared to all other groups.**”

In figure 5B, indicate what the colours encode (likely fold-change or row-wise z-score).

- The reviewer is correct, the colors encode z-score. We have added descriptive terms to the scale bar in Figure 5B and in the associated figure legend.

“**B)** DEG analysis from bulk RNA-sequencing in cells from A displayed via heatmap. Notable differentially expressed genes (fold change cutoff of $|1|$ and minimum FDR cutoff of 5%) observed in E μ -PRMT5/TCL1 mice at 6 months of age compared to other groups are indicated. **A total of 2167 genes (807 upregulated and 1360 downregulated) were dysregulated in E μ -PRMT5/TCL1 vs all other groups at 6 months. Scale bar indicates z-score.**”

About ATAC-seq:

The methods describe mapping the data to the human genome, whereas the data appear to be from mice (based on text and GEO records). This needs to be corrected.

- We apologize for this error. The reviewer is correct, the ATAC-seq data was derived from mice. Our methods section incorrectly described mapping to the human genome. We have updated the methods section to more accurately describe this experiment.

“ATAC-seq and data processing:

Accessible chromatin mapping was performed using the ATAC-seq method as previously described^{71,72}. In each experiment, 1×10^5 cells were washed once in 50 μ l PBS, resuspended in 50 μ l ATAC-seq lysis buffer (10 mM Tris-HCl pH 7.4, 10 mM NaCl, 3 mM MgCl₂ and 0.1% IGEPAL CA-630) and centrifuged for 10 minutes at 4 °C. On centrifugation, the pellet was washed briefly in 50 μ l MgCl₂ buffer (10 mM Tris pH 8.0 and 5 mM MgCl₂) before incubating in the transposase reaction mix (12.5 μ l 2 \times TD buffer, 2 μ l transposase (Illumina) and 10.5 μ l nuclease-

free water) for 30 min at 37 °C. After DNA purification with the MinElute kit, 1 µl of the eluted DNA was used in a qPCR reaction to estimate the optimum number of amplification cycles. Library amplification was followed by SPRI size selection to exclude fragments larger than 1,200 bp. DNA concentration was measured with a Qubit fluorometer (Life Technologies). Library amplification was performed using custom Nextera primers. **The libraries were sequenced by the Genomics Services Laboratory at Nationwide Children’s Hospital using an Illumina HiSeq 4000 platform. The 150 bp paired-end reads were trimmed for adaptors and low-quality bases with Cutadapt and mapped to the mouse reference genome GRCm38 using the Bowtie2 algorithm⁷³ with maximum fragment length of 1000 and no-mixed, no-discordant and very-sensitive options. After removal of duplicate reads, deepTools⁷⁴ were used to generate normalized genome coverage files. ATAC-seq peak locations were determined using the MACS2 algorithm (v2.1.1.)⁷⁵ with a cutoff p-value of 1E-5 with “--broad --nomodel --shift -100 --extsize 200” options. Consensus peak regions from all samples were generated as the union of the overlapping peaks using R GenomicRanges package⁷⁶, then annotated using R ChIPpeakAnno package⁷⁷.**”

In figure 5C it is unclear from either the legend description or the methods what association rule was used to link genes to accessible regions.

- We apologize for omitting this information. We have updated the methods section to include this information.

“ATAC-seq and data processing:

[...] The libraries were sequenced by the Genomics Services Laboratory at Nationwide Children’s Hospital using an Illumina HiSeq 4000 platform. The 150 bp paired-end reads were trimmed for adaptors and low-quality bases with Cutadapt and mapped to the mouse reference genome GRCm38 using the Bowtie2 algorithm⁷³ with maximum fragment length of 1000 and no-mixed, no-discordant and very-sensitive options. After removal of duplicate reads, deepTools⁷⁴ were used to generate normalized genome coverage files. ATAC-seq peak locations were determined using the MACS2 algorithm (v2.1.1.)⁷⁵ with a cutoff p-value of 1E-5 with “--broad --nomodel --shift -100 --extsize 200” options. Consensus peak regions from all samples were generated as the union of the overlapping peaks using R GenomicRanges package⁷⁶, then annotated using R ChIPpeakAnno package⁷⁷. **Using annotatePeakInBatch function from ChIPpeakAnno package each ATAC peak was assigned to closest/overlapping gene within 5000bp.**”

In figure 5C, it is suggested in the legend description that every dot is an individual gene, whereas Sox4 appears to be displayed twice, hinting that perhaps every dot is a peak-gene link instead of a gene.

- Thanks for bringing this to our attention. As the reviewer pointed out each point in the plot in Figure 5C is indeed a gene-peak pair. The figure legend has been updated to correctly describe this plot.

“C) Comparative gene expression analysis of spleen cells from Eµ-PRMT5/TCL1 mice at 6 months of age compared to all other groups (from A) by log2FC in RNA-seq (x-axis) and log2FC in ATAC-seq (y-axis). Points represent gene-peak pairs. The plot represents genes exceeding a cutoff of log2 fold change in RNA-seq >|1|. Notable genes identified in B are highlighted by symbol.”

About scRNA-seq:

It is generally hard to follow when two UMAP plots share the same coordinates. For example, in figure 1A, it appears as if the data share coordinates due to their overlap, whereas this is much clearer in figure 1D, where the overlap between Pt1 and Pt2 appears smaller. One way this might be remedied is that when presenting UMAPs faceted by sample/subset, it might be a good idea to show cells that are not members of the subset in muted/dimmed colours/grey in the background. Otherwise, it is visually hard to judge whether populations overlap between samples.

- We thank the reviewer for this suggestion and apologize for the lack of clarity. We have adjusted the figures to display the coordinates for each UMAP plot individually. Additionally, the overlap in the cells contributed by patient 1 and patient 2 can be seen in the new supplemental figure 1E.

In figure 1A, it is hard to judge whether the increase in *PRMT5* is driven primarily by cluster 4 cells or is additionally driven by cluster 1.

- We thank the reviewer for this observation. We have updated figure 1A and added Figure 1G & supplemental Figure 1F to further demonstrate that *PRMT5* expression is present across multiple clusters. In Figure 1G, leiden based clustering was implemented and, when paired with figure 1A and supplemental figure 1F, we see that several B cell clusters display varying levels of *PRMT5* expression (clusters 3, 11, 6, 5, and 1). Notably several of these *PRMT5* expressing clusters are absent or minimally present in the CLL sample compared to the RT samples. A key takeaway from Figure 1A is that upon Richter Transformation this patient developed an expansion of cells with high density in UMAP space that presented with elevated *PRMT5* expression. Further evaluation of the gene signature of these clusters is explored in Figure 1 and supplemental figure 1.

The dotplot in figure 1A might better be replaced with a labelled scatterplot, as it is easier to read xy-coordinates than two separate size and colour scales.

- We thank the reviewer for this suggestion. The authors have chosen to remove this plot from the figure.

In figure 1D, please report effect sizes for PRMT5, MYC and MKI67 as for example (log) fold changes.

- We thank the reviewer for this suggestion, however the purpose of Figure 1D is to show that both RT tumors express the listed genes (*PRMT5*, *MYC*, *MKI67*). Further data is reported in Supplemental Figure 1F demonstrating some analyzed cells have co-expression of all three genes (*PRMT5*, *MYC*, *MKI67*). No distinct comparison between groups is being made in this figure, and thus reporting fold changes for effect size is not the desired nor appropriate takeaway the authors intend the reader to gain.

In figure 1E, clusters 3, 4, and 9 are missing and it is unclear from the description or the methods how this enrichment was calculated. We suggest to display all the clusters, but highlight the relevant ones.

- We thank the reviewer for this suggestion and agree these figures could have been unclear to the reader. We have updated the UMAP clustering, now in Figure 1F, to more clearly displaying all of the leiden-based clustering in the combined RT LN samples. From these, we were interested in further evaluating gene expression in B cell clusters (B cells defined in Supplemental figure 1B), and thus deemed clusters 9, 1, 5, 6, 2, 3, and 11 to be relevant in this study and shown in the heatmap in Figure 1G. These clusters were deemed relevant as we were interested in the differential expression of genes within B cell populations, rather than differential expression of genes in RT B cells when compared to other T and monocyte cells in the LN tissue. The results section has been updated to better describe our intent.

In figure 3A/B/C, it appears as if the aspect ratio of UMAP coordinates is distorted. It is generally recommended to let one unit of dimension 1 be of equal length of one unit of dimension 2.

- We thank the reviewer for this suggestion. We have adjusted the figures accordingly.

In figure 3B, it is unclear how the CD18+/CD5+ cells were identified, or to be more precise, which thresholds were used.

- We thank the reviewer for this suggestion. We have updated the figure legend to more clearly describe this figure.

“‘CLL-like’ cells with co-expression of *Cd19* and *Cd5* (UMI counts > 0 for both) identified in the spleen of both E μ -PRMT5 and E μ -TCL1 (mice from A), localized to specific cell clusters.”

In figure 3C, the legend description mentions ‘relative gene expression’, whereas the colour bar indicates absolute (normalized and log-transformed) gene expression.

- We thank the reviewer for this comment. We have updated the legend to more clearly describe the data in Figure 3C.

“C) Relative gene expression of B cell maturation and leukemogenic markers in the spleen cells of E μ -PRMT5 and E μ -TCL1 mice (from A). Gene expression is shown as the Log10-transformed expression where expression is size factor normalized UMI counts.”

In figure 3C, please report effect sizes for the displayed genes.

- Similar to Figure 1D, the purpose of Figure 3C is to describe the B cells comprising tumors derived from E μ -PRMT5 and E μ -TCL1 mice by demonstrating markers of B cell maturation and phenotype in relation to their UMAP space. No direct comparison of gene expression between models is being made, and thus reporting fold changes for effect size is not the desired nor appropriate takeaway the authors intend the reader to gain.

In figure 3D, the same comments as for figure 1E apply, but clusters 1, 5, 7, 9 and 10 are missing.

- Similar to our response to comments regarding Figure 1E, we have updated Figure 3D and 3E. Figure 3D now more clearly displaying all of the clustering in spleen tissue from E μ -PRMT5 and E μ -TCL1 mice. From these, we were interested in further evaluating gene expression in B cell clusters (B cells defined in Supplemental figure 2E), and thus deemed clusters 3.2, 3.3, 3.4, 3.6, and 3.8 to be relevant in this study and shown in the heatmap in Figure 3E. These clusters were deemed relevant as we were interested in the differential expression of genes within B cell populations, rather than differential expression of genes in E μ -PRMT5 and E μ -TCL1 mice B cells when compared to other T and monocyte cells in the spleen tissue. The results section has been updated to better describe our intent.

In figure 3D, the colour bar for the middle heatmap is missing a minus sign. Moreover, it would probably more intuitive to have the values in left-to-right order go from negative to positive instead of from positive to negative, in all the heatmaps.

- The display format of the heatmap in Figure 3D has been updated in this version of the manuscript.

In figure 5F, please report effect sizes for the displayed genes.

- Now in Figure 5H, we are indeed directly comparing gene expression between E μ -PRMT5/TCL1 and E μ -TCL1 B cell clusters. As the author suggested, effect size has been added to the figure for each respective gene. Similar data is also presented in Supplemental Figure 2F, S4G, and S5D, for which effect sizes have also been added.
- Formerly Fig 5F, now Fig 5H

- Supp 2F

- Supp 4G

G

Supp 5D

In figure 5G, the same comments as for figure 1E apply, but clusters 2, 4, 7 and 10 are missing.

- Similar to our response to comments regarding Figure 1E and Figure 3D, we have updated Figure 5F and 5G. Figure 5F now more clearly displaying all of the clustering in spleen tissue from μ -PRMT5/TCL1 and μ -TCL1 mice. From these, we were interested in further evaluating gene expression in B cell clusters (B cells defined in Supplemental figure 4E), and thus deemed clusters 5.1, 5.3, 5.5, 5.6, 5.8, and 5.9 to be relevant in this study and shown in the heatmap in Figure 5G. These clusters were deemed relevant as we were interested in the differential expression of genes within B cell populations, rather than differential expression of genes in μ -PRMT5/TCL1 and μ -TCL1 mice B cells when compared to other T and monocyte cells in the spleen tissue. The results section has been updated to better describe our intent.

Additional Revisions:

1. Author Ethan C. Whipp has been listed as a co-first author on this manuscript to reflect overall contribution to this project.
2. Authors Selen A. Yilmaz and Kyle Shin were added to the manuscript to reflect contribution to these revisions.
3. Clarification was added to the chemical naming of compound PRT382:
 “[...] PRT382 is distinguishable by a **3,4-dichlorophenyl** at the 5' position of the ribose and a **methoxyimine at the 4' position of the pyrrolo pyrimidin moiety.**”

REVIEWER COMMENTS

Reviewer #4 (Remarks to the Author):

The authors have incorporated most of the suggestions. They have performed new 4C experiments with alternative restriction enzymes and have added replicates to the RNAseq experiments. Overall this has improved the manuscript.

Reviewer #4

1. The authors have incorporated most of the suggestions. They have performed new 4C experiments with alternative restriction enzymes and have added replicates to the RNAseq experiments. Overall this has improved the manuscript.

- We thank this reviewer for carefully reviewing our manuscript and the overall positive remarks.